# TANDEMFOILSET: DATASETS FOR FLOW FIELD PREDICTION OF TANDEM-AIRFOIL THROUGH THE REUSE OF SINGLE AIRFOILS

**Wei Xian Lim**[*]
Emerging nanoscience Research Institute
Nanyang Technological University
50 Nanyang Ave, Singapore 639798
weixian001@e.ntu.edu.sg

**Loh Sher En Jessica**[*]
College of Computing and Data Science
Nanyang Technological University
50 Nanyang Ave, Singapore 639798
lohs0037@e.ntu.edu.sg

**Zenong Li**
College of Computing and Data Science
Nanyang Technological University
50 Nanyang Ave, Singapore 639798
lize0018@e.ntu.edu.sg

**Thant Zin Oo**
College of Computing and Data Science
Nanyang Technological University
50 Nanyang Ave, Singapore 639798
tzoo@khu.ac.kr

**Wai Lee Chan**[†‡]
School of Mechanical and Aerospace Engineering
Nanyang Technological University
50 Nanyang Ave, Singapore 639798
wailee_chan@sutd.edu.sg

**Adams Wai-Kin Kong**[§]
College of Computing and Data Science
Nanyang Technological University
50 Nanyang Ave, Singapore 639798
AdamsKong@ntu.edu.sg

## ABSTRACT

Accurate simulation of flow fields around tandem geometries is critical for engineering design but remains computationally intensive. Existing machine learning approaches typically focus on simpler cases and lack evaluation on multi-body configurations. To support research in this area, we present **TandemFoilSet**: five tandem-airfoil datasets (4152 tandem-airfoil simulations) paired with four single-airfoil counterparts, for a total of 8104 CFD simulations. We provide benchmark results of a curriculum learning framework using a directional integrated distance representation, residual pre-training, training schemes based on freestream conditions and smooth-combined estimated fields, and a domain decomposition strategy. Evaluations demonstrate notable gains in prediction accuracy. We believe these datasets will enable future work on scalable, data-driven flow prediction for tandem-airfoil scenarios.

## 1 INTRODUCTION

In engineering design, simulating flow around geometries using computational fluid dynamics (CFD) is a critical step for component modelling. Studying partial differential equations (PDEs) with intricate, multi-body geometries is particularly important for applications like high-lift aircraft wings (Rumsey & Ying, 2002) and wind farm wake interactions (Deskos et al., 2020). Due to the complexity of these problems, analytical solutions are generally unavailable. In practice, such complex geometries can be built by assembling simpler shapes, with the downside of increased mesh resolution and domain size requirements, leading to high computational cost (Spalart & Venkatakrishnan, 2016).

---

[*]Equal contribution
[†]Corresponding author for fluid dynamics
[‡]Currently at Engineering Product Development, Singapore University of Technology and Design, Singapore
[§]Corresponding author for machine learning

Previous work on accelerating flow field prediction using neural networks (NNs) has largely focused on single-object cases with abundant data (Pfaff et al., 2021; Lino et al., 2021; Fortunato et al., 2022; Cao et al., 2023). Industries such as aerospace and marine have previously generated simulation data for single-body geometries like airfoils and hydrofoils, presenting an opportunity to exploit existing datasets for more efficient prediction of flow fields around multiple geometries. However, the extension to interacting multi-body systems, such as tandem configurations, a sequentially arranged airfoil setting, remains underexplored. Such a configuration is representative of many engineering applications, including compressor blades in turbomachinery (McGlumphy et al., 2007), unmanned aerial vehicles (Yin et al., 2021; Okulski & Ławryńczuk, 2022), hydrofoil systems for maritime vessels (Maram et al., 2021), and race car engineering (Azmi et al., 2017).

To address this gap, we make the following contributions:

- **TandemFoilSet** [1]: A collection of five high-fidelity tandem-airfoil flow field datasets paired with four single-airfoil datasets under various flow settings. It is the first to comprehensively capture interactions between front and rear airfoils, providing a foundation for benchmarking data-driven surrogate models in tandem-airfoil aerodynamic environments.
- **Novel benchmark curriculum learning scheme** in which models are first trained on single-airfoil flow fields and then fine-tuned on tandem-airfoil configurations in a memory-efficient multi-network approach which, on average, outperforms the baselines by $65\%$.
- **First use of freestream condition** (Ferm, 1990) as a physics prior for residual pre-training and smooth combination of single-airfoil flow predictions, enabling reduced training complexity and improved accuracy.
- **Extension of directional integrated distance (DID)** (Jessica et al., 2024) to represent two-body flow fields, effectively encoding directional pressure and velocity distributions in tandem-airfoil configurations. This is the first application of DID in setups with more than one object.

We believe that these datasets and benchmark results will enable further research into data-driven prediction of flow around tandem objects.

## 2 Preliminaries and Related Work

### 2.1 Preliminaries

**Graph Construction**    The CFD simulation mesh $M$ is represented as a graph $G = (V, E)$, where $V$ and $E$ are the sets of nodes and edges, respectively. The mesh nodes are directly represented as graph nodes $i \in V$, and the faces between them as bi-directional edges $(i, j), (j, i) \in E$.

**Geometry Representations**    This work employs the shortest vector (SV) and DID methods introduced by Jessica et al. (2024) to encode geometries as input node features. The SV represents the shortest vector from a node to the geometry, while each DID value captures the average distance between the node and the geometry within a given angular segment, up to a maximum distance $d_{max}$. These were originally demonstrated only on single-object geometries. The DID was numerically calculated via the procedure in Alg. 1 in Appx. E. Although extending the DID to multiple geometries is conceptually straightforward, the numerical calculations grow significantly more complex with each additional object. These challenges are discussed in Appx. E.

**Solid Bodies within a Flow**    Flow around solid bodies is a fundamental problem in fluid mechanics and, in applications such as aircraft design, is typically modelled within an infinite flow region (Wu, 1976). In such settings, the influence of solid bodies can be viewed as a localised force field that accelerates or diverts the flow, with the velocity gradually recovering towards a freestream condition at large distances (Fan & Li, 2019). Mathematically, the flow velocity $\boldsymbol{U}$ can be decomposed as:

$$\boldsymbol{U} = \boldsymbol{U}_\infty + \boldsymbol{U}',$$

where $\boldsymbol{U}_\infty$ is the freestream condition and $\boldsymbol{U}'$ the flow perturbation induced by the body. This decomposition is analogous to transformations from inertial to body frames (Speziale, 1998), and

---

[1] The datasets and the generation code can be accessed here.

is applicable across turbulent flows (Speziale, 1998) and potential flows (Collicott et al., 2017). This underlying principle informs the design of the smooth-combining and residual pre-training techniques in Secs. 4.1 and 4.3, which aim to enhance learning efficiency when leveraging freestream conditions. The datasets constructed reflect these flow characteristics to support future studies exploring freestream-informed models.

## 2.2 RELATED WORK

Previous NN-based strategies for accelerating CFD simulations have explored various architectural adaptations. Graph neural networks (GNNs) have become a leading approach, outperforming MLPs and CNNs (Pfaff et al., 2021; Ogoke et al., 2020; Bonnet et al., 2022). Chen et al. (2021) improved performance by incorporating both node and edge features. Encoder-processor-decoder architectures with such graph convolutions were subsequently adopted by Pfaff et al. (2021) and Sanchez-Gonzalez et al. (2020), and further improvements in scalability and information propagation speed were achieved through multi-scale designs (Lino et al., 2021; Fortunato et al., 2022; Cao et al., 2023; Gladstone et al., 2024). These developments largely shaped network and convolutional design for CFD prediction, though relatively little attention has been given by these to leveraging existing datasets or embedding physical priors.

Physical priors have been incorporated through different means by other works. Raissi et al. (2019) integrated physical equations directly into the training loss of NNs, a technique later extended to graph convolutional networks (GCNs) (Würth et al., 2024). While effective in reducing reliance on training data, these approaches were evaluated primarily on scenarios without internal objects. Physics-based solvers were also incorporated into the model itself (Obiols-Sales et al., 2020; De Avila Belbute-Peres et al., 2020; Lim et al., 2024). Physics-inspired features and loss terms were combined (Libao et al., 2023), building on Pfaff et al. (2021). However, none of these approaches address the interaction of flows between two objects in tandem configurations. The concept of using coarse simulations as a basis for improved predictions has also been explored using CNNs (Kochkov et al., 2021). More recently, Jessica et al. (2024) and Lim et al. (2025) applied residual learning techniques, previously used in image super-resolution (Zhang et al., 2018; Yang et al., 2019), to GCNs for fluid prediction, though this too relied on coarse simulations. Domain decomposition for grid-based simulations has been addressed using fixed-size subdomains and CNN-based architectures (Mao et al., 2024), but did not directly address mesh-based scenarios with tandem shapes.

Finally, public datasets only feature single-body flows (Bonnet et al., 2022; Schillaci et al., 2021; Liu et al., 2024; Agarwal et al., 2024; Yang et al., 2024) and have limited focus on interactions between multiple bodies like tandem airfoils. On the other hand, multi-geometry data like that of Bartoldson et al. (2023) were not paired with their single-object counterparts. In all, the existing datasets did not facilitate research of reusing single-object data to improve tandem-object flow field prediction.

## 3 TANDEMFOILSET

**TandemFoilSet** is a comprehensive collection of nine 2D datasets that span a wide range of configurations across both aerial and ground-effect environments. Built primarily from four-digit NACA airfoil ($MPXX$), the datasets vary systematically to cover diverse aerodynamic characteristics. They are categorized into two domains: seven air- and two land-based datasets.

The aerial datasets simulate cruise and takeoff conditions. **Cruise** datasets assume ideal freestream (far from ground), while **Takeoff** datasets include ground effect. For both, NACA parameters are uniformly sampled as $M, P \in [0, 6]$ and $XX \in [5, 25]$, covering both symmetric and asymmetric airfoils. The land-based **Race Car** datasets model inverted airfoils as simplified 2D spoilers operating near the ground. Their NACA parameters are also uniformly sampled with $M \in [2, 9]$, $P \in [2, 8]$, and $XX \in [5, 20]$. Additionally, five high-lift airfoils (CH10, E423, FX74-CL5-140, LA5055, S1210) were included to expand aerodynamic diversity. With 8104 cases, including 4152 tandem-airfoil configurations, **TandemFoilSet** is the first public tandem-airfoil dataset of its kind, supporting advanced studies in aerodynamics modelling and machine learning-based flow prediction.

Table 1: Total number of cases, average grid cells and the range of $Re$ and AOA as well as stagger, gap, and height (see Fig. 1) normalised by the chord length, for **TandemFoilSet**. *Asterisk marks indicate the datasets are in tandem-airfoil configuration. More are available in Appx. A.

| DATASET | CASES | AVE. CELLS | $Re$ | AOA [°] | STAGGER | GAP | HEIGHT |
|---|---|---|---|---|---|---|---|
| SINGLE AOA= 0° | 1014 | 122788 | 500 | 0 | - | - | - |
| *CRUISE AOA= 0° | 784 | 351315 | 500 | 0 | [0.5, 2] | [−0.4, 0.4] | - |
| SINGLE AOA= 5° | 1014 | 122788 | 500 | 5 | - | - | - |
| *CRUISE AOA= 5° | 784 | 351315 | 500 | 5 | [0.5, 2] | [−0.4, 0.4] | - |
| *TAKEOFF | 784 | 271316 | 500 | 5 | [0.5, 2] | [−0.2, 0.6] | [0.4, 1] |
| SINGLE RANDOM | 1025 | 111370 | $[10^5, 5 \times 10^6]$ | [−5, 7] | - | - | - |
| *CRUISE RANDOM | 900 | 210181 | $[10^5, 5 \times 10^6]$ | [−5, 6] | [0.5, 2] | [−0.8, 0.8] | - |
| SINGLE INVERTED | 899 | 87108 | $[10^5, 5 \times 10^6]$ | [−10, 0] | - | - | [0.1, 1.1] |
| *RACE CAR | 900 | 130276 | $[10^6, 5 \times 10^6]$ | [−10, 0] | [−0.5, 0.05] | [0.05, 0.1] | [0.1, 1.1] |

## 3.1 DATASET CATEGORISATION AND CONFIGURATION

The datasets are categorised based on flow conditions, providing a mix of low and high Reynolds number, $Re$, scenarios. The **Cruise** and **Takeoff** datasets are designed to capture stable aerodynamic characteristics under fixed and low $Re = 500$ conditions, with angles of attack (AOA) set at 0° or 5°. The **Takeoff** dataset incorporates ground effect, with height variations between the airfoil and the ground, simulating conditions typical of UAV takeoff as illustrated in Fig. 1(a). These configurations enable the analysis of steady aerodynamic behaviours under controlled flow conditions. In contrast, the **Cruise Random** and **Race Car** datasets operate under high $Re$, randomly selected from the range $[10^5, 5 \times 10^6]$, and AOA varying between $[−10°, 7°]$. The **Race Car** dataset captures aerodynamic interactions near the ground, representing the downwash effect caused by inverted airfoils, a critical feature in high-performance automotive applications (Fig. 1(b)). It includes a mix of standard NACA airfoils and five high-lift airfoils, enhancing the diversity of aerodynamic shapes.

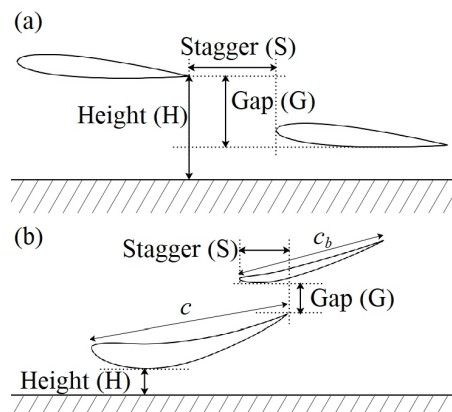

Figure 1: The schematic drawings of tandem-airfoil configuration with ground effect, such as (a) **Takeoff** and (b) **Race Car**.

The datasets are summarised in Tab. 1. The near-field $x-$component velocity, $u$, and kinematic pressure, $p$, contours of selected cases are shown in Fig. 5. In experiments, the datasets are uniformly sampled in a 8:1:1 ratio for training, validation, and test sets, respectively, unless otherwise stated.

## 3.2 MESHING AND SIMULATION SETUP

Two meshing methods were employed to simulate the aerodynamic properties of the airfoils. For single airfoils, the *blockMesh* utility from OpenFOAM-v2112 (Jasak et al., 2007) was used to create a C-grid hexahedral mesh. For tandem-airfoils, the overset meshing method (Benek et al., 1986) with a combination of background meshes and overset grids generated by Pointwise (Karman & Wyman, 2019) was used. This allowed flexible manipulation of airfoil orientation while maintaining quality.

For low $Re$ cases, the freestream velocity was 0.146 m/s with a kinematic viscosity of $2.92 \times 10^{-4}$ m²/s, resulting in $Re = 500$. Mesh resolution was tuned to achieve boundary layer resolution of $y^+ \approx 1$, ensuring accurate modelling of near-wall flow characteristics. High $Re$ simulations setups have higher grid density to accommodate the high flow velocities. Further details are in Appx. A.

Simulations were done using the *simpleFoam* solver for single airfoil setups and *overSimpleFoam* for tandem-airfoil setups. Both used the $k − \omega$ SST turbulence model (Menter et al., 2003), employing the SIMPLE algorithm (Caretto et al., 2007) for steady-state incompressible Navier–Stokes (NS) equations. Detailed configurations of solver settings and turbulence model parameters are in Appx. A.

### 3.3 DATASET VALIDATION AND VERIFICATION

For all datasets, we performed mesh independence studies to guarantee numerical convergence. For high $Re$ datasets, we validated simulation results against experimental data from literature, ensuring that the chosen mesh configuration maintained high accuracy across a range of AOAs. For tandem-airfoil configurations, we replicated previous studies on tandem-airfoil ground effects to benchmark the simulation setup, confirming consistency with established results. These comprehensive checks ensure that the dataset accurately represents aerodynamic phenomena relevant to both low and high $Re$ conditions. The full details of the procedures and results are in Appx. A.

## 4 BENCHMARKING SETUP

This section introduces the schemes developed to optimise the use of existing single-airfoil datasets for predicting flow around tandem-airfoil configurations—a traditionally resource-intensive task. The overall framework is illustrated in Fig. 2 and proceeds as follows:

1. A NN is pre-trained on single-airfoil cases to predict flow fields from geometry representations and boundary conditions, with freestream conditions serving as prior estimate fields for residual training.

2. The pre-trained network is used to predict flow fields around single airfoils. The $x$-velocity, $y$-velocity, and pressure fields are then smoothly combined to generate preliminary tandem-airfoil flow fields.

3. The NNs for the multiple-geometry task are initialised with the weights of the pre-trained single-geometry model.

4. These networks are then residually trained to predict tandem airfoil flow fields, using the smoothly combined fields as estimates.

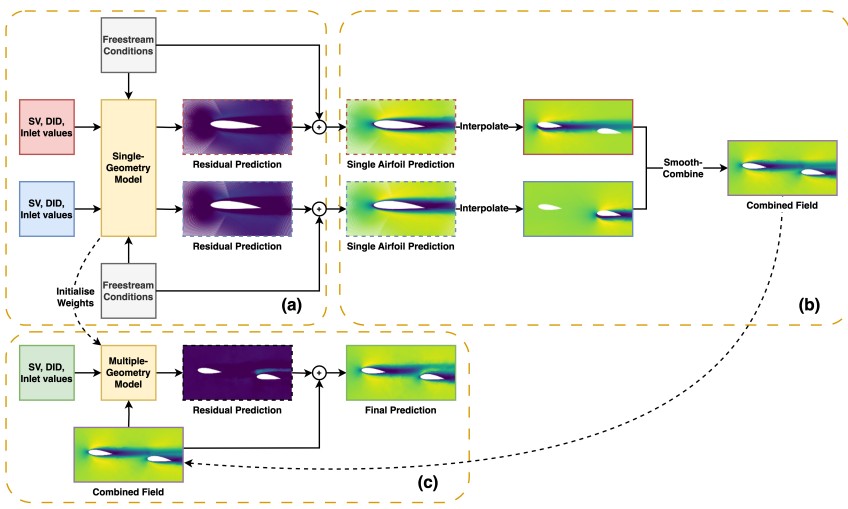

Figure 2: Overview of the benchmarking method, using (a) freestream based residual pre-training, (b) smooth-combining, and (c) combined field based residual training. Here, the multiple-geometry model is portrayed as a single network for simplicity. Note that a multi-NN as shown in Fig. 4 may be used instead.

The multiple-geometry model may consist of multiple neural networks. Full technical details of each stage are provided to establish a benchmark for future research on single-to-tandem flow prediction.

### 4.1 SMOOTH-COMBINING

This section introduces the procedure for combining multiple fields to obtain a cost-effective amalgamate field. Let $\boldsymbol{y}_1, \ldots, \boldsymbol{y}_L$ denote the $L$ fields to be combined. The combined field $\widetilde{\boldsymbol{y}}$ at node $i$ is then computed as:

$$\widetilde{\boldsymbol{y}}(i) = \boldsymbol{\gamma_1}(i) \cdot \boldsymbol{y}_1(i) + \cdots + \boldsymbol{\gamma_L}(i) \cdot \boldsymbol{y}_L(i) \,. \tag{1}$$

Here, $\gamma_1, \ldots, \gamma_L$ represents the weight of the respective original fields in the combined field. These will be assigned based on their absolute deviation from a reference field $y_0$:

$$\gamma_l(i) = \frac{|y_0(i) - y_l(i)|}{|y_0(i) - y_1(i)| + \cdots + |y_0(i) - y_L(i)|} \, . \tag{2}$$

At nodes $i$ where all fields do not deviate from the reference field, or $y_1(i) = \cdots = y_L(i) = y_0(i)$, the weights can be set to $\gamma_1(i), \ldots, \gamma_L(i) = 1/L$. This results in the final combined field exactly matching the reference field at these nodes, i.e., $\widetilde{y}(i) = y_0(i)$.

**Deviation from Freestream**  When combining flow fields (see Fig. 2(b)), we can assign weights to each field based on their deviation from the freestream. Let $y_1, y_2 = U_1, U_2$ be two flow fields such as the x-velocity fields. Let $y_0 = U_\infty$ be the freestream flow field with no internal geometry. Then:

$$\widetilde{U}(i) = \gamma_1(i) \cdot U_1(i) + \gamma_2(i) \cdot U_2(i) \, ,$$
$$\gamma_l(i) = \frac{|U_\infty(i) - U_l(i)|}{|U_\infty(i) - U_1(i)| + |U_\infty(i) - U_2(i)|} = \frac{|U'_l(i)|}{|U'_1(i)| + |U'_2(i)|} \, .$$

The approach is guided by the physical principles outlined in Sec. 2.1, which suggest that the influence of a solid body in a flow field can be conceptualised as deviation from the freestream, $U'$. Hence, employing weights based on these deviations creates a combined field that preserves the influences of both airfoils. This novel concept offers both accuracy and efficiency due to its simplicity and the extra-low computational cost of the freestream conditions.

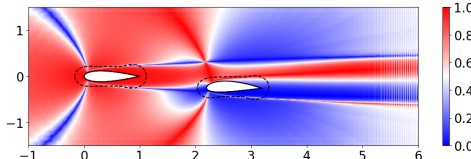

Figure 3: $\gamma_1$ for the front airfoil (NACA 0024), as described in Eqn. (2). Note that $\gamma_2$ for the aft airfoil (NACA 4424) is given by $1 - \gamma_1$.

Figure 3 provides an example of the resulting weights. Note that, where the flow fields are very similar but not equal to the freestream, the weights may still significantly favour the field that is most different from the freestream, such as in the blue area close to the front (left) airfoil.

## 4.2 DID CALCULATION FOR MULTIPLE OBJECTS

As mentioned, the SV and DID are incorporated into the neural network inputs as geometry representations. While both were utilised in previous applications, this marks the inaugural use of DID in a multiple-object scenario. Numerically calculating DID values using the original algorithm becomes progressively more complex and time-consuming with the addition of each object.

**Deviation from Maximum**  We propose an alternative procedure that capitalises on the smooth-combining scheme, using the deviation from the maximum value $d_{max}$ as weights. Here, $y_0 = d_{max}$, and $y_1, \ldots, y_L$ are the individual DID fields to be combined. This alternative method is detailed in Alg. 2 in Appx. E, providing an estimated DID representation of multiple geometries in a significantly reduced time-frame, ensuring computational efficiency of DID in a multi-object setting.

## 4.3 RESIDUAL TRAINING

In this section, we will introduce two residual training schemes, one supporting the utilisation of pre-training, and the other capitalising on the smooth-combining techniques outlined in the previous section. Residual training, extensively used in image super-resolution, involves utilising an estimate to ease the learning. Let the network output be $\widehat{U}$. Instead of directly predicting the flow field $U_{gt}$, the model is trained to predict the residual field $U_{gt} - \widetilde{U}_{est}$ and minimise the loss function,

$$\widetilde{\mathcal{L}} = \alpha \left\{ \mathcal{L}\left(U_{gt}, \widehat{U} + \widetilde{U}_{est}\right) \right\}_{\text{boundary cells}} + \left\{ \mathcal{L}\left(U_{gt}, \widehat{U} + \widetilde{U}_{est}\right) \right\}_{\text{internal cells}} \, , \tag{3}$$

where $\widetilde{U}_{est}$ represents an estimated flow field and $\alpha$ is a weight parameter. In CFD cases, this estimate often takes the form of a cost-effective lower-resolution simulation result (Jessica et al., 2024).

**Freestream Conditions**   Rather than a lower-resolution result, this paper suggests an innovative approach: using freestream conditions as an estimate for single geometries, or setting $\widetilde{U}_{est} = U_\infty$. This concept, like the smooth-combining procedure, aligns with the physical principles discussed in Sec. 2.1 that assert that freestream conditions should serve as a reliable estimate for the majority of the field. In contrast to lower-resolution fields, freestream conditions do not need to be derived from any physics-based simulator, so its computational cost is minimal. The residually pre-trained network is used not only to predict the flow fields of single airfoils for smooth-combining, but also to initialise the weights of the network for predicting tandem-airfoil. This initialisation can improve the final prediction performance. Figure 2(a) illustrates the freestream based residual pre-training.

**Combined Flow Fields**   To estimate the flow field of tandem-airfoil, we propose employing combined flow fields, obtained through the smooth-combining procedure discussed in the previous section and illustrated in Fig. 2(b), or setting $\widetilde{U}_{est} = \widetilde{U}$. The combined flow field may still differ from the target tandem-airfoil flow field, but it provides a cost-effective estimate for improving the learning through residual training. The combined field-based residual training process is visually presented in Fig. 2(c), where it can be seen as part of the consolidated pre-training, smooth-combining, and residual training method. Both residual training procedures in the proposed method use different estimates, one based on the freestream conditions and the other on the combined flow fields, which are unlike previous methods based on low-resolution simulation results.

## 4.4   MULTI-NN INFERENCE PROCEDURE

Multi-NN inference involves training multiple NNs to predict CFD domains with tandem geometries. While the multiple-geometry model shown in Fig. 2(c) can be conceivably handled by a single NN, predicting a tandem-geometry case after being pre-trained on solely single-geometry instances would be particularly demanding. The multi-NN technique hence ensures that each NN exclusively predicts a field with at most one airfoil, mitigating these challenges.

To implement this procedure, the domain is partitioned into distinct sub-graphs. In our implementation, these partitions segregate the graph into front, back, upper, and lower flow fields, with overlapping regions to ensure continuity and synchronisation between the sub-fields. An individual NN is then trained to predict each sub-field. The detailed steps involved in the multi-NN inference procedure are depicted in Fig. 4, and can be summarised as follows:

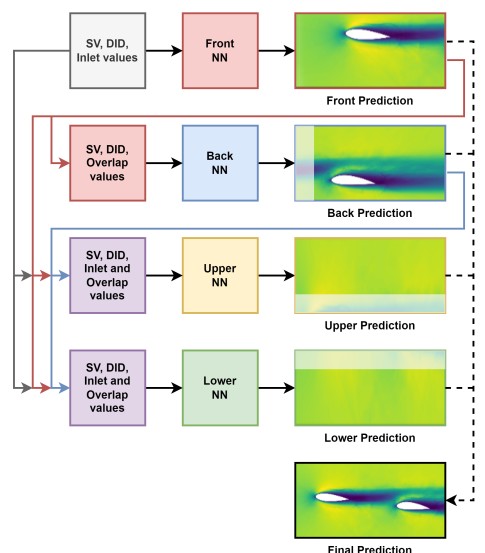

Figure 4: Inference of tandem-airfoil fields using a multi-NN inference procedure. Overlap regions (highlighted) are updated by the most recent NN.

1. The CFD domain is subdivided into front, back, upper, and lower flow fields. Inputs like the SV, DID or estimated fields are segmented accordingly.

2. The front flow field is predicted, with inlet values serving as an input feature for nodes along the inlet. The rest of the field will receive a zero-value here.

3. The predicted values within the overlap regions between the front and back sub-graphs are utilised as the corresponding input features to predict the back flow field.

4. The previous step is repeated, employing both the inlet values and appropriate overlap data from the front and back fields, to compute the upper and lower fields.

5. All sub-fields are combined to generate the final prediction for the complete flow field, with the latest and most updated prediction used for the overlap regions.

The dependency structure of our multi-NN approach is tied to geometric settings where the division of flow fields aligns with predefined boundaries and inlets. This structure ensures coherent

information propagation across sub-domains, similar to domain decomposition in CFD (Chan & Mathew, 1994). Upper and lower fields can be predicted using diverse strategies (e.g., freestream conditions, interpolation), to reduce the required NNs. Sub-domains may even be omitted if deemed unnecessary, providing flexibility for diverse CFD applications, similar to domain decomposition customisation (Lim et al., 2023). This procedure is then applied to all experiments conducted in this work, except the single-NN variants in Experiment 3, which evaluate the effectiveness of Multi-NN.

## 5 BENCHMARKING RESULTS

In this section, we evaluate the effectiveness of our schemes through four experiments, assessing: (i) the multi-object DID representation method, (ii) the smooth-combining, pre-training, and residual training schemes, (iii) the multi-NN inference, and (iv) the overall framework in varying flow conditions. We employ the five datasets in **TandemFoilSet**, and two highly influential GNN architectures: MeshGraphNet (MGN) (Pfaff et al., 2021) and invariant edge-GCNN (IVE) (Chen et al., 2021), to test performance across various flow conditions and convolutional types.

### 5.1 EXPERIMENT 1: MULTI-OBJECT DID EFFECTIVENESS

In the first experiment, to assess the effectiveness of the multiple-object DID representation, we compare MGN performance with and without DID on the **Cruise AOA=0°** and **Takeoff** datasets. No additional methods are applied. As shown in Tab. 2, the DID significantly improves performance for both datasets, with over $91\%$ reduction in MSE test loss for the **Cruise AOA=0°** case. This suggests that the DID representation remains effective when estimated using the smooth-combining method.

Table 2: MSE ($\times 10^{-2}$) performance evaluation of DID.

| MODEL / DATASET | CRUISE AOA=0° | | TAKEOFF | |
|---|---|---|---|---|
| MGN + SV | $11.51 \pm 5.48$ | - | $8.17 \pm 3.37$ | - |
| MGN + SV + DID | $\mathbf{1.03 \pm 0.60}$ | $\mathbf{91.1\%}$ | $\mathbf{3.74 \pm 1.95}$ | $\mathbf{54.22\%}$ |

### 5.2 EXPERIMENT 2: ABLATION STUDY

The second set of experiments evaluates the effectiveness of pre-training, smooth-combining, and residual training in various geometric shapes and configurations on both MGN and IVE. Hence, three datasets with fixed and low $Re$ conditions were considered: **Cruise AOA=0°**, **Cruise AOA=5°**, and **Takeoff**. We compare the baseline against the following four combinations of the suggested schemes:

- **PRE**: A single-airfoil model is pre-trained, and its weights are used to initialise the networks of the main tandem-airfoil model before training.

- **PRE-FREE + COMB**: Additionally, the freestream condition is used as an input feature to the pre-trained single-airfoil model, and the combined flow fields from the single-airfoil network are used as input features to the main tandem-airfoil model.

- **RES-FREE + RES-COMB**: The single-airfoil model is residually trained using the freestream conditions as the estimate fields. Likewise, the main tandem-airfoil model is residually trained using the combined flow fields as estimate fields. However, its weights are not initialised with those of the single-airfoil model.

- **PRE-RES-FREE + RES-COMB**: The pre-trained single-airfoil model is freestream-based residually trained. Its weights are used to initialise the main combined field-based residually trained tandem-airfoil model.

The single-airfoil dataset with the same AOA value ($0°$ or $5°$) as the tandem-airfoil dataset is used. All models, including the baselines, incorporate both SV and DID. Likewise, multi-NN inference was used in all models. The outcome of this ablation study is shown in Tab. 3.

The importance of the smooth-combined field, as either an input or for residual training, is demonstrated by the last three models always outperforming the first two significantly, with up to $70\%$ improvement of accuracy over the MGN and IVE baseline models. While the best performance was occasionally achieved through pre-training or residual training alone, the comprehensive method with

Table 3: MSE $(\times 10^{-2})$ performance evaluation of ablation study. Average improvement is measured relative to baseline over three datasets. The best performance is in bold while the second best is underlined.

| MODEL / DATASET | CRUISE AOA=0° | | CRUISE AOA=5° | | TAKEOFF | | IMPROV. |
|---|---|---|---|---|---|---|---|
| MGN (BASELINE) | $1.03 \pm 0.60$ | - | $1.34 \pm 0.67$ | - | $3.74 \pm 1.95$ | - | |
| MGN + PRE | $1.04 \pm 0.62$ | -0.7% | $1.21 \pm 0.64$ | 10.1% | $3.69 \pm 2.01$ | 1.27% | 3.55% |
| MGN + PRE-FREE + COMB | $\mathbf{0.42 \pm 0.50}$ | **59.3%** | $0.74 \pm 0.58$ | 44.7% | $1.31 \pm 0.67$ | 64.9% | 56.3% |
| MGN + RES-FREE + RES-COMB | $0.49 \pm 0.67$ | 52.0% | $\underline{0.68 \pm 0.37}$ | 48.9% | $\underline{1.24 \pm 0.67}$ | 66.9% | 55.9% |
| MGN + PRE-RES-FREE + RES-COMB | $\underline{0.45 \pm 0.69}$ | 56.1% | $\mathbf{0.67 \pm 0.50}$ | **49.4%** | $\mathbf{1.12 \pm 0.57}$ | **70.0%** | **58.5%** |
| IVE (BASELINE) | $0.85 \pm 0.55$ | - | $1.05 \pm 0.69$ | - | $2.53 \pm 1.24$ | - | |
| IVE + PRE | $0.98 \pm 0.55$ | -16.0% | $1.05 \pm 0.59$ | -0.07% | $2.73 \pm 1.37$ | -7.74% | -7.94% |
| IVE + PRE-FREE + COMB | $0.60 \pm 0.40$ | 29.2% | $0.73 \pm 0.40$ | 30.0% | $0.89 \pm 0.39$ | 65.0% | 41.4% |
| IVE + RES-FREE + RES-COMB | $\underline{0.54 \pm 0.34}$ | 36.2% | $\mathbf{0.62 \pm 0.36}$ | **40.4%** | $\mathbf{0.77 \pm 0.36}$ | **69.7%** | **48.8%** |
| IVE + PRE-RES-FREE + RES-COMB | $\mathbf{0.52 \pm 0.29}$ | **38.9%** | $\underline{0.63 \pm 0.30}$ | 40.3% | $\underline{0.83 \pm 0.37}$ | 67.3% | **48.8%** |

Table 4: MSE $(\times 10^{-2})$ aerodynamic performance analysis of the airfoil boundary cells, lift, $c_l$, and drag, $c_d$, coefficients for **Cruise AOA=5°** and **Takeoff** datasets, using the comprehensive model relative to the baseline.

| MODEL / DATASET | CRUISE AOA=5° | | | TAKEOFF | | |
|---|---|---|---|---|---|---|
| | $\mathrm{MSE}_b$ | $\mathrm{MSE}_{c_l}$ | $\mathrm{MSE}_{c_d}$ | $\mathrm{MSE}_b$ | $\mathrm{MSE}_{c_l}$ | $\mathrm{MSE}_{c_d}$ |
| MGN (BASELINE) | $2.52 \pm 2.19$ | $\mathbf{0.05 \pm 0.07}$ | $0.32 \pm 0.29$ | $7.61 \pm 7.27$ | $0.15 \pm 0.38$ | $0.23 \pm 0.16$ |
| MGN + PRE-RES-FREE + RES-COMB | $\mathbf{1.06 \pm 0.84}$ | $0.05 \pm 0.08$ | $\mathbf{0.21 \pm 0.40}$ | $\mathbf{1.55 \pm 1.32}$ | $\mathbf{0.04 \pm 0.07}$ | $\mathbf{0.13 \pm 0.26}$ |
| IMPROVEMENT | 58.2% | 0% | 34.4% | 79.6% | 73.3% | 43.5% |

both schemes together shows the most consistent performance, often having errors within an order below the standard deviation of the best model. Overall, these results suggest that the freestream condition and combined-field based residual training both individually enhance the performance of models across various scenarios, but perform most consistently when used in conjunction with one another. As it is observed that the MGN models demonstrate a higher average improvement from the schemes at over 55%, subsequent experiments focused on the MGN architecture.

Also, to analyse the model's prediction on the aerodynamic performance of the tandem-airfoil, Tab. 4 presents the MSE of the airfoil boundary cells, as well as the lift and drag coefficients for **Cruise AOA=5°** and **Takeoff** datasets. The comprehensive method reduced error from the baseline by up to almost $80\%$, and showed superior capability in more complex flow fields as in the **Takeoff** dataset.

## 5.3 EXPERIMENT 3: MULTI-NN EFFECTIVENESS

In the third experiment, we assess the effectiveness of the multi-NN inference by comparing MGN performance in a single-NN versus a multi-NN setup, where the field is split into front and back sub-fields. The upper and lower fields were excluded due to memory limitations. The experiment uses just the **Cruise AOA=0°** dataset to showcase its performance. As in Tab. 5, multi-NN outperforms the single-NN in both models, suggesting that separate and specialised NN predictions enhance accuracy.

Table 5: MSE $(\times 10^{-2})$ performance evaluation of Multi-NN on **Cruise AOA=0°** dataset.

| MODEL / DATA SCHEME | SINGLE-NN | MULTI-NN | IMPROVEMENT |
|---|---|---|---|
| MGN + RES-FREE + RES-COMB | $1.66 \pm 1.71$ | $\mathbf{0.49 \pm 0.67}$ | 70% |
| MGN + PRE-RES-FREE + RES-COMB | $1.51 \pm 1.61$ | $\mathbf{0.45 \pm 0.69}$ | 70% |

## 5.4 EXPERIMENT 4: EFFECTIVENESS IN VARYING FLOW CONDITIONS

In the final experiment, we assess the model under varying conditions using the **Cruise Random** dataset through two sampling styles: Uniform and Extrapolation. In Extrapolation, the data reflecting the highest and lowest $5\%$ of the AOA, $Re$, Stagger, or Gap value range is used as the test set, while the training and validation sets are uniformly sampled from the middle $90\%$ range. In addition, the **Race Car** dataset is also used with uniform sampling to examine the model performance in sophisticated flow conditions. Table 6 shows that the MSE test losses are higher than the datasets with fixed and low $Re$ flow conditions (compared to Tab. 3), which is expected due to the varied conditions without an increase in dataset or model size. The proposed model still achieves up to $94\%$ and $65\%$ reductions in MSE test losses of **Cruise Random** and **Race Car** datasets, respectively, demonstrating its effectiveness.

Table 6: MSE performance evaluation on **Cruise Random** and **Race Car** datasets with varying flow conditions.

| MODEL / DATA SCHEME | UNIFORM | AOA | $Re$ | STAGGER | GAP | RACE CAR |
|---|---|---|---|---|---|---|
| MGN (BASELINE) | $1.79 \pm 1.38$ | $2.03 \pm 1.96$ | $4.85 \pm 1.82$ | $1.74 \pm 1.66$ | $1.95 \pm 1.68$ | $0.61 \pm 0.51$ |
| MGN + PRE-RES-FREE + RES-COMB | $\mathbf{0.10 \pm 0.13}$ | $\mathbf{0.18 \pm 0.24}$ | $\mathbf{0.36 \pm 0.53}$ | $\mathbf{0.13 \pm 0.17}$ | $\mathbf{0.14 \pm 0.20}$ | $\mathbf{0.21 \pm 0.29}$ |
| IMPROVEMENT | 94.4% | 91.1% | 92.6% | 92.5% | 92.8% | 65.6% |

## 5.5 EXPERIMENT 5: MULTI-TASK TRAINING WITH MIXED SINGLE- AND TANDEM-AIRFOIL DATASETS.

Beyond the proposed curriculum scheme, we also evaluate a simple multi-task setup in which a model is trained jointly on both single- and tandem-airfoil cases. Table 7 reports results for the **Cruise AOA=5°** and **Cruise Random** datasets under this setting. Compared to the baseline multi-task model, our proposed curriculum (PRE-RES-FREE + RES-COMB) model reduces the MSE values by at least one order of magnitude, corresponding to a 66.3% and 94.4% relative improvements for **Cruise AOA=5°** and **Cruise Random** datasets, respectively, showcasing its superior performance.

Table 7: MSE performance comparison between a multi-task setup and the proposed curriculum scheme for the **Cruise AOA=5°** and **Cruise Random** datasets.

| MODEL / DATASET | CRUISE AOA=5° ($\times 10^{-2}$) | CRUISE RANDOM |
|---|---|---|
| MGN + MULTI-TASK | $1.99 \pm 1.04$ | $1.80 \pm 1.34$ |
| MGN + PRE-RES-FREE + RES-COMB | $\mathbf{0.67 \pm 0.50}$ | $\mathbf{0.10 \pm 0.13}$ |
| IMPROVEMENT | 66.3% | 94.4% |

## 5.6 ADDITIONAL RESULTS

Additional experiments with a Transformer-based architecture, Transolver (Luo et al., 2025), which further confirm the robustness of our transfer-learning pipeline, are reported in Appx. J (see Tab. 26). Neural network prediction times showed a 76% reduction in wall time compared to simulations (Appx. C). A parametric study of DID parameters demonstrated that it remains robust and consistently outperforms the baseline (Appx. F). The smooth-combining method was evaluated against freestream and linear interpolation approaches, and showed to be the best performing (Appx. H). Model accuracy was validated across varying tandem-airfoil positions and flow parameters, with the method consistently outperforming the baseline (Appx. I). Navier–Stokes residual analysis confirmed high prediction accuracy, with residuals significantly below maximum thresholds (Tab. 25).

## 5.7 EXTENSION TO THREE-BODY CONFIGURATIONS

While our primary evaluation focuses on two-body tandem configurations, we additionally examine a more complex three-airfoil setting to probe the scalability of our geometry encoding and multi-NN training pipeline. Specifically, we construct a new three-airfoil tandem dataset under the same flow conditions as the **Cruise AOA=5°** subset and benchmark both the baseline MGN and our best variant (MGN + PRE-RES-FREE + RES-COMB). As detailed in Appx. K, our method achieves substantial error reductions in both overall and near-boundary metrics on this three-body dataset, providing direct evidence that the proposed framework extends beyond the original two-body configuration without architectural changes.

## 6 CONCLUSION

This paper introduces **TandemFoilSet**: five tandem-airfoil datasets with four paired single-airfoil datasets to support future research on flow prediction in multi-airfoil scenarios. It also utilises a smooth-combining technique that extends the DID representation to multi-object flow fields, a pre-training and residual training procedure leveraging freestream and combined fields, and a multi-NN inference scheme for tandem geometries. Our benchmark method was evaluated across two baseline settings and the datasets, demonstrating consistent improvements in prediction performance. Due to computational limitations, we were not able to test the method on three-dimensional objects or other geometries. For future work, the generalising capabilities of the method should be further tested in, for instance, multiple-airfoil cases like in a turbomachinery stage, and bluff body configurations such as cylinder or sphere.

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

## A    DATABASE GENERATION

### A.1    DATASET CHARACTERISTICS AND CONFIGURATION

In this study, **TandemFoilSet** is created to form a comprehensive collection of five tandem- and four single-airfoil datasets that explore a wide range of aerodynamic configurations in both aerial and ground-effect environments. They mainly consist of four-digit ($MPXX$) NACA airfoil, with systematically varied parameters to capture diverse aerodynamic characteristics. The datasets can be broadly categorized into two application domains: air- and land-based configurations.

For aerial configurations, five datasets simulate cruise and takeoff scenarios. The **Cruise** datasets assume an idealized freestream setting (i.e., height, $H \to \infty$), while the **Takeoff** datasets incorporate ground effect through finite height values. In both cases, the NACA airfoil parameters are uniformly sampled with camber magnitude $M \in [0, 6]$, camber location $P \in [0, 6]$, and thickness $XX \in [5, 25]$. These settings ensure a diverse distribution of symmetric and asymmetric airfoils across typical flight conditions.

For land-based configurations, two **Race Car** datasets model the behavior of inverted airfoils as simplified 2D representations of a spoiler operating near the ground. Here, the NACA parameters are also uniformly sampled, with $M \in [2, 9]$, $P \in [2, 8]$, and $XX \in [5, 20]$, reflecting common high-performance automotive aerodynamic profiles. To further enrich the dataset with complex, high-lift geometries, five additional non-NACA airfoils: CH10, E423, FX74-CL5-140, LA5055, and S1210, were included in the land-based configurations.

In general, the datasets can also be categorised based on their flow conditions, providing a balanced exploration of both low and high Reynolds number, $Re$, scenarios as tabulated in Tab. 1. The **Cruise** and **Takeoff** datasets are generated under fixed low Reynolds number conditions ($Re = 500$), with angles of attack (AOA) set at $0°$ or $5°$. The **Takeoff** dataset is further distinguished by the inclusion of ground effect, where height variations between the airfoil and the ground are introduced to capture near-ground aerodynamic interactions. In contrast, the **Cruise Random** and **Race Car** datasets are designed to capture high $Re$ conditions, where $Re$ is randomly sampled from the range $[10^5, 5 \times 10^6]$, and AOA is varied between $[-10°, 7°]$. The **Race Car** dataset is also subjected to ground effect, specifically configured to simulate the aerodynamic characteristics of high-performance vehicles, where inverted airfoils generate substantial downwash forces.

Figure 5 illustrates representative flow field visualisations for the **Cruise AOA=5°**, **Takeoff**, **Cruise Random**, and **Race Car** datasets, showcasing the diversity of aerodynamic interactions captured. The x-component velocity, $u$, and kinematic pressure, $p$, contours highlight the differences in flow behavior across configurations, including the noticeable ground effect in the **Takeoff** and **Race Car** datasets.

In short, **TandemFoilSet** encompasses 8104 cases, including 4152 tandem-airfoil configurations, making it one of the largest publicly available tandem-airfoil datasets for machine learning applications. The datasets are divided into training, validation, and test sets following an 8:1:1 ratio, ensuring a balanced representation of aerodynamic characteristics across all partitions. The distribution of key geometric and flow parameters across the training, validation, and test sets is illustrated in Figs. 6 to 9. For extrapolation experiments, data representing the top and bottom 5% of the AOA or $Re$ ranges are isolated for testing, while the remaining 90% are uniformly sampled to a $8 : 1$ ratio for training and validation.

### A.2    MESH GENERATION

For single airfoil datasets, a two-dimensional C-grid type of hexahedral mesh is adopted where the airfoil is situated at the center of the domain with a domain size of 20 chord lengths as shown in Fig. 10 (a), whereas the near-field views of the airfoil are illustrated on the left of Fig. 10, showing (b) symmetric and (c) chambered NACA airfoils. For low $Re$ datasets, a total number of 278 airfoil cells is used upon grid convergence study in Appx. A.4 with both the leading edge, $\Delta x_{LE}$, and trailing edge, $\Delta x_{TE}$ grid sizes equal to 5 mm. the first cell thickness adjacent to the airfoil wall, $\Delta x_{BL} = 1$ mm such that $y^+ \approx 1$ to fully resolve the boundary layer around the airfoil geometry without adopting any wall function. Near the domain boundary, the far-field grid resolution, $\Delta x_{\text{far-field}} = 0.2$ m with a total expansion ratio of 40 along the outwards direction, resulting in a total of 188 cells.

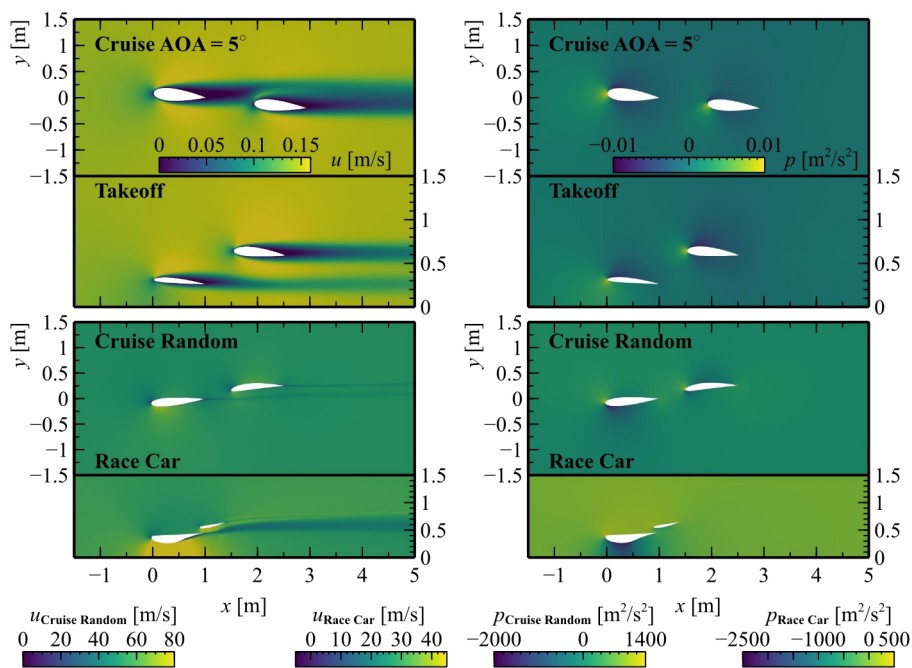

Figure 5: Examples of the (left) $x-$component velocity, $u$, and (right) kinematic pressure, $p$, contours of **Cruise AOA=5°**, **Takeoff**, **Cruise Random**, and **Race Car** datasets.

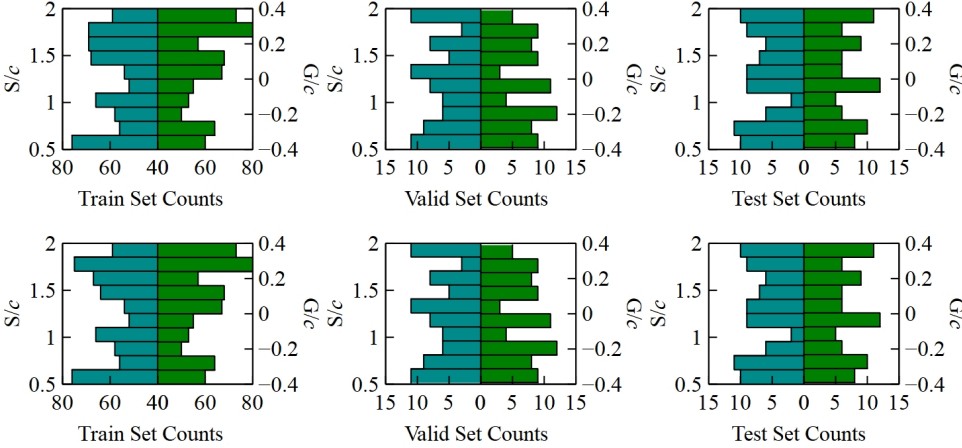

Figure 6: Distribution of stagger and gap parameters in train, valid, and test sets for (top row) **Cruise AOA=0°** and (bottom row) **Cruise AOA=5°** datasets.

For tandem-airfoil datasets, the the computational domain is divided into a background mesh generated by *blockMesh* and two independent overset meshes created using Pointwise (Karman & Wyman, 2019), which defines the grids surrounding the airfoil as presented in Fig. 11 for configuration with ground effect. To set up an overset computational domain, each subdomain is required to assign a 'zoneID' via *setFields* and/or *topoSet* utility to identify the types of mesh and further classify the cell types as shown in Fig. 11. For background mesh, its 'zoneID' is always set to 0, whereas other overset meshes can take any number but must contain an overset boundary patch where the information is shared. Figure 11(a) shows the red zones having the airfoil shapes identified as hole cells (i.e. omitted during simulation), surrounded by interpolated cells (white contour lines along the airfoils) of the background mesh to couple with the overset meshes. For the overset meshes, as illustrated in Fig. 11 (b), the hole (red) cells are identified where they are outside of the background mesh. The internal

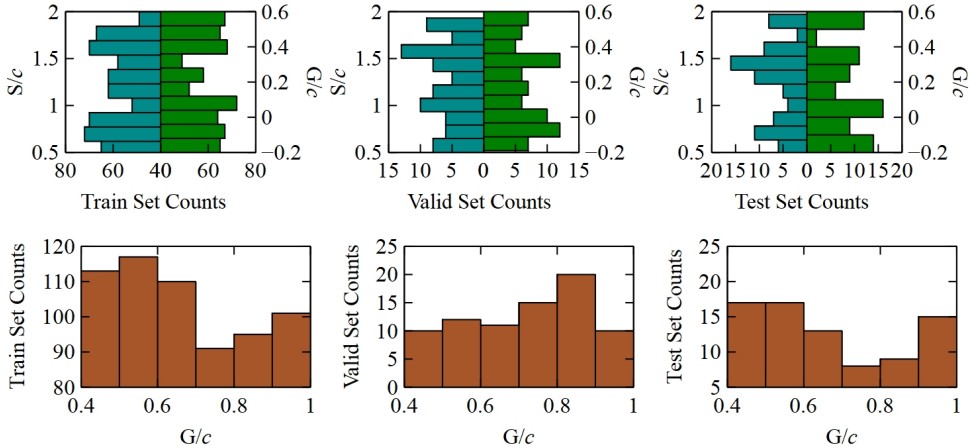

Figure 7: Distribution of stagger, gap, and height parameters in train, valid, and test sets for **Takeoff AOA=5°** dataset.

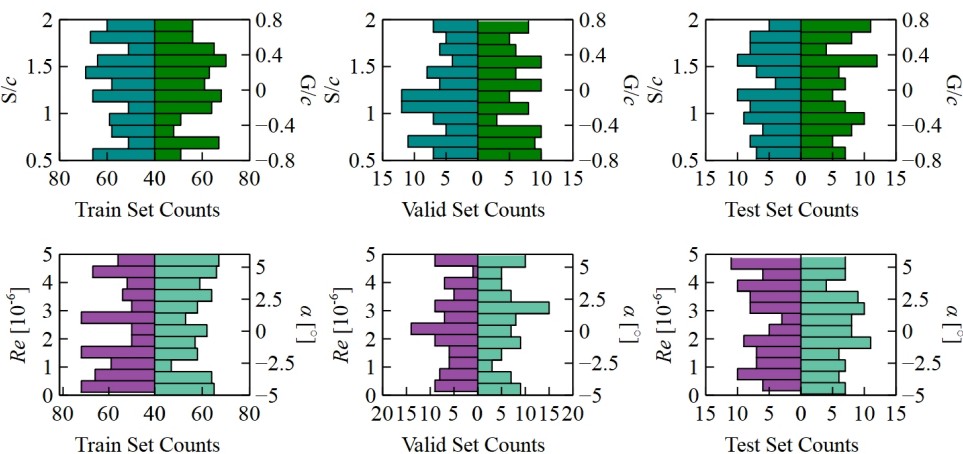

Figure 8: Distribution of stagger, gap, $Re$, and AOA parameters in train, valid, and test sets for **Cruise Random** dataset.

cells of overset meshes are labeled as calculated type (blue), whereas the overset boundary cells are labeled as interpolated type to couple with the background mesh.

For the **Takeoff** dataset, the background mesh with ground has a domain size of $24 \times 11$ m which consists of total 240000 structural grids with the smallest grid resolutions ($\Delta x = 0.01$ m) between $-0.5 < x < 4.5$ and $-1 < z < 0.5$ where the overset meshes are located at. For the **Cruise** datasets, the background domain size is larger, having 320000 cells with a domain size of $24 \times 20$ m, and the finest resolution grids range between $-0.5 < x < 4.5$ and $-0.5 < z < 0.5$. The grid size is geometrically increased to 0.02m within the extension of 0.5m from the finest grid region; thereafter, the grid size is then further increased to 0.2m geometrically to domain boundaries. 28 four-digit NACA airfoils are meshed as overset meshes using Pointwise (Karman & Wyman, 2019). This helps to reduce the total number of cells while retaining the quality of the mesh. Similar to the single airfoil dataset, the thickness of the first cell adjacent to the airfoil wall is $x_{\text{BL}} = 1$ mm to keep $y^+ \approx 1$. The wall boundary cells are then extruded normally until achieving a maximum grid size of 0.01m, resulting in approximately 15000 cells for each overset mesh.

The same meshing method used in the **Cruise** and **Takeoff** datasets is used in high $Re$ datasets (both **Cruise Random** and **Race Car** datasets). Each simulation contains approximately a total number of 798 airfoil cells with $\Delta x_{LE} = \Delta x_{TE} = 0.2$mm. $\Delta x_{BL} = 2 \times 10^{-5}$ m is used such that to reduce

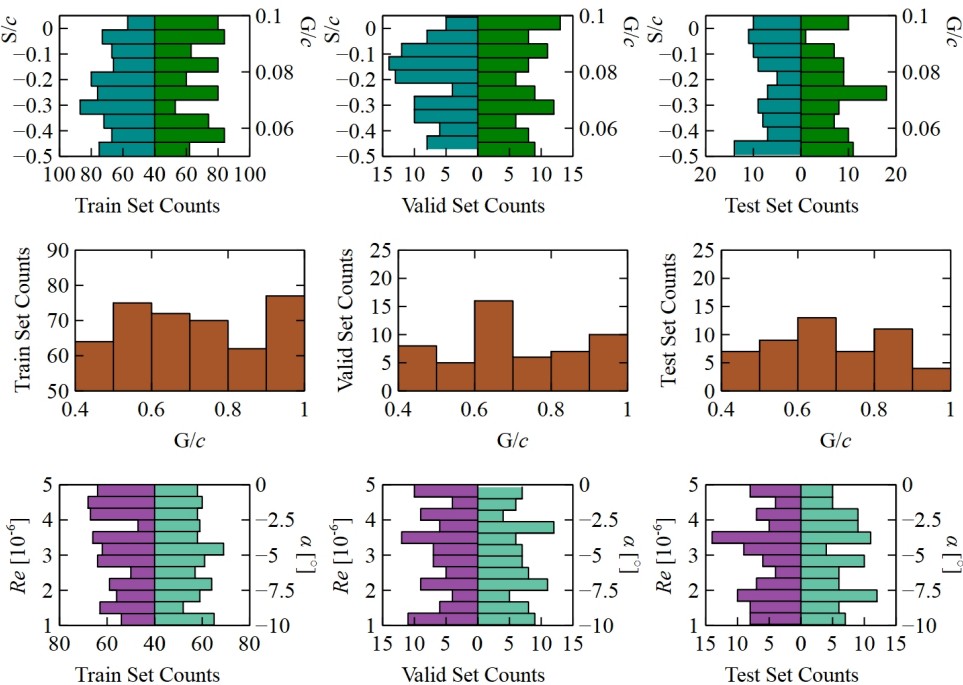

Figure 9: Distribution of stagger, gap, height, $Re$, and AOA ($\alpha$) parameters in train, valid, and test sets for **Race Car** dataset.

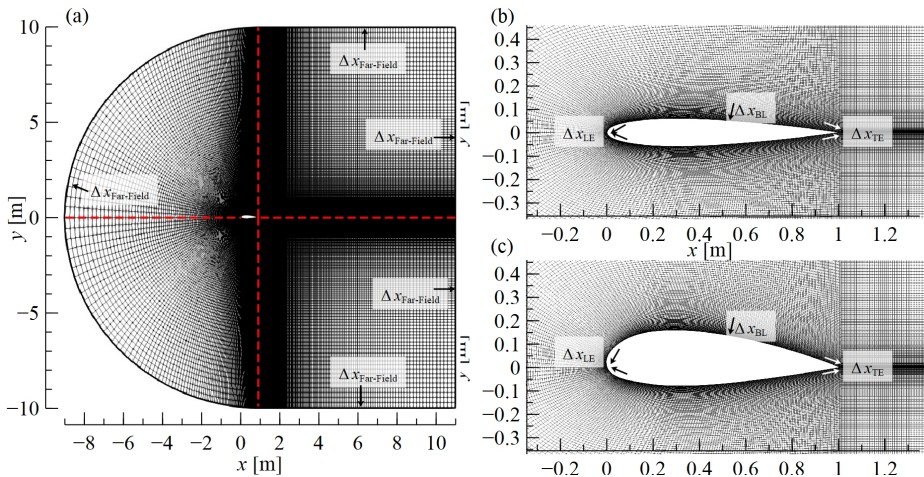

Figure 10: (a) A type C-grid mesh is adopted where the airfoil is situated at the center of the domain. The near-field views of the airfoil are illustrated on the left side, showing (b) symmetric and (c) chambered NACA airfoils

the non-orthogonality of the mesh and to ensure the $30 \leq y^+ \leq 100$ for the wall function to work efficiently. For the datasets that utilize the overset technique, the background mesh consists of a domain size of roughly $-21 \times 10$. or $-21 \times 20$ for simulation with or without ground effect. The flow and geometry interaction regions, where the overset meshes are placed, are refined to have a grid size of $\Delta x = \Delta y = 0.01$ m, then geometrically coarsened to domain boundaries at a cell-to-cell expansion ratio of 1.1.

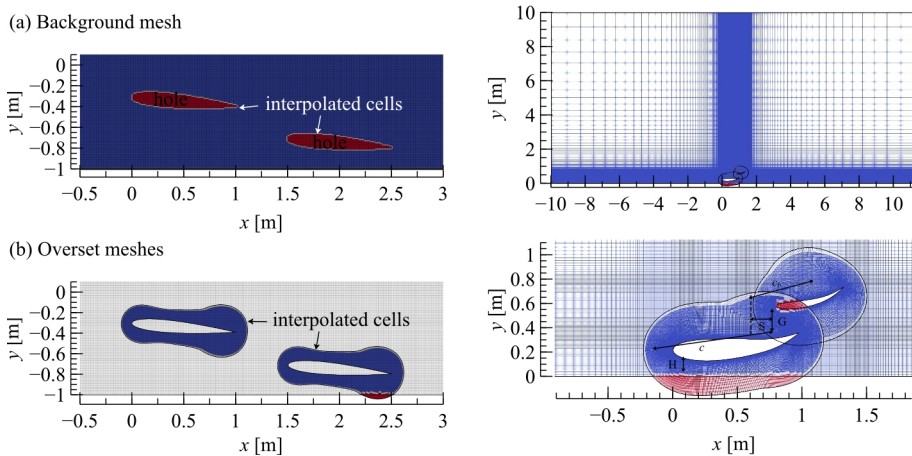

Figure 11: Calculated cells (blue cells), interpolated cells (white cells), and hole (red cells) defined for (a) background and (b) overset meshes. The left column represents the **Takeoff** meshes, while the right column represents the **Race Car** meshes.

### A.3    BOUNDARY CONDITIONS AND SOLVER SETTINGS

Tables 8 and 9 tabulate the applied boundary conditions on the steady-state flow variables, namely the ensemble-averaged velocity, $\mathbf{U}$ and kinematic pressure, $p$ which is divided by density, $\rho$, and the additional turbulent-related variables, such as turbulent kinetic energy, $k$, turbulent specific dissipation rate, $\omega$, and turbulent kinematic viscosity, $\nu_t$, are introduced due to adopting $k - \omega$ SST turbulence model. No slip condition is used on the airfoil wall, however, slip condition (normal velocity is zero) is applied on the ground to avoid the viscous boundary layer formation. Freestream boundaries refer to the outer domain boundary for which the positive flux (outflow) uses *zeroGradient* whereas the negative flux (inflow) uses *fixedValue*.

Table 8: The boundary conditions used for the datasets considering low $Re$ flow condition.

| Fields | Airfoil Walls | Freestream boundaries | Ground |
|---|---|---|---|
| $\mathbf{U}$ $[\text{m/s}]$ | *noSlip* | *freestreamVelocity* $\mathbf{U}_\infty$ | *slip* |
| $p$ $[\text{m}^2/\text{s}^2]$ | *zeroGradient* | *freestreamPressure* $p_\infty$ | |
| $k$ $[\text{m}^2/\text{s}^2]$ | *fixedValue* $k_{wall}$ | *freestream* $k_\infty$ | |
| $\omega$ $[\text{s}^{-1}]$ | *fixedValue* $\omega_{wall}$ | *freestream* $\omega_\infty$ | *zeroGradient* |
| $\nu_t$ $[\text{m}^2/\text{s}]$ | *fixedValue* 0 | *freestream* $\nu_{t\infty}$ | |

Table 9: The boundary conditions used for the datasets considering high $Re$ flow condition.

| Fields | Airfoil Walls | Freestream boundaries | Ground |
|---|---|---|---|
| $\mathbf{U}$ $[\text{m/s}]$ | *noSlip* | *freestreamVelocity* $\mathbf{U}_\infty$ | *slip* |
| $p$ $[\text{m}^2/\text{s}^2]$ | *zeroGradient* | *freestreamPressure* $p_\infty$ | |
| $k$ $[\text{m}^2/\text{s}^2]$ | *kLowReWallFunction* | *freestream* $k_\infty$ | |
| $\omega$ $[\text{s}^{-1}]$ | *omegaWallFunction* | *freestream* $\omega_\infty$ | *zeroGradient* |
| $\nu_t$ $[\text{m}^2/\text{s}]$ | *nutLowReWallFunction* | *freestream* $\nu_{t\infty}$ | |

For low $Re$ datasets, the freestream velocity, $U_\infty = 0.146$ m/s and the kinematic viscosity, $\nu = 2.92 \times 10^{-4}$ m$^2$/s such that the Reynolds number, $Re = 500$. Gauge pressure is considered, hence the freestream pressure, $p_\infty = 0$ m$^2$/s$^2$. The turbulent freestream values are chosen based on

recommendations in Menter (1994),

$$\nu_{t\infty} = 10^{-3}\nu \;, \tag{4a}$$

$$\omega_\infty = \frac{C_\omega \mathbf{U}_\infty}{L} \;, \tag{4b}$$

$$k_\infty = \nu_{t\infty}\omega_\infty \;, \tag{4c}$$

where $L$ is the approximate length of the computational domain, $C_\omega = 2$ for thin airfoil, $XX \geq 20$, and $C_\omega \geq 20$ for thick airfoil, $XX > 20$. For the wall boundary condition, the following is recommended,

$$k_{wall} = 0 \;, \tag{5a}$$

$$\omega_{wall} = 10\frac{6\nu}{\beta_1 y_{BL}^2} \;, \tag{5b}$$

where $\beta_1 = 0.075$. For high $Re$ datasets, the $U_\infty$ is calculated based on the $Re$, which varies between $10^5$ and $5 \times 10^6$ with $\nu = 1.461 \times 10^{-5}$ and the chord length of the airfoil as the characteristic length. Note that, the characteristic length in **Race Car** dataset is the effective chord length, $c_{eff}$, between the leading edge of the main airfoil and the trailing edge of the secondary airfoil. *omegaWallFunction* is used as the wall functions of the $\omega$ field, estimating $y^+$ by blending between the viscous and inertial sublayer estimations with a binomial function (Menter, 2001). The corresponding wall functions for $k$ and $\nu_t$ are *kLowReWallFunction* and *nutLowReWallFunction*, respectively.

*simpleFoam*, an incompressible steady-state turbulence flow solver, is used to generate the single airfoil datasets, whereas all the tandem airfoil datasets, which involve overset meshes, are simulated by using *overSimpleFoam*, the *simpleFoam* counterpart. Both solvers discretize the steady-state incompressible NS equations coupled with $k - \omega$ SST turbulence model (Menter et al., 2003), using SIMPLE algorithm (Caretto et al., 2007). The temporal and spatial discretization schemes used in each solver are summarized in Tab. 10 together with the convergence criteria. The resulting linear equations are solved using preconditioned (bi-)conjugate gradient (*PCG*) and stabilized preconditioned (bi-)conjugate gradient (*PBiCGStab*) for symmetric and asymmetric matrices, respectively. The selected preconditioners are diagonal incomplete-Cholesky, (*DIC*), and diagonal incomplete LU, (*DILU*) for symmetric and asymmetric, respectively. The convergence criteria of each field vary in between the order of $10^{-6}$ to $10^{-5}$.

Table 10: The discretization schemes chosen for dataset generation

| Terms | Discretization Scheme |
|---|---|
| Time ($\partial/\partial t$) | Steady state |
| Gradient ($\nabla$) | $2^{nd}$ order Gauss linear |
| Divergence ($\nabla\cdot$) | $2^{nd}$ order Upwind biased |
| Laplacian ($\nabla^2$) | $2^{nd}$ order Gauss linear |
| Interpolation | $2^{nd}$ order linear interpolation |

### A.4 DATASET VALIDATION AND VERIFICATION

#### A.4.1 DATASETS WITH $Re = 500$ FLOW CONDITION

To validate the low $Re$ datasets, the grid sensitivity study was conducted on three different levels of grid resolutions as tabulated in Tab. 11 with their corresponding lift, $c_l$, and drag, $c_d$, coefficients. Note that the thickness of the first cell adjacent to the airfoil wall is maintained at $y^+ < 1$ for all three grid resolutions. The $c_l$ and $c_d$ differences between the fine and dense cases are less than $0.5\%$, suggesting that the solutions are converged. Thus, the fine grid resolution is used for the datasets with low $Re$ flow conditions.

For the tandem-airfoil configurations, the parametric study of tandem-airfoil simulations with ground effect in Yin et al. (2021) is replicated for validation purposes. The optimum computational domain size as described in Appx. A.2 is obtained upon verifying the boundaries are far enough so that the near-field flow interaction between two tandem-airfoils is not affected by the numerical boundaries. In this validation, the airfoil geometry is fixed using NACA 0012 airfoil with a $5°$ angle of attack

Table 11: The mesh independent analysis for NACA 0012 at $Re = 500$ and $\alpha = 5°$

| Resolutions | Total number of cells | $C_l$ | $C_d$ |
|---|---|---|---|
| Coarse | 30832 | 0.25881 | 0.18191 |
| Fine | 122952 | 0.26326 | 0.18179 |
| Dense | 512864 | 0.26254 | 0.18175 |

under a low Reynolds number of 500. Figure 12 compares and shows that the simulation results (red lines) of $c_l$, $c_d$, and $c_l/c_d$ are in excellent agreement with the reference solutions (Yin et al., 2021) (black lines) for parameter study of stagger, S, height, H, and gap, G, respectively.

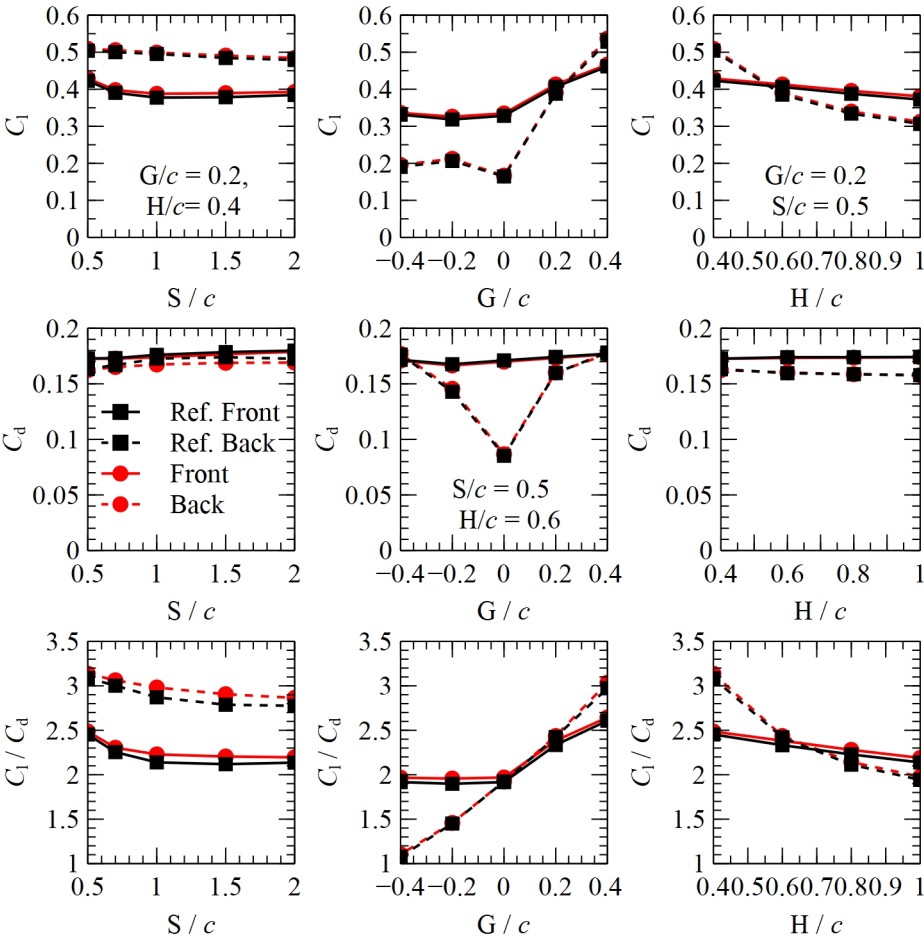

Figure 12: (top row) Lift coefficient, $c_l$, (middle row) drag coefficient, $c_d$, and (bottom row) lift-to-drag ratio, $c_l/c_d$ as a function of (left column) stagger, (middle column) gap, and (right column) height distances.

### A.4.2 DATASETS WITH HIGH $Re$ FLOW CONDITION

The mesh independence study for datasets that consider high $Re$ flow conditions was conducted with NACA 0012 airfoil at $Re = 6 \times 10^6$ and $\alpha = 0°$ flow condition. The domain size of 20 chord lengths is found sufficient as shown in Fig. 13(a). Thereafter, three different grid resolutions were conducted for grid sensitivity analysis, and the $c_d$ results are tabulated in Tab. 12, indicating the fine grid resolution is converged. With the fine grid resolution and same $Re$, a comparison with experimental data (Abbott & Von Doenhoff, 2012; Ladson, 1988; McCroskey, 1987) was then conducted with a range of angles of attack between $-16°$ to $18°$ as shown in Fig. 13(b). The simulation results agree

with the references, especially between $-10°$ to $10°$, thus, the fine grid resolution is adopted for the generation of the datasets, which consider high $Re$. For the **Race Car** dataset, the analyses of computational domain size and accuracy of $c_l$ and $c_d$ are presented in Fig. 14, indicating that the adopted domain size and grid resolution are well aligned with the reference Grabis & Agarwal (2019).

Table 12: The mesh independent analysis for NACA 0012 at $Re = 6 \times 10^6$ and $\alpha = 0°$

| Resolutions | Total number of cells | $C_d$ |
|---|---|---|
| Coarse | 60876 | 0.007575 |
| Fine | 111036 | 0.007594 |
| Dense | 345280 | 0.007565 |

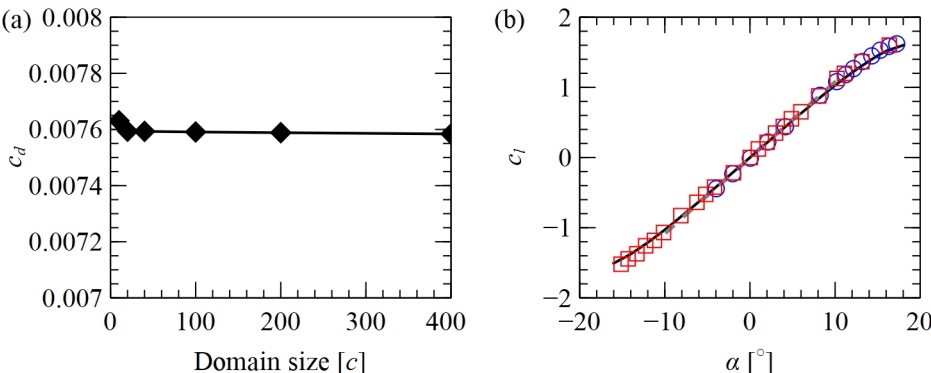

Figure 13: Analyses of (a) domain independence and (b) accuracy for high $Re$ datasets (NACA 0012, $Re = 6 \times 10^6$, $\alpha = 0°$). In (b), the simulated $c_l$ is given by a solid line, while its experimental counterparts are denoted by dotted-line (McCroskey, 1987), circles (Ladson, 1988), and squares (Abbott & Von Doenhoff, 2012).

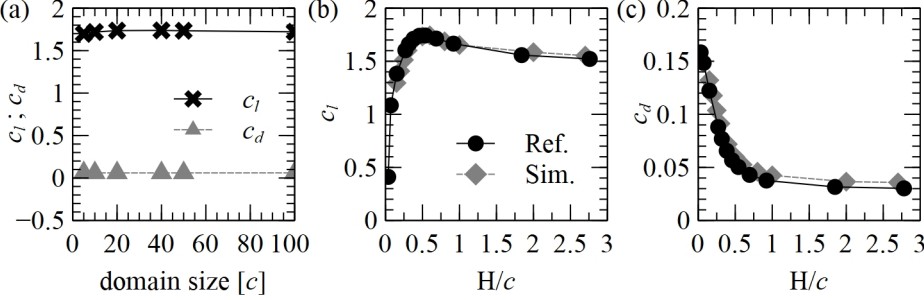

Figure 14: Analyses of (a) domain independence, benchmarking of (b) $c_l$ and (c) $c_d$ of **Race Car** datasets through Grabis & Agarwal (2019).

## B  INPUT AND NEURAL NETWORK PARAMETERS

In all experiments, the angle segments $(\theta_j, \theta'_j) \in \left[\left(-\frac{\pi}{8}, \frac{\pi}{8}\right), \left(\frac{\pi}{8}, \frac{3\pi}{8}\right), \dots, \left(\frac{13\pi}{8}, \frac{15\pi}{8}\right)\right]$ were used to calculate the DID estimates. These were chosen to give 8 arcs each spanning $\frac{\pi}{4}$ degrees, centred at $\frac{\pi}{4}$ intervals. Also, the maximum distance value used was $d_{max} = 5$. The weighing parameter $\alpha = 1$ was used for the training loss.

Table 13 shows the neural network model parameters and training parameters used in the experiments. The input features (and their sizes) to all models consist of the node positions (2), SV (2), DID (8), inlet/overlap values (3) and freestream/combined field (3) when residual training was done, making the input layer size 15 without residual training and 18 with. Likewise, the number of output features

Table 13: Neural network parameters

|  | **MGN** | **IVE** |
|---|---|---|
| NUMBER OF HIDDEN LAYERS | 15 | 8 |
| HIDDEN LAYER SIZE (NODE) | $[128, 128, \ldots, 128]$ | $[128, 256, \ldots, 256, 128]$ |
| HIDDEN LAYER SIZE (EDGE) | $[128, 128, \ldots, 128]$ | $[64, 128, 256, \ldots, 256]$ |
| LOSS FUNCTION | MSE | MSE |
| OPTIMIZER | ADAM | ADAM |
| LEARNING RATE SCHEDULER | LAMBDA DECAY | LAMBDA DECAY |
| LAMBDA FUNCTION | $(1 + k \cdot \lambda_0)^{-1}$ | $(1 + k \cdot \lambda_0)^{-1}$ |
| INITIAL LEARNING RATE ($\lambda_0$) | 5.0E-05 | 2.0E-04 |

Table 14: Number of GPUs used in parallel

| DATASET \MODEL | MGN | IVE |
|---|---|---|
| CRUISE AOA=0° | 4 | 4 |
| CRUISE AOA=5° | 4 | 4 |
| TAKEOFF AOA=5° | 4 | 4 |
| CRUISE RANDOM | 4 | - |
| RACE CAR | 4 | - |

of all models was 3, for the x-velocity, y-velocity and pressure fields. We use the standard mean squared error (MSE) loss function and Adam optimizer to train the neural networks. A custom decay function is used for the learning rate, as defined in Tab. 13. A standard z-score style normalisation is used for all trainings with statistical mean and standard deviation of the training dataset, yielding non-zero values of normalised freestream/combined field. All models were trained using half-precision, NVIDIA GeForce RTX 4080 SUPER and RTX 4080 graphics cards and distributed data parallel with the number of GPUs as specified in Tab. 14.

## C  SIMULATION AND PREDICTION TIMINGS

This section compares the average wall time required to simulate a flow scenario compared to predicting the flow fields using a neural network. Table 15 shows the average wall time per simulation for each dataset used in training and testing. Note that, each simulation runs in parallel with 64 CPUs. Likewise, Tab. 16 shows the average time per double-airfoil case for each step involved in the neural network prediction using the MGN + PRE-RES-FREE + RES-COMB model.

Table 15: Average simulation timings in term of wall time, which uses 64 CPU cores and total CPU times.

| NUMBER OF AIRFOILS | DATASET | WALL TIME (S) | TOTAL CPU TIMES (S) |
|---|---|---|---|
| SINGLE AIRFOIL | SINGLE AOA=0° | $304.42 \pm 32.59$ | $19483 \pm 2086$ |
|  | SINGLE AOA=5° | $152.64 \pm 14.99$ | $9769 \pm 959$ |
|  | SINGLE RANDOM | $249.71 \pm 149.04$ | $15981 \pm 9539$ |
|  | SINGLE INVERTED | $364.73 \pm 206.25$ | $23343 \pm 13200$ |
|  | **AVERAGE** | **267.88** | **17144** |
| DOUBLE AIRFOIL | CRUISE AOA=0° | $252.02 \pm 56.63$ | $16129 \pm 3624$ |
|  | CRUISE AOA=5° | $284.93 \pm 127.49$ | $18236 \pm 8159$ |
|  | TAKEOFF AOA=5° | $292.96 \pm 246.70$ | $18749 \pm 15789$ |
|  | CRUISE RANDOM | $435.47 \pm 132.39$ | $27870 \pm 8473$ |
|  | RACE CAR | $560.56 \pm 545.15$ | $35876 \pm 34890$ |
|  | **AVERAGE** | **365.19** | **23372** |

Note that the single-airfoil cases are meshed using the standard C-grid mesh, while an overset or "chimera" mesh is used in the double-airfoil cases. The background mesh uses a rectangular mesh and the overset mesh uses a handcrafted mesh. The background mesh cells are then connected to

Table 16: Average neural network prediction timings on a single GPU.

| STAGE | OPERATION | SIMULATION TIME (S) |
|---|---|---|
| SINGLE AIRFOIL PREDICTIONS | READ CFD FILE | $12.00 \pm 0.72$ |
| | CALCULATE GEOMETRIC FEATURES ($\times 2$) | $3.34 \pm 0.27$ |
| | INFERENCE ($\times 2$) | $0.30 \pm 0.00$ |
| DOUBLE AIRFOIL PREDICTION | READ CFD FILE + PROCESS OVERSET MESH | $18.97 \pm 0.72$ |
| | CALCULATE GEOMETRIC FEATURES | $17.03 \pm 1.02$ |
| | COMBINE FIELDS | $8.60 \pm 0.68$ |
| | INFERENCE | $1.10 \pm 0.00$ |
| | EXPORT CFD FILE | $4.22 \pm 0.39$ |
| | TOTAL | $\mathbf{65.57 \pm 3.80}$ |

their nearest neighbour cells in the overset mesh to avoid importing two disjointed graphs. Hence, importing a double-airfoil mesh takes longer than a single-airfoil mesh.

From Tabs. 15 and 16, we can see that the average total prediction time comes up to only 65.57 seconds. This is a 99.7% reduction from the average CPU times of 23372 seconds it takes to simulate a double-airfoil case using OpenFOAM.

## D   TRAINING SCHEMES AND TIMINGS

This section presents the average time required by the various NNs in the multi-NN model to train. Note that while the NNs had a maximum epoch of 300, an early-stopping mechanism was utilised, such that training would cease if the validation loss did not improve after 20 epochs, indicating convergence. Additionally, the MGN front models would have a minimum epoch of 200 to ensure sufficient training, due to having more turbulent validation losses.

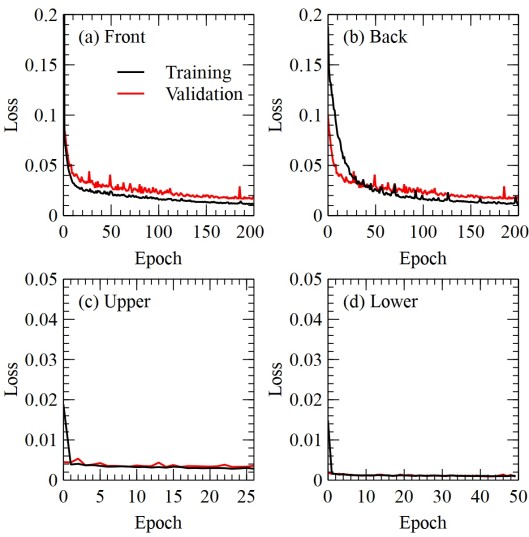

Figure 15: Training and validation loss curves of the various sub-domains.

The training and validation loss curves as shown in Fig. 15 for each sub-domain: (a) front, (b) back, (c) upper, and (d) lower, exhibit consistent convergence. For the front and back sub-domains, the validation losses closely follow training losses across 200 epochs, indicating good generalization to unseen data. Similarly, for the upper and lower sub-domains, both losses converge rapidly and remain stable, suggesting no signs of overfitting. It is worth noting that the size of the computational domain (graph) is large. Prior to domain decomposition, the model may have been underfitting, and this structural decomposition allows the networks to effectively capture sub-domain-specific flow

features without increasing the risk of overfitting. These results collectively demonstrate that our approach maintains a balance between model complexity and generalization.

The average training times of the MGN models using the **Takeoff** AOA=5° dataset are shown in Tab. 17. The single-airfoil models had larger training set sizes, leading to longer training times. Likewise, due to the simplicity of the upper and lower fields, the upper and lower NNs take the least training time to converge. Hence, these categories are separated.

Table 17: Average neural network training timings

| NN | MODEL | AVERAGE TRAINING TIME (S) | |
|---|---|---|---|
| SINGLE-AIRFOIL | MGN (BASELINE) | 27,067 | |
| | MGN + FREE | 28,020 | 21,883 |
| | MGN + RES- FREE | 10,562 | |
| FRONT AND BACK | MGN (BASELINE) | 10,452 | |
| | MGN + PRE-FREE | 15,739 | |
| | MGN + PRE-FREE + COMB | 14,056 | 14,517 |
| | MGN + RES-FREE + RES-COMB | 11,878 | |
| | MGN + PRE-RES-FREE + RES-COMB | 20,459 | |
| UPPER AND LOWER | MGN (BASELINE) | 2,853 | |
| | MGN + PRE-FREE | 2,644 | |
| | MGN + PRE-FREE + COMB | 1,905 | 2,191 |
| | MGN + RES-FREE + RES-COMB | 1,770 | |
| | MGN + PRE-RES-FREE + RES-COMB | 1,782 | |

To facilitate fair comparison of cost–accuracy trade-offs, we report the average training wall-time and peak GPU memory usage for the different training variants used in this work. All measurements are obtained on the same GPU and **Takeoff** dataset for the MGN model. The four regimes in Tab. 18 correspond to: (a) the baseline direct training model, (b) the PRE-FREE (pretrain-only) modle, (c) the RES-FREE + RES-COMB (residual-only) model, and (d) the full PRE-RES-FREE + RES-COMB curriculum model.

Table 18: Average neural network training wall-time and peak GPU memory usage for different training variants.

| Regime | Total Training Time (s) | Max GPU Memory (GB) |
|---|---|---|
| (a) | 10,452 | 72.88 |
| (b) | 14,517 | 72.88 |
| (c) | 11,878 | 76.73 |
| (d) | 20,459 | 76.75 |

As seen in Tab. 18, the residual-only configuration (c) adds only a modest overhead over the baseline (a), while the full curriculum (d) roughly doubles the training time but operates within a similar peak memory envelope.

## E  A DISCUSSION ON THE DID

In this section, we will discuss the challenges of calculating the DID as done in the original work and present the justification for using the smooth-combine method to estimate it instead.

As mentions previously, the DID was estimated numerically following the procedure outlined in Algorithm 1. Although extending the theoretical definition of DID to multiple geometries is conceptually straightforward, the numerical calculations grow significantly more complex with each additional object. These challenges are indicated in red within Alg. 1, and are illustrated in Figs. 16 and 17.

The first challenge is in determining whether the point on the object boundary $k$ is obstructed from the point of reference $i$. As shown in Fig. 16(a), in a single object scenario, it suffices to ascertain

**Algorithm 1** DID calculation. Steps that gain complexity with additional objects are shown in red.

---

**Input:** nodes $V$; positions $[(x_i, y_i) : i \in V]$; boundary indices $bd = [k \in V : k$ is on the boundary of a geometry$]$; angle segments $\left[\left(\theta_j, \theta'_j\right) : 0 \le j < J\right]$; maximum $d_{max}$

$\text{DID} \leftarrow [\,]$
**for** $j \in [0, \dots, J-1]$ **do**
    $\text{DID}_j \leftarrow [\,]$
    **for** $i \in V$ **do**
        $d \leftarrow [\,$ distance between $i$ and $k, \forall k \in [k \in bd : (\theta_j < \theta_{i,k} < \theta'_j)$ and $(k$ is unobstructed from $i)] \,]$
        $d \leftarrow \text{minimum}(d, d_{max})$
        $\text{DID}_\theta \leftarrow$ average values of $d$
        $w_\theta \leftarrow$ proportion of $\left(\theta_j, \theta'_j\right)$ where $(k$ is unobstructed from $i), \forall k \in [k \in bd : (\theta_j < \theta_{i,k} < \theta'_j)]$
        $\text{DID}_i \leftarrow w_\theta * \text{DID}_\theta + (1 - w_\theta) * d_{max}$
    **end for**
    append $\text{DID}_i$ to $\text{DID}_j$
**end for**
append $\text{DID}_j$ to $\text{DID}$
**Return:** DID

---

that either boundary face adjacent to $k$ is on the side of the object that faces $i$. However, as seen in Fig. 16(b), there is the possibility that $k$ is obstructed from $i$ by the boundary faces of another object. Determining obstruction is a process that increases in complexity with the addition of every object.

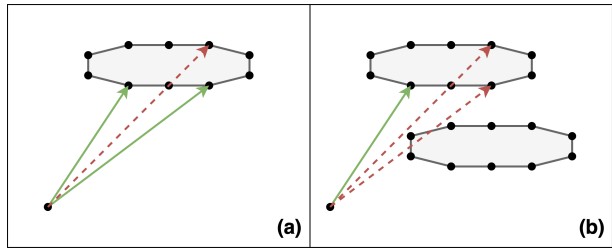

Figure 16: Determining obstruction of a boundary point from the reference point in a **(a)** single-object case and **(b)** double-object case. Note how a boundary point that is unobstructed in the first case may be obstructed by another object in the second case.

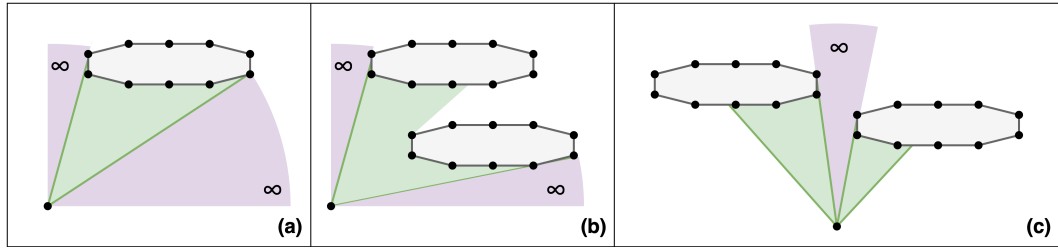

Figure 17: Determining the angular range that faces an object boundary (shown in green) in a **(a)** single-object case, **(b)** overlapping double-object case, and **(c)** non-overlapping double-object case. Note that the angle segment $(\theta_j, \theta'_j)$ used was $(0, \pi/2)$ in (a) and (b) and $(\pi/4, 3\pi/4)$ in (c).

Likewise, the second challenge is in determining the proportion of the angular range $(\theta_j, \theta'_j)$ where $i$ is obstructed by an object. As shown in Fig. 17(a), in a singular object scenario, this proportion can be represented as one continuous segment using the minimum and maximum value of $\theta_{i,k}$, the angle at which $k$ is with respect to $i$. However, in a double object scenario, this proportion may be represented as one continuous segment as seen in Fig. 17(b), or two separate segments as seen in Fig. 17(c). Multiple objects involve pair-wise comparisons of each object in determining whether they overlap (as in the former case) or not (as in the latter case), greatly increasing complexity.

To circumvent these challenges, a smooth-combining method using the deviation from the maximum value $d_{max}$ was utilised to estimate the DID fields for multiple objects in this paper, detailed in Alg. 2.

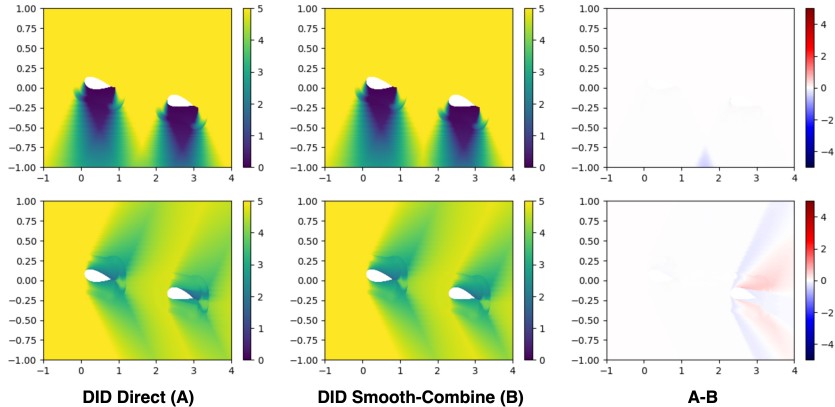

DID Direct (A)          DID Smooth-Combine (B)          A-B

Figure 18: Comparison of the DID field calculated directly with that from smooth-combined estimate. **Above:** Angular range is $[0, \pi]$. **Below:** Angular range is $[\pi/2, 3\pi/2]$.

---

**Algorithm 2** DID estimation for $L$ number of objects.

---

**Input:** nodes $V$; positions $[(x_i, y_i) : i \in V]$; boundary indices $bd_l = [k \in V : k$ is on the boundary of geometry $l] \quad \forall$ geometries $l \in \{1, \ldots, L\}$; angles segments $\left[\left(\theta_j, \theta'_j\right) : 0 \leq j < J\right]$; maximum value $d_{max}$

$\boldsymbol{y}_0 \leftarrow d_{max}$
**for** $l = 1$ **to** $L$ **do**
    $\boldsymbol{y}_l \leftarrow \text{DID}_l$ value calculated using Alg. 1 and boundary indices $bd_l$.
**end for**
$\text{DID}_{est} \leftarrow$ combined field $\widetilde{\boldsymbol{y}}$ calculated using Eqn. 1 and Eqn. 2, with $(\boldsymbol{y}_0, \ldots, \boldsymbol{y}_L)$
**Return:** $\text{DID}_{est}$

---

While there is a difference between the resulting smooth-combined fields from the direct DID calculation, it is important to highlight that these differences are minimal. To illustrate this, the DID fields for two angle segments from a direct calculation and smooth-combining, as well as their difference, are shown in Fig. 18.

As can be seen from the figures, the most significant difference occurs in points $i$ when both objects are within its angular range. In these areas, the smooth-combination will overestimate the DID when the objects do not overlap (blue regions), and underestimate the DID when the objects overlap (red regions). On the other hand, when only one object is in the angular range of $i$, both the direct numerical calculation and the smooth-combined calculation calculates its average distance to every unobstructed boundary point $j$ on the objects using a relatively uniform weight. Hence, there is little difference between the two. Importantly, the smooth-combined calculation produces a close estimate without harsh lines.

The timings and maximum memory usage of the DID calculation done directly was estimated using a small sample of the datasets. It is compared against the estimation using the smooth-combine scheme in Tab. 19. Note that the direct calculation was not optimised, and some steps that could be done in parallel were instead done in succession. Doing them in parallel would decrease the timing but increase the memory overhead.

Table 19: Comparison of DID calculation times and memory overhead

| CALCULATION TYPE | AVG. TIME PER FIELD (S) | MAX. MEMORY USAGE (GB) |
|---|---|---|
| DIRECT | 5221.641 | 25.5 |
| WITH SMOOTH-COMBINE | 3.492 | 23.3 |

As can be seen, using the smooth-combined estimate saves a significant amount of calculation time for a similar amount of memory required, making it the ideal choice. For a dataset of size $784$, the direct DID calculations would take an estimated $47$ days. The accuracy performance of the direct DID is hence irrelevant when the goal is to produce faster results than numerical simulations.

# F  SENSITIVITY STUDIES ON THE DID

To ensure a fair comparison with the baseline methods, hyperparameters such as learning rate, network depth, and layer sizes were kept consistent with the baseline settings. This minimizes variability and ensures that the observed improvements are due to our proposed approach rather than hyperparameter tuning. However, sensitivity to certain domain-specific parameters, such as the maximum DID distance, $d_{max}$ and the number of angle segments used in DID computation, could impact performance. These parameters influence the granularity of the directional distance representation and its ability to capture relevant physical interactions. Sensitivity studies were conducted using the model MGN + PRE-RES-FREE + RES-COMB on the Cruise AOA= 5° dataset. The results, as shown in Tab. 20, reveal the impact of varying $d_{max}$ and the number of angle segments on MSE.

Table 20: MSE performance evaluation for sensitivity studies on DID parameters.

| $d_{max}$ | NO. OF ANGLE SEGMENTS | MSE($\times 10^{-2}$) |
|---|---|---|
| 5 | 8 | 0.68±0.50 |
| 2.5 | 8 | 0.85±0.38 |
| 5 | 4 | 0.89±0.44 |
| 5 | 16 | 0.51±0.29 |

Reducing $d_{max}$ to 2.5 increases MSE, likely because a smaller $d_{max}$ limits the model's ability to capture longer-range interactions. Similarly, decreasing the number of angle segments to 4 also leads to higher errors, suggesting that fewer angle segments reduce the directional resolution of the DID representation. In contrast, increasing the number of angle segments to 16 improves the performance at the expense of higher computation time for the DID features. Compared with the baseline MGN, whose MSE is $1.34 \times 10^{-2}$ on the same Cruise AOA= 5° dataset, the variations of these results are relatively minor, indicating the proposed method's robustness and insensitivity to these parameters.

# G  FEASIBILITY OF INDIVIDUAL DID OF EACH OBJECT

Computing a single DID for both objects simultaneously is primarily a practical decision aimed at improving efficiency and scalability. While Alg. 1 can technically compute a single DID for multiple objects simultaneously, its numerical complexity increases significantly with each additional object, resulting in slower training and inference speeds. For instance, Alg. 1 required 5222 seconds to compute the DID for tandem airfoils, whereas Alg. 2, which computes separate DIDs for individual airfoils and then combines them, completed the task in only 3.5 seconds.

Additionally, calculating separate DIDs for each object would increase the input size proportionally to the number of objects. If the dimension of a single object's DID is $N$ and there are $M$ objects, the total input size would scale as $N \times M$, leading to higher memory requirements and computational load on GPUs. By combining the DIDs into a single representation, our approach maintains scalability and significantly reduces computational overhead. Algorithm 2 strikes an effective balance, allowing for efficient handling of multi-object scenarios without sacrificing performance, as discussed in Appx. E.

To further evaluate the feasibility of using individual DIDs for each object, an experiment was conducted to compare the performance and resource usage of individual DIDs versus a single combined DID using the model MGN + PRE-RES-FREE + RES-COMB on the Cruise AOA= 5° dataset. The results, as tabulated in Tab. 21, reveal that the single combined DID achieves better computational efficiency and prediction accuracy.

Table 21: Performance evaluation of experiment using single combined and individual DID on Cruise AOA= $5°$ dataset.

| METHOD | AVERAGE GPU MEMORY USAGE (GB) | MSE ($\times 10^{-2}$) |
|---|---|---|
| SINGLE COMBINED DID | 16.64 | **0.68±0.50** |
| INDIVIDUAL DID | 23.37 | 0.80±0.42 |

## H SMOOTH-COMBINING METHOD VALIDATION

To validate the effectiveness of the smooth-combining method, we compared its performance against freestream and a simple linear interpolation weighted by the distance to each airfoil as defined,

$$\widetilde{\boldsymbol{U}}(i) = \boldsymbol{\gamma}(i) \cdot \boldsymbol{U}_1(i) + \big(1 - \boldsymbol{\gamma}(i)\big) \cdot \boldsymbol{U}_2(i) \,,$$
$$\boldsymbol{\gamma}(i) = \frac{d_2(i)}{d_1(i) + d_2(i)} \,, \tag{6}$$

where $d_1$ and $d_2$ are the shortest distances to front (leading) and back (trailing) airfoils, respectively. Figure 19 illustrates the weighting field, $\boldsymbol{\gamma_1}$, generated from distance-based linear interpolation, showing a smooth gradient between the two airfoils. The comparison between (a) freestream, (b) distance-based linear interpolation, and (c) smooth-combining methods is presented in Fig. 20, which shows the absolute error contours of the combined velocity components and pressure fields relative to their corresponding ground truths. The smooth-combining method demonstrates the lowest errors, particularly in the downstream and flow interaction regions, where the freestream and linear interpolation methods show pronounced inaccuracies.

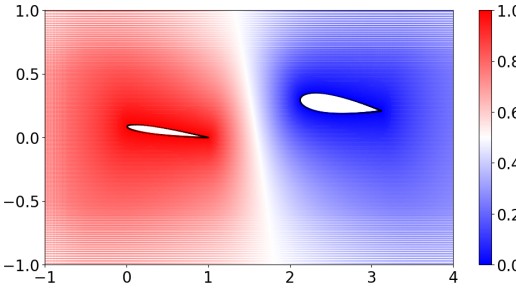

Figure 19: Distance-based linear interpolation weight values for the front airfoil, $\boldsymbol{\gamma_1}$.

This qualitative observation is supported by the quantitative results in Tab. 22, where the smooth-combining method achieves the lowest mean absolute error (MAE) with respect to ground truths across all evaluated metrics, including velocity components and pressure. Specifically, the smooth-combining method outperforms both freestream and linear interpolation with an overall MAE $(\times 10^{-3})$ of $1.46 \pm 0.31$, compared to $1.76 \pm 0.23$ and $6.45 \pm 0.53$ for linear interpolation and freestream, respectively.

Table 22: MAE of combined flow fields via various methods against ground truths.

| METHOD / VARIABLE | $u$ ($\times 10^{-2}$) | $v$ ($\times 10^{-3}$) | $p$ ($\times 10^{-4}$) | OVERALL ($\times 10^{-3}$) |
|---|---|---|---|---|
| FREESTREAM | $1.57 \pm 0.13$ | $3.08 \pm 0.28$ | $5.45 \pm 0.58$ | $6.45 \pm 0.53$ |
| LINEAR INTERPOLATION | $0.39 \pm 0.05$ | $1.14 \pm 0.15$ | $1.95 \pm 0.25$ | $1.76 \pm 0.23$ |
| SMOOTH-COMBINING | $\mathbf{0.31 \pm 0.07}$ | $\mathbf{1.07 \pm 0.16}$ | $\mathbf{1.71 \pm 0.31}$ | $\mathbf{1.46 \pm 0.31}$ |

To further assess the utility of smooth-combining, we conducted an additional experiment using the linear-interpolated flow fields as initial estimators for training the model (MGN + PRE-RES-FREE + RES-COMB) on the Cruise AOA=$5°$ dataset. As shown in Tab. 23, the smooth-combining approach results in significantly lower MSE than linear interpolation. These results confirm that smooth-combining not only provides a more accurate starting point for further training, but also captures

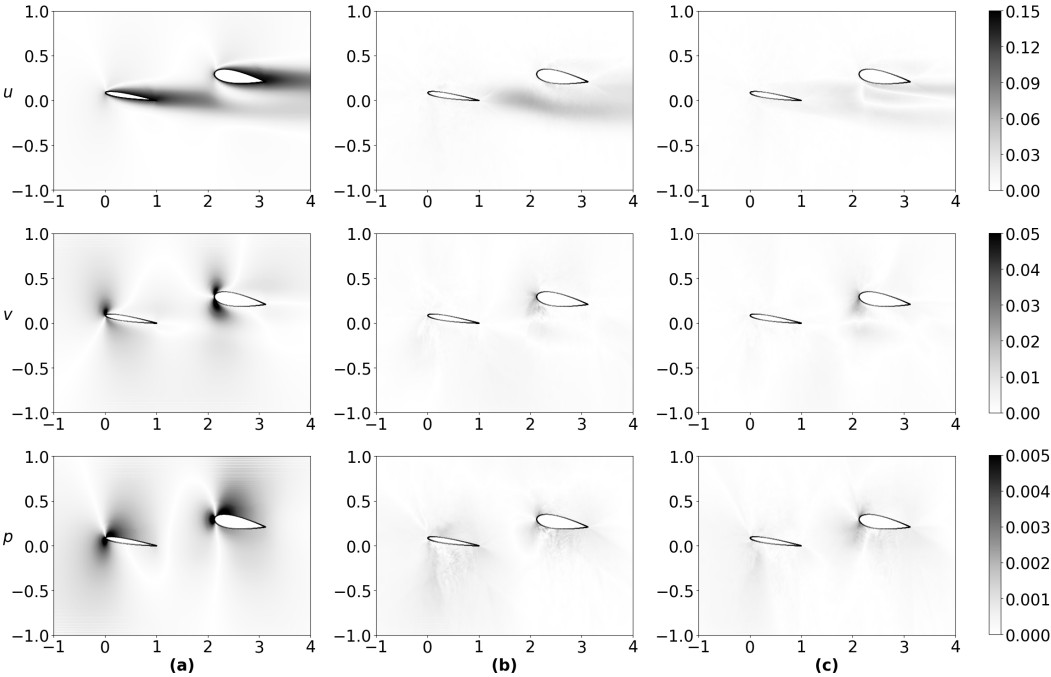

Figure 20: Absolute error contours of combined flow field variables $u$ (top row), $v$ (middle row), and $p$ (bottom row) via (a) freestream, (b) distance-based linear interpolation, and (c) smooth-combining with respect to ground truth $u$, $v$, and $p$.

complex flow interactions more effectively than alternative approaches. Its superior performance in both initial approximation and subsequent training highlights its importance for this framework.

Table 23: Evaluation of model performance using linear interpolated vs smooth-combined flow fields

| METHOD | MSE ($\times 10^{-2}$) |
|---|---|
| LINEAR INTERPOLATION | $1.15 \pm 0.58$ |
| SMOOTH-COMBINING | $\mathbf{0.68 \pm 0.50}$ |

## I ANALYSIS OF PREDICTED FLOW FIELDS

To showcase the effects of varying distance between the two airfoils and increasing AOA on the accuracy of our NN model, we have crafted Figs. 21 and 22. The initial qualitatively compares the prediction of x-velocity, $\hat{u}$, to the ground truth, $u_{GT}$, for (a) two closely-separated airfoils (S = 0.92, G = -0.2) with strong influence by the front airfoil on the aft airfoil and (b) two distant airfoils (S = 1.8, G = 0.38) that are just mildly interacting with each other. The latter shows contours of x-velocity for approximately (a) positive and (b) negative AOA extremes considered in this work (i.e., $[-5°, 6°]$). All the cases illustrate that, visually, there are little differences between the ground truths and corresponding predictions, thus verifying the robustness of our NN model for a decent range of separation distance between the two airfoils and AOA.

As the loss function used in training the MGN model includes a boundary loss component, which directly penalizes errors on the airfoil surface, ensuring that the model learns to capture surface flow characteristics accurately. Figure 23 illustrate the correlation between the ground truth and MGN-predicted lift, $c_l$, and drag, $c_d$, coefficients for three datasets: (a) **Cruise AOA=5°**, (b) **Takeoff**, and (c) **Cruise Random**. For $c_l$, the data points generally follow the diagonal line, indicating a strong correlation between the MGN predictions and the ground truth values for both front and back airfoils. However, a slight deviation is observed in the **Cruise Random** dataset, suggesting greater prediction variability under complex flow conditions. For $c_d$, the predicted values also show a

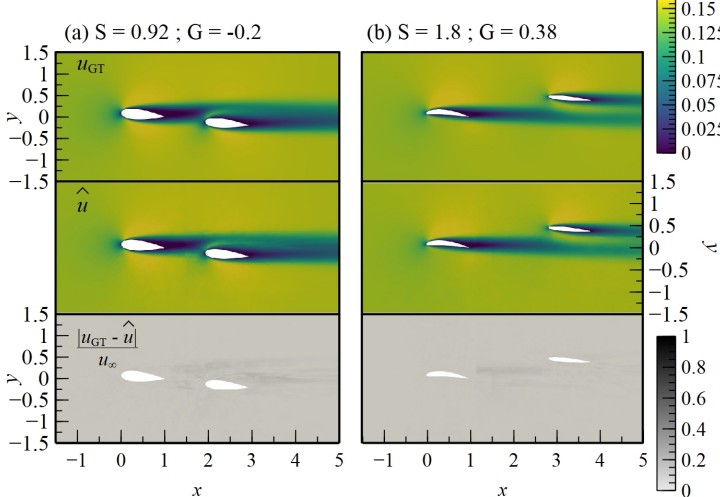

Figure 21: Comparison of **(top)** ground truth $x$-velocity, $u_{GT}$, **(middle)** predicted $\hat{u}$ flow fields by model MGN + PRE-RES-FREE + RES-COMB, and **(bottom)** normalised $x$-velocity error flow fields at **(a)** closely-separated and **(b)** distant airfoils for **Cruise AOA**$= 5°$ dataset.

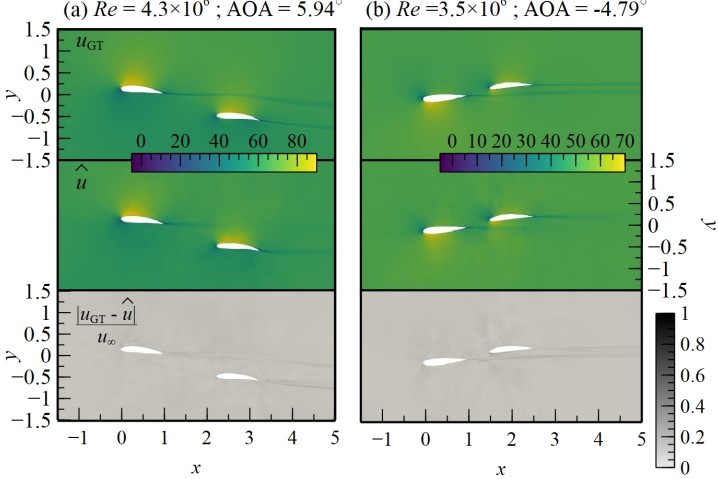

Figure 22: Comparison of **(top)** ground truth $x$-velocity, $u_{GT}$ and **(middle)** predicted $\hat{u}$ flow fields by model MGN + PRE-RES-FREE + RES-COMB, and **(bottom)** normalised $x$-velocity error flow fields at **(a)** positive and **(b)** negative AOA for **Cruise Random** datasets.

reasonable correlation with the ground truth, though a noticeable spread exists, especially for the back airfoil (blue circles). This spread is more significant in the **Takeoff** and **Cruise Random** datasets, indicating that the MGN model has difficulties in capturing velocity-related drag characteristics in these scenarios. These results demonstrate that the MGN model effectively predicts lift coefficients, dominated by pressure, for both front and back airfoils, but its performance on drag prediction is more sensitive to variations in flow conditions and the relative position of the airfoil.

To quantify the accuracy of our NN model across different scenarios, we have tabulated the MSE values of our predictions relative to their ground truths under varying Reynolds number, $Re$, AOA, S, and G in Tab. 24. Like the qualitative assessment in Figs. 21 and 22, Tab. 24 confirms once again the consistent robustness of our NN model, with the MSE remaining within a remarkable range of 0.10 (an order smaller than the baseline MSE of 1.79 for the uniform training condition in Tab. 6) regardless of $Re$, AOA, S, and G.

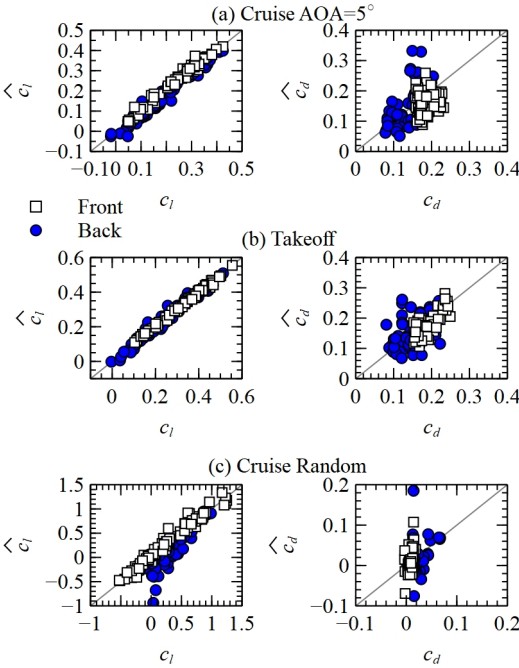

Figure 23: Correlation of ground truth and MGN-predicted (left) lift and (right) drag coefficients for (a) **Cruise AOA=5°**, (b) **Takeoff**, and (c) **Cruise Random** datasets. White squares and blue circles represent the front and back airfoils, respectively, in the tandem-airfoil configuration.

Additionally, we evaluated the normalized residual values of the discrete incompressible Navier–Stokes equations of the SIMPLE algorithm with the predicted x- and y-velocity, $\hat{u}$ and $\hat{v}$, respectively, in Tab. 25. Both variables were predicted with residual that is at least two orders smaller than the maximum value of 1, thus reinforcing the accuracy of our NN as the residual for Navier–Stokes equations is a direct indicator of error relative the exact solution to the simulation (Versteeg & Malalasekera, 2007).

Table 24: Normalized MSE of NN predictions under varying $Re$, AOA, S, and G.

| VARIABLE | RANGE | MSE |
|---|---|---|
| $Re$ | $Re < 10^6$ | $0.007 \pm 0.004$ |
| | $10^6 \leq Re < 3 \times 10^6$ | $0.04 \pm 0.03$ |
| | $Re \geq 3 \times 10^6$ | $0.21 \pm 0.14$ |
| AOA | AOA $< -2°$ | $0.07 \pm 0.08$ |
| | $-2° \leq$ AOA $< 2°$ | $0.08 \pm 0.09$ |
| | AOA $\geq 2°$ | $0.15 \pm 0.18$ |
| S | $S < 1.0$ | $0.11 \pm 0.13$ |
| | $1.0 \leq S < 1.5$ | $0.10 \pm 0.17$ |
| | $S \geq 1.5$ | $0.08 \pm 0.08$ |
| G | $G < -0.4$ | $0.09 \pm 0.10$ |
| | $-0.4 \leq G < 0.4$ | $0.10 \pm 0.15$ |
| | $G \geq 0.4$ | $0.10 \pm 0.10$ |

Table 25: Mean and standard deviation of normalised residuals of discrete incompressible Navier–Stokes equations of SIMPLE algorithm over 10% of the random cruise tandem airfoils datasets.

| VARIABLE | NORMALISED RESIDUAL (MAXIMUM OF 1) |
|:---:|:---:|
| $u$ | $0.00203 \pm 0.00017$ |
| $v$ | $0.0168 \pm 0.00074$ |

## J    ADDITIONAL RESULTS WITH A TRANSFORMER-BASED ARCHITECTURE

To further assess whether the proposed PRE-RES-FREE + RES-COMB model is tied to a particular architecture, we evaluate a Transformer-based surrogate model, denoted Transolver (Luo et al., 2025), on the **Cruise AOA=5°** and **Cruise Random** datasets. We keep the same mesh-based inputs and training scheme as for the GCN-based models in the main text, and simply replace the network architecture with a Transformer encoder.

Table 26 reports the MSE ($\times 10^{-2}$) and the relative improvement by comparing Transolver with and without our curriculum learning pipeline, alongside the corresponding GCN baselines (MGN and IVE).

Table 26: MSE ($\times 10^{-2}$) performance comparison between the Transformer-based model (Transolver (Luo et al., 2025)) and GCN-based models on the **Cruise AOA=5°** and **Cruise Random** datasets.

| MODEL / DATASET | CRUISE AOA = 5° | | CRUISE RANDOM | |
|:---|:---:|:---:|:---:|:---:|
| | MSE | IMPR. % | MSE | IMPR. % |
| TRANSOLVER | $0.103 \pm 0.076$ | – | $18.68 \pm 17.48$ | – |
| TRANSOLVER + PRE-RES-FREE + RES-COMB | $\mathbf{0.060 \pm 0.082}$ | **41.7** | $\mathbf{6.25 \pm 8.88}$ | **66.5** |
| MGN (BASELINE) | $1.34 \pm 0.67$ | – | $179 \pm 138$ | – |
| MGN + PRE-RES-FREE + RES-COMB | $\mathbf{0.67 \pm 0.50}$ | **50.0** | $\mathbf{10 \pm 13}$ | **94.4** |
| IVE (BASELINE) | $1.05 \pm 0.69$ | – | $177 \pm 137$ | – |
| IVE + PRE-RES-FREE + RES-COMB | $\mathbf{0.63 \pm 0.30}$ | **40.0** | $\mathbf{7.91 \pm 9.30}$ | **95.5** |

Across all three architectures, applying the PRE-RES-FREE + RES-COMB model leads to substantial error reductions, particularly on the more challenging **Cruise Random** dataset where the MSE often drops by more than an order of magnitude. This supports our claim that the proposed curriculum learning scheme is a general, architecture-agnostic component of our contribution rather than a model-specific trick.

Across all three backbones, applying the PRE-RES-FREE + RES-COMB pipeline reduces the error substantially, especially on the more challenging Cruise Random subset. This supports our claim that the proposed transfer-learning scheme is a general, architecture-agnostic component of our contribution, rather than being tailored to a specific GCN model.

## K    EXTENSION TO THREE-AIRFOIL TANDEM CONFIGURATIONS

To assess whether the proposed framework generalizes beyond the two-body tandem setting, we construct an additional dataset consisting of three-airfoil tandem configurations (i.e., three airfoils placed in series along the streamwise direction). All cases share the same inflow conditions as the **Cruise AOA=5°** dataset used in the main experiments, so that any performance differences can be attributed to the increased geometric complexity rather than changed flow parameters.

**Dataset setup.**    We vary both the streamwise stagger and vertical alignment between neighboring airfoils. Let the streamwise distance between the leading and middle airfoils be $S_{12}$, and between the middle and trailing airfoils be $S_{23}$. Both are sampled in the range

$$S_{12}, S_{23} \in [0.5, 1.0],$$

while the vertical gaps between consecutive airfoils are sampled as

$$G_{12}, G_{23} \in [-0.4, 0.4],$$

measured in chord length units relative to the leading airfoil. In total, this yields 125 three-airfoil configurations. For each configuration, we run high-fidelity unsteady CFD simulations using the same numerical setup and temporal sampling as in the **Cruise AOA=5°** tandem dataset described in the main text.

**Models and training protocol.**    We evaluate (i) the baseline MeshGraphNet (MGN) and (ii) our best-performing variant from the main paper, MGN + PRE-RES-FREE + RES-COMB. Both models use the same architecture, DID geometry encoding, and training hyperparameters as in the two-body tandem case. We do not introduce any additional tuning specific to the three-body dataset. The only change is that the DID preprocessing now accounts for three airfoils instead of two, which is a direct extension of the original formulation.

**Quantitative results.**    Table 27 reports the MSE on the three-airfoil dataset, using the same evaluation metrics as in the main text. The overall MSE over the full flow field, and the MSE evaluated on mesh nodes lying on the airfoil boundaries, which is more directly related to lift and drag forces prediction.

Table 27: MSE performance on the newly generated three-airfoil tandem dataset (125 configurations). "Overall" reports the MSE over the full flow field, while "Boundary" reports the MSE restricted to nodes on the airfoil surfaces.

| Model | Overall MSE ($\times 10^{-2}$) | Boundary MSE ($\times 10^{-1}$) |
|---|---|---|
| MGN (Baseline) | $6.49 \pm 2.04$ | $1.80 \pm 1.10$ |
| MGN + PRE-RES-FREE + RES-COMB | $2.55 \pm 1.12$ | $0.97 \pm 1.18$ |
| Improvement | **60**.**7**% | **46**.**1**% |

Compared to the baseline MGN, our proposed variant reduces the overall MSE by approximately 60.7% and the boundary MSE by 46.1%. These gains are comparable to (and in some cases larger than) those observed in the two-body tandem experiments in the main paper. Importantly, the improvement is achieved without any additional architectural changes or three-body-specific tuning, suggesting that the combination of DID geometry encoding, freestream/combined-field residual training, and smooth-combining naturally scales to more complex multi-body interactions.

**Qualitative error analysis.**    To visualize how well the model captures the three-airfoil flow, Fig. 24 shows flow fields for a representative configuration from the new dataset. For each variable, $x$-velocity $u$, $y$-velocity $v$, and pressure $p$, we plot (left) the CFD ground-truth field, (middle) the prediction of our best model (MGN + PRE-RES-FREE + RES-COMB), and (right) the corresponding absolute error $|\text{GT} - \text{pred}|$.

Despite the strong wake–wake interactions and the narrow gaps between airfoils, the predicted fields closely follow the ground truth solution. The absolute error plots remain nearly uniform and very close to zero in the far field, and the noticeable discrepancies are confined to thin shear layers near the airfoil surfaces and in the downstream wake regions. The error magnitudes are small relative to the dynamic range of each variable, and the wake structures behind all three airfoils are well reproduced. These qualitative observations are consistent with the quantitative improvements reported in Tab. 27, indicating that the proposed framework can accurately model complex three-body interactions using the same architecture and training pipeline as in the two-body experiments.

**Summary.**    Overall, the three-airfoil experiments support our claim that the proposed framework is not restricted to two-body tandem configurations. The same architecture and training pipeline transfer to more complex multi-body setups and continue to yield substantial accuracy gains over a strong MGN baseline. Extending the same ideas to 3D geometries (e.g., finite-span wings or cascades) is a natural next step, which we leave as future work.

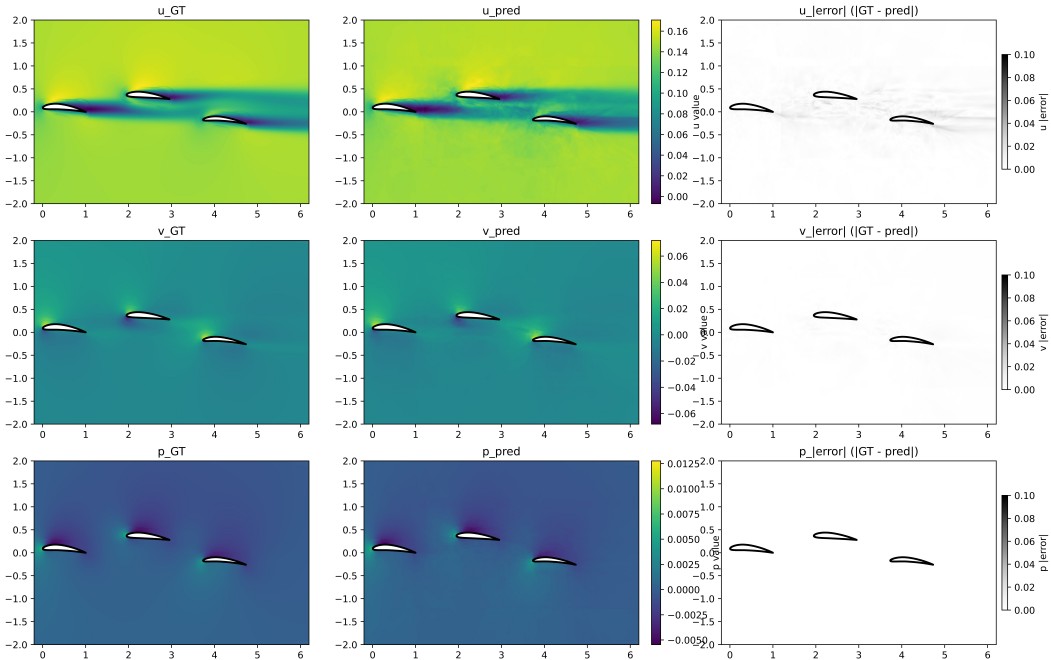

Figure 24: Representative three-airfoil configuration from the new dataset. Rows correspond to $x$-velocity $u$ (top), $y$-velocity $v$ (middle), and pressure $p$ (bottom). For each quantity, the left column shows the CFD ground-truth field, the middle column shows the prediction of MGN + PRE-RES-FREE + RES-COMB, and the right column shows the corresponding absolute error $|GT - pred|$. The errors are small and mainly localised near shear layers and wake regions, indicating that the proposed model accurately captures the three-body wake interactions and near-body flow.

