# OpenReview forum: "TandemFoilSet: Datasets for Flow Field Prediction of Tandem-Airfoil Through the Reuse of Single Airfoils"
_ICLR.cc/2026/Conference — ICLR 2026 Poster_

### Official Review · Reviewer_CkWJ · 2025-10-23

**Soundness:** 4
**Presentation:** 2
**Contribution:** 1
**Rating:** 2
**Confidence:** 4

**Summary:**

The paper presents a dataset of fluid simulations around tandem airfoils, as opposed to single airfoils.  4000 simulations are included in the dataset.  The authors further demonstrate training models on single-airfoil datasets and then fine-tuning them on tandem-airfoil configurations.  This is done in Section 4.4 using multiple NNs as opposed to a single NN.  The datasets all appear to be two-dimensional.  OpenFOAM is used for the CFD simulations.  If I am reading the tables correctly, the errors of their NN predictions appear to range from 0.5-8% for various tests.  A DID discretization is used to represent the solids/flow field.

**Strengths:**

The paper presents some interesting analysis of what is needed to use a discretization like DID when multiple solids are present in a solid-fluid simulation.  This results in some thoughtful usage of multiple NNs and the "smooth combining" procedure in Section 4.1.

**Weaknesses:**

- The paper claims that the proposed dataset is more useful than single-foil datasets due to motivations from real-world engineering, yet, no demonstrations are provided of how this dataset helps solve any real engineering problems better than a single-foil dataset.  The paper is thus unconvincing on this point.
- 4000 is a relatively small dataset size in the era of deep learning.  It is not sufficiently justified why this is chosen.
- The data are 2D-only.  Again, the real world is 3D, so it does not seem very useful to have a dataset of mere 2D configurations.
- I also don't think tandem airfoils occur in isolation in the real world.  For instance, they should be attached to cars or airplanes.  A dataset of real-world objects that happen to have tandem airfoils, would likely be more useful.
- The paper appears to use a very niche discretization, DID, as opposed to the various FDM, FEM, FVM, or other discretizations that have been popular in CFD for many decades.  This may limit the usefulness of the analysis in the paper, although the tandem-foil geometries could still be used by other researchers (though that is a pretty trivial contribution).
- Regarding DID and the benchmarks, the high error rates of 0.5-8% would generally not be considered acceptable by CFD researchers.  Classical CFD solvers can get errors down to e.g. machine precision of like 1e-15.  That being said, other methods like PINNs also produce results with high errors.  But there is no comparison of this method to other AI-based methods to justify its performance - perhaps other PINN or similar NN-based methods would do worse, and it's just the size of the dataset that's the constraint.  There are no such comparisons in the paper, though.
- It is not clear that a method like DID is really practical for CFD when it becomes so much more computationally difficult with each object added to a simulation, when compared to typical one-way solid-fluid coupling algorithms that can handle hundreds of thousands of solids just about as quickly as one.

**Questions:**

None

---

> ### Author Response · Authors · 2025-11-19
> **Response to W1**
>
> We thank the reviewer for the positive feedback on the usage of multi-NN and the smooth-combining procedure in Sec. 4.1. However, we believe that there are **some misunderstandings or misinterpretations in our work**. Thus, we will address these misconceptions point by point in our responses.
>
> **In response to W1 regarding the "claim of more useful than single-airfoil":**
>
> *The paper claims that the proposed dataset is more useful than single-foil datasets due to motivations from real-world engineering, yet, no demonstrations are provided of how this dataset helps solve any real engineering problems better than a single-foil dataset. The paper is thus unconvincing on this point.*
>
> Our intention is **not to state that TandemFoilSet is “more useful” than existing single-airfoil datasets** in general. Instead, we explicitly position it as a complementary resource that **enables the reuse of abundant single-airfoil data** to tackle tandem / multi-body configurations in the original manuscript:
> - In Sec. 2.2, ln. 117-121, we state that existing public datasets “feature single-body flows” and that they “do not facilitate research of reusing single-object data to improve tandem-object flow field prediction.”
> - In Sec. 4, ln. 200-201, we write: “This section introduces the schemes developed to optimise the use of existing single-airfoil datasets for predicting flow around tandem-airfoil configurations—a traditionally resource-intensive task.”
>
> Thus, the central claim is that TandemFoilSet, together with our benchmark schemes, provides the first systematic setting to leverage widely available single-airfoil data to improve predictions for tandem-airfoil flows, not that tandem data is inherently more useful than single-airfoil data.
>
> Regarding “real engineering” motivation, the tandem configuration we model is not artificial, but directly aligned with several established applications that we cite in the introduction, including:
> - Compressor blades in turbomachinery [22], where tandem blade rows are used to control losses and loading distribution.
> - Unmanned aerial vehicles (UAVs) [23,24] employing tandem-wing / tandem-airfoil layouts, particularly in takeoff and low-altitude operations.
> - Race-car rear wings [25], where inverted multi-element airfoils operate near the ground to generate downforce.
> - Hydrofoil systems for maritime vessels [26], where multiple lifting surfaces in close proximity interact strongly.
>
> These uses are reflected in our dataset design. For instance, the Cruise and Takeoff datasets emulate high-lift aircraft/UAV conditions with and without ground effect, while the Race Car dataset models inverted tandem airfoils near the ground, under high Reynolds numbers and realistic angle-of-attack ranges. This is precisely the regime where accurate, fast tandem-airfoil surrogates can shorten design loops for, e.g., spoiler geometry or tandem blade spacing studies.

---

> ### Author Response · Authors · 2025-11-19
> **Response to W2, W3, and W4**
>
> **In response to W2, W3, and W4 regarding the dataset:**
>
> *4000 is a relatively small dataset size in the era of deep learning. It is not sufficiently justified why this is chosen.*
>
> *The data are 2D-only. Again, the real world is 3D, so it does not seem very useful to have a dataset of mere 2D configurations.*
>
> *I also don't think tandem airfoils occur in isolation in the real world. For instance, they should be attached to cars or airplanes. A dataset of real-world objects that happen to have tandem airfoils, would likely be more useful.*
>
> We understand that physics-informed AI is an inherently interdisciplinary area, and that aspects which are standard on the AI side may not be immediately familiar or intuitive from a traditional CFD perspective, and vice versa. Our work follows a typical CFD/physics benchmarks in top AI conferences, where high-fidelity CFD is used to generate ground-truth data, while the learning side introduces additional geometric features (such as DID) and schemes to help neural networks better capture boundary and interaction effects, similar in spirit to other ML-for-CFD works [1-16]. Our work, including 2D datasets and 4000 tandem-airfoil simulations, is well aligned with previous settings in this area.
>
> First, TandemFoilSet contains 8,104 CFD simulations in total, of which 4,152 are tandem-airfoil cases, not merely “around 4000”. We have clarified this total explicitly in the revised manuscript to avoid any ambiguity.
>
> Second, in the context of physics-based AI, datasets with O(10³) simulations are in fact standard, and **TandemFoilSet is larger than many widely used CFD benchmarks**. As summarised in Tab. 1, typical datasets in this area contain a few hundred to a few hundred thousand 2D simulations. All the datasets in Tab. 1 are **merely single-object configuration**. Compared to these, TandemFoilSet (8,104 2D high-fidelity simulations) is **at the upper end of current practice with having more than one body and aiming to study their interactions**. This regime is precisely where physics-based AI is most relevant, as each sample is a computationally expensive, high-fidelity simulation, not a cheap image or text token, so “ImageNet-scale” sample counts are neither realistic nor necessary to demonstrate meaningful generalisation.
>
> Table 1: List of available physics-related datasets.
> | Dataset  | Venue / Year  | Citations | Type| Size  | Types of Physcis  |
> |-|-|-|-|--|-|
> | AirfRANS [1] | NeurIPS 2022  | 103 | 2D | 1,000  | Incompressible Navier–Stokes (airfoil external aerodynamics) |
> | MGN airfoil [2] | ICML 2021 | 1365 | 2D    | 230  | High-Reynolds-number incompressible Navier–Stokes (airfoils)    |
> | CylinderFlow [2]     | ICML 2021| 1365| 2D    | 800  | Incompressible Navier–Stokes (bluff-body / cylinder wake) |
> | FLAGSimple [3] | ICML 2020| 1676| 2D    | 800  | Deformable solid mechanics (flexible flags in flow) |
> | Curated Dataset [4]  | Scientific Data 2021 | 84  | 2D    | 841  | Turbulent incompressible flows in canonical channel/duct setups |
> | Wind Turbine Airfoil Noise DB [5] | Applied Energy 2024 | 16 | 2D |11,700 profiles | Aeroacoustics and aerodynamics of wind-turbine airfoils |
> | MegaFlow2D [6] | CPSWEEK 2023   | 11  | 2D    | 3,000 cases| External & internal incompressible flows; steady + unsteady     |
> | EAGLE [7]| ICLR 2023| 60  | 2D    | 1,200| External aerodynamic flow around complex bodies     |
> | Fluid Flow Dataset [8] | IEEE TVCG 2020     | 52  | 2D    | 8,000 (1001 time steps each) | Unsteady incompressible Navier–Stokes parameterized by Reynolds number |
> | InflatingFont [9]    | ICML 2023| 62  | 2D    | 1,000| Fluid–structure interaction with deformable elastic shells|
> | Cavity Flow [10]     | ACM SIGKDD 2023| 11  | 2D    | 1,000| Incompressible Navier–Stokes (lid-driven cavity)    |
> | Bouncing Ball [11]   | NeurIPS 2021   | 38  | 1D    | 1,100| Rigid-body dynamics with impact and contact   |
> | Simulated Cloth [12] | ICLR 2023| 33  | 2D    | 1,050| Deformable-body dynamics (cloth / thin-sheet mechanics)   |
> | AFBench [13]| NeurIPS 2024 |  6 | 2D / Parametric | 20,000+ airfoil configurations | Aerodynamic simulation across laminar-transitional-turbulent regimes; subsonic & transonic compressible flow |
> | CheMixHub [14] | NeurIPS 2025   | –   | Tabular/graph | 500k points, 11 tasks     | Physical chemistry of mixtures (transport, thermodynamics, electrolyte & olfactory properties) |
> | Bubbleformer / BubbleML 2.0 [15] | NeurIPS 2025 | –   | 2D | 160 sims | Multiphase boiling: two-phase liquid–vapor NS + energy equation |
> | UniFoil [16] | NeurIPS 2025 | – | 2D | 500,000 | Transitional & fully-turbulent compressible/incompressible Navier–Stokes (single airfoil flows) |

---

> ### Author Response · Authors · 2025-11-19
> **Continue For Response to W2, W3, and W4**
>
> Regarding the concerns that “*the real world is 3D*” and 2D datasets are therefore “*not very useful*” and “*don't think tandem airfoils occur in isolation in the real world*”, we would like to address this from the perspectives of **(1) the AI/ML community** and **(2) the CFD/engineering community**.
>
> **(1) From the AI/ML community perspective.**
>
> In the machine-learning-for-physics literature, 2D CFD datasets are standard and highly impactful testbeds for developing methods, architectures, and training schemes that can later be adapted or extended to more complex 3D cases. As summarised in Tab. 1, many influential datasets published at top AI venues are 2D, including AirfRANS at NeurIPS 2022 (2D incompressible RANS over airfoils) [1], MGN airfoil and CylinderFlow at ICML 2021 (2D incompressible RANS over cylinder and airfoil) [2], and EAGLE at ICLR 2023 (2D meshes for turbulent flows around UAV-like bodies) [7]. These 2D benchmarks are routinely used to:
> - (i) perform controlled ablations on architecture and training schemes, and
> - (ii) provide reproducible, moderate-cost testbeds before moving to far more expensive 3D simulations.
>
> TandemFoilSet follows this well-established 2D paradigm but introduces a **new and practically important challenge** which is strongly interacting multi-body flows (tandem airfoils) with varying spacing, angle of attack, Reynolds number, and near-ground effects. These settings already exhibit complex wake–wake interaction and non-local pressure coupling that are highly non-trivial for current GNN-based models, as our benchmarks show. From the AI perspective, TandemFoilSet is therefore a realistic yet tractable dataset for developing general-purpose architectures and training schemes that can later be modified or extended to 3D geometries in future work.
>
> **(2) From the CFD / engineering community perspective.**
>
> Within the CFD community, **2D simulations are widely accepted and routinely used for preliminary design, parametric studies, and code validation, especially for lifting surfaces and cascades**. Examples include 2D CFD-based airfoil shape optimisation and design studies [17], 2D simulations of compressor and turbine cascades for performance assessment and loss reduction [18], and 2D high-lift and multi-element airfoil computations used to validate CFD capability and support airfoil design choices [19]. 2D CFD is also used to compare turbulence models and match experimental data for specific airfoil sections before moving to more complex configurations [20]. More broadly, several aero-structural and aircraft-design frameworks explicitly rely on 2D airfoil performance computed by CFD as inputs to quasi-3D lifting-line or strip models in the early design stages [21].
>
> In aerodynamics, it is **standard practice to study canonical subsystems**, such as single airfoils, tandem airfoils, blade rows, and cylinder wakes, in isolation as proxies for sections of more complex 3D configurations. For example, 2D cylinder and airfoil benchmarks are widely used by both the CFD and AI communities (including major research labs like DeepMind [2,3]) as primary testbeds for understanding unsteady wake dynamics and for developing new numerical and learning methods. Similarly, compressor/turbine cascades [22], tandem-wing UAV sections [23,24], and race-car rear-wing profiles [25] are routinely analysed in 2D or quasi-2D to study lift, drag, and wake–wake interactions before considering a full 3D vehicle.
>
> In this context, our tandem-airfoil dataset is **fully aligned with established practice** in the CFD community by providing canonical 2D sections representative of tandem lifting surfaces which can be used for preliminary design studies, understanding multi-body interaction mechanisms, or building fast surrogates that feed into higher-level design and optimisation loops. The fact that ultimate industrial applications are 3D does not diminish the usefulness of such 2D canonical datasets, rather 2D studies are a standard and necessary step in both method development (for AI) and preliminary design / physics understanding (for CFD).
>
> We would like to emphasise again that
> - **(i) TandemFoilSet’s size (8,104 simulations) is competitive or larger compared to existing physics-ML datasets**, and
> - **(ii) its 2D design is a deliberate choice aligned with the vast majority of current CFD/physic benchmarks in top AI conferences**,
>
> providing a realistic yet tractable testbed for developing methods that can later be extended to 3D configurations.

---

> ### Author Response · Authors · 2025-11-19
> **Response to W5**
>
> **In response to W5 regarding the misinterpretation of DID:**
>
> *The paper appears to use a very niche discretization, DID, as opposed to the various FDM, FEM, FVM, or other discretizations that have been popular in CFD for many decades. This may limit the usefulness of the analysis in the paper, although the tandem-foil geometries could still be used by other researchers (though that is a pretty trivial contribution).*
>
> We respectfully disagree with the reviewer’s characterization of DID as a “very niche discretization.” DID in our work is **not a numerical discretization scheme** in the sense of FDM/FEM/FVM. Instead, it **is a geometric descriptor** used purely as an input feature for the training model. Precisely, DID encodes distance-based information relative to the boundaries (e.g. airfoil surfaces) at each graph node, in a similar spirit to signed-distance function of geometric embedding. It does not alter how the governing equations are discretized or solved. It simply augments the training data with richer geometric context so that the model can better capture boundary-related effects and wake interactions.
>
> As detailed in the original manuscript where DID was introduced (see Sec. 4.2 and Appx. F-G), the dataset itself remains fully usable with or without this feature. Other researchers can ignore DID entirely, define their own geometric encodings, or use different model architectures. Therefore, the **usefulness of our analysis and of TandemFoilSet is not limited by DID**, and we believe the manuscript already makes this distinction between numerical discretization and geometric input features sufficiently clear.

---

> ### Author Response · Authors · 2025-11-19
> **Response to W6**
>
> **In response to W6 regarding the misconception of error reading:**
>
> *Regarding DID and the benchmarks, the high error rates of 0.5-8% would generally not be considered acceptable by CFD researchers. Classical CFD solvers can get errors down to e.g. machine precision of like 1e-15. That being said, other methods like PINNs also produce results with high errors. But there is no comparison of this method to other AI-based methods to justify its performance - perhaps other PINN or similar NN-based methods would do worse, and it's just the size of the dataset that's the constraint. There are no such comparisons in the paper, though.*
>
> We thank the reviewer for this comment, but we have to disagree with the reviewer’s interpretation of both the error levels and the lack of comparisons.
>
> First, we are not sure where the quoted “0.5–8%” error range comes from. Our paper **reports errors in terms of MSE and related field-wise metrics** of a trained model with respect to CFD ground truth.  We do not present any “percentage error” numbers of 0.5–8%. Moreover, these MSEs are properties of the trained model, not of a CFD solver or a numerical discretisation.
>
> Second, the expectation that errors should be reduced “down to machine precision of like 1e-15” is not applicable in this context. A classical CFD solver may indeed converge its iterative residuals for the Navier–Stokes equations to a tolerance near machine precision, but this only guarantees convergence of the numerical scheme. It does not imply that the converged solution is nearly 0% error of the exact physical solution or experimental measurements. In practice, due to modelling assumptions (turbulence models, boundary conditions, geometry simplifications), validation studies routinely report CFD–experiment discrepancies of order 10-15% as acceptable for design and analysis purpose. For example, Calis et al. [27] note that a 10% error in pressure-drop prediction is “acceptable for design purposes”, and Guichet et al. [28] consider predictive errors “below 15%” acceptable in their heat-transfer validation study. **Comparing a data-driven NN’s field error directly to the solver’s internal residual tolerance is therefore an unfair and unsuitable criterion.**
>
> More broadly, the ML-for-CFD community routinely evaluates surrogate models using **MSE and relative errors at levels far above 1e-15**, and these are widely regarded as meaningful advances. For instance, the AirfRANS [1] benchmark reported MSEs on normalized flow fields of order $10^{-2}$ for all their metrics, together with drag/lift relative errors up to ~8% in scarce-data regimes, and still concludes that relative errors below 5% represent “effective and accurate” models. MeshGraphNet [2,3] by Deepmind adapts the same practice as well. If one were to require surrogate errors to match a CFD solver’s $10^{-15}$ residual tolerance, this would, by the same logic, render a large body of influential work in ML-based CFD “unacceptable”, which is clearly not how the field is assessed in practice.
>
> In our method, beyond the datasets' contribution, the goal is to train a fast surrogate model that approximates high-fidelity CFD solutions over a large parametric space, not to match a solver’s internal convergence tolerance. Within this standard surrogate-modelling context, the error levels we report and the improvements we obtain over strong baselines are well aligned with current practice in the ML-based CFD literature.
>
> Third, the statement that there is “no comparison of this method to other AI-based methods” is **factually incorrect**. The paper already includes multiple ablation and comparison experiments (Experiment 2, 3, and 4) between MeshGraphNet (MGN) and Invariant Edge-GCNN (IVE), with and without our proposed training schemes. In the revised manuscript (Appx. J), we **further strengthen this by adding a Transformer-based architecture** (Transolver [29]) to demonstrate that our approach is **robust across different neural architectures**. As summarised in Tab. 26 (of the revised manuscript) below, incorporating our PRE-RES-FREE and RES-COMB schemes consistently improves performance for all three families of models. These results show substantial and consistent error reductions across both GCN- and Transformer-based models, directly addressing the concern about justification relative to other AI-based methods.
>
> Table 26 (of revised manuscript): MSE (×10⁻²) performance comparison between Transformer vs. GCN models.
> | Model Variant  | Cruise AOA=5$^\circ$ | Cruise Random  |
> |- |- |-|
> | **Transolver**  | 0.103 ± 0.076     | 18.68 ± 17.48   |
> | **Transolver + PRE-RES-FREE + RES-COMB** | **0.060 ± 0.082** | **6.25 ± 8.88** |
> | **MGN (Baseline)** | 1.34 ± 0.67  | 179 ± 138  |
> | **MGN + PRE-RES-FREE + RES-COMB**  | 0.67 ± 0.50  | 10 ± 13 |
> | **IVE (Baseline)**| 1.05 ± 0.69       | 177 ± 137       |
> | **IVE + PRE-RES-FREE + RES-COMB** | 0.63 ± 0.30  | 7.91 ± 9.30 |

---

> ### Author Response · Authors · 2025-11-19
> **Response to W7**
>
> **In response to W7 regarding the comparison of DID to solid-fluid coupling algorithms:**
>
> *It is not clear that a method like DID is really practical for CFD when it becomes so much more computationally difficult with each object added to a simulation, when compared to typical one-way solid-fluid coupling algorithms that can handle hundreds of thousands of solids just about as quickly as one.*
>
> We believe there is a misunderstanding about the role of DID in our work.
>
> As clarified in the manuscript, DID is **not a CFD solver, nor any kind of fluid–structure coupling algorithm**. It does not participate in solving the Navier–Stokes equations, and it does not replace or modify standard CFD pipelines. Instead, DID is a geometric descriptor used purely as an input feature for the neural networks. At each graph node, it encodes distance-based information to the boundaries, such as airfoil surfaces, in the same spirit as signed-distance function that provide richer geometric context to training process.
>
> The comparison with “typical one-way solid–fluid coupling algorithms that can handle hundreds of thousands of solids” is therefore **not appropriate**, because DID is not a solver at all. The computational cost of DID is a separate, lightweight pre-processing step applied to a fixed CFD dataset, not a per-time-step cost during simulation. As we report in Appx. E, the construction of multi-DID features is computationally negligible compared to running the CFD simulations and is a one-time offline process carried out before training. Once computed, these DID features are reused across all training and evaluation runs at essentially no additional cost.
>
> In summary, **DID is a low-cost geometric encoding for neural network, not a CFD method, and its overhead is negligible in practice relative to the cost of generating the CFD data themselves**.

---

> ### Author Response · Authors · 2025-11-19
> **References**
>
> [1] Bonnet, Florent, et al. "Airfrans: High fidelity computational fluid dynamics dataset for approximating reynolds-averaged navier–stokes solutions." Advances in Neural Information Processing Systems 35 (2022): 23463-23478.
>
> [2] Tobias Pfaff, Meire Fortunato, Alvaro Sanchez-Gonzalez, and Peter Battaglia. Learning mesh-based simulation with graph networks. In International Conference on Learning Representations, 2021.
>
> [3] Alvaro Sanchez-Gonzalez, Jonathan Godwin, Tobias Pfaff, Rex Ying, Jure Leskovec, and Peter W. Battaglia. Learning to simulate complex physics with graph networks. In Proceedings of the 37th International Conference on Machine Learning, ICML 2020, 13-18 July 2020, Virtual Event, volume 119 of Proceedings of Machine Learning Research, pages 8459–8468. PMLR, 2020.
>
> [4] Ryley McConkey, Eugene Yee, and Fue-Sang Lien. A curated dataset for data-driven turbulence modelling. Scientific Data, 8(1), September 2021. ISSN 2052-4463. doi: 10.1038/s41597-021-01034-2. URL http://dx.doi.org/10.1038/s41597-021-01034-2.
>
> [5] Yang, Han, et al. "Wind turbine airfoil noise prediction using dedicated airfoil database and deep learning technology." Applied Energy 364 (2024): 123165.
>
> [6] Wenzhuo Xu, Noelia Grande Gutierrez, and Christopher McComb. MegaFlow2D: A parametric dataset for machine learning super-resolution in computational fluid dynamics simulations. In Proceedings of Cyber-Physical Systems and Internet of Things Week 2023. Association for Computing Machinery, 2023. URL https://doi.org/10.1145/3576914.3587552.
>
> [7] Janny, Steeven, et al. "Eagle: Large-scale learning of turbulent fluid dynamics with mesh transformers." arXiv preprint arXiv:2302.10803 (2023).
>
> [8] Jakob Jakob, Markus Gross, and Tobias Gunther. A fluid flow data set for machine learning and its application to neural flow map interpolation. IEEE Transactions on Visualization and Computer Graphics, 27(2):1279–1289, 2020.
>
> [9] Cao, Y., Chai, M., Li, M., & Jiang, C. (2023, July). Efficient learning of mesh-based physical simulation with bi-stride multi-scale graph neural network. In International conference on machine learning (pp. 3541-3558). PMLR.
>
> [10] Hu, Yeping, Bo Lei, and Victor M. Castillo. "Graph learning in physical-informed mesh-reduced space for real-world dynamic systems." Proceedings of the 29th ACM SIGKDD Conference on Knowledge Discovery and Data Mining. 2023.
>
> [11] Ansari, Abdul Fatir, et al. "Deep explicit duration switching models for time series." Advances in Neural Information Processing Systems 34 (2021): 29949-29961.
>
> [12] Tailin Wu, Takashi Maruyama, Qingqing Zhao, Gordon Wetzstein, and Jure Leskovec. Learning controllable adaptive simulation for multi-resolution physics. In The Eleventh International Conference on Learning Representations, ICLR 2023, Kigali, Rwanda, May 1-5, 2023. OpenReview.net, 2023.
>
> [13] Liu, Jian, et al. "Afbench: A large-scale benchmark for airfoil design." Advances in Neural Information Processing Systems 37 (2024): 82757-82780.
>
> [14] Rajaonson, Ella Miray, et al. "CheMixHub: Datasets and Benchmarks for Chemical Mixture Property Prediction." arXiv preprint arXiv:2506.12231 (2025).
>
> [15] Hassan, Sheikh Md Shakeel, et al. "Bubbleformer: Forecasting Boiling with Transformers." The Thirty-ninth Annual Conference on Neural Information Processing Systems Datasets and Benchmarks Track. 2025.
>
> [16] Kanchi, Rohit Sunil, et al. "UniFoil: A Universal Dataset of Airfoils in Transitional and Turbulent Regimes for Subsonic and Transonic Flows." arXiv preprint arXiv:2505.21124 (2025).
>
> [17] Epstein, Boris, and Sergey Peigin. "Accurate CFD driven optimization of lifting surfaces for wing-body configuration." Computers & fluids 36.9 (2007): 1399-1414.
>
> [18] Ahmed, N., B. S. Yilbas, and M. O. Budair. "Computational study into the flow field developed around a cascade of NACA 0012 airfoils." Computer methods in applied mechanics and engineering 167.1-2 (1998): 17-32.
>
> [19] Bak, Christian, Franck Bertagnolio, and Helge Aagaard Madsen. "Design of High Lift Airfoils With Low Noise and High Aerodynamic Performance." Research in aeroelasticity efp-2007-ii. Danmarks Tekniske Universitet, Risø Nationallaboratoriet for Bæredygtig Energi, 2009. 76-86.
>
> [20] Shelil, Nasser. "2D numerical simulation study of airfoil performance." Wind Energy Science Discussions 2021 (2021): 1-19.

---

> > ### Author Response · Authors · 2025-11-19
> > **Reference List 2**
> >
> > [21] Şugar-Gabor, Oliviu, and Andreea Koreanschi. "Fast and accurate quasi-3D aerodynamic methods for aircraft conceptual design studies." The Aeronautical Journal 125.1286 (2021): 593-617.
> >
> > [22] Jonathan McGlumphy, Wing-Fai Ng, Steven R Wellborn, and Severin Kempf. Numerical investigation of tandem airfoils for subsonic axial-flow compressor blades. In ASME International Mechanical Engineering Congress and Exposition, volume 43092, pp. 53–62, 2007.
> >
> > [23] Bo Yin, Yu Guan, Ao Wen, Nader Karimi, and Mohammad Hossein Doranehgard. Numerical simulations of ultra-low-Re flow around two tandem airfoils in ground effect: Isothermal and heated conditions. Journal of Thermal Analysis and Calorimetry, 145:2063–2079, 2021.
> >
> > [24] Michał Okulski and Maciej Ławry´nczuk. A small UAV optimized for efficient long-range and VTOL missions: An experimental tandem-wing quadplane drone. Applied Sciences, 12(14):7059, 2022.
> >
> > [25] ARS Azmi, A Sapit, AN Mohammed, MA Razali, A Sadikin, and N Nordin. Study on airflow characteristics of rear wing of F1 car. In IOP Conference Series: Materials Science and Engineering, volume 243, pp. 012030. IOP Publishing, 2017.
> >
> > [26] Moloud Arian Maram, Hamid Reza Ghafari, Hassan Ghassemi, and Mahmoud Ghiasi. Numerical study on the tandem submerged hydrofoils using RANS solver. Mathematical Problems in Engineering, 2021:1–17, 2021.
> >
> > [27] Calis, H. P. A., et al. "CFD modelling and experimental validation of pressure drop and flow profile in a novel structured catalytic reactor packing." Chemical Engineering Science 56.4 (2001): 1713-1720.
> >
> > [28] Guichet, Valentin, Bertrand Delpech, and Hussam Jouhara. "Experimental investigation, CFD and theoretical modeling of two-phase heat transfer in a three-leg multi-channel heat pipe." International Journal of Heat and Mass Transfer 203 (2023): 123813.
> >
> > [29] Huakun Luo, Haixu Wu, Hang Zhou, Lanxiang Xing, Yichen Di, Jianmin Wang, and Mingsheng Long. Transolver++: An accurate neural solver for PDEs on million-scale geometries. In Forty-second International Conference on Machine Learning, 2025.

---

> ### Author Response · Authors · 2025-11-27
> **Gentle reminder**
>
> Dear Reviewer CkWJ,
>
> We hope you are doing well. We are writing to kindly follow up on my previous reply and to check whether you’ve had a chance to review it.
>
> If you could share your feedback or confirmation at your earliest convenience, we would really appreciate it.
>
> Thank you again for your time and comments.
>
> Regards,
> Authors of Submission 14913

---

### Official Review · Reviewer_HGdM · 2025-10-31

**Soundness:** 3
**Presentation:** 3
**Contribution:** 3
**Rating:** 6
**Confidence:** 5

**Summary:**

TandemFoilSet introduces nine 2D CFD datasets (five tandem-airfoil and four single-airfoil counterparts; 8,104 total cases, 4,152 tandem) spanning cruise, takeoff (ground effect), and race-car regimes across wide ranges of NACA parameters, Reynolds number, Angle of Attack, stagger, gap, and height. The paper also proposes a curriculum/transfer pipeline that (i) pretrains on single-airfoil flows, (ii) smooth-combines single-airfoil predictions into a low-cost estimate of tandem flow, and (iii) uses residual training to refine tandem predictions. Benchmarks with MeshGraphNet (MGN) and an invariant edge-GCN (IVE) show significant MSE reductions and improved force-coefficient errors compared to baselines. Datasets, meshing/solver details (OpenFOAM, (k-\omega) SST), and verification/validation procedures are documented. A notable technical advancement is the extension of Directional Integrated Distance (DID) from single-object to multi-object geometry encoding via a fast “deviation-from-maximum” combination, yielding significant gains over shortest-vector (SV) features alone.

**Strengths:**

First paired single-to-tandem dataset collection enabling principled curriculum transfer for multi-body aerodynamics (including ground effect). The explicit coupling of dataset design with a smooth-combining + residual strategy is clean and practical. Extension of DID to two-body settings with an efficient approximation is a useful geometry-aware feature innovation.


Datasets cover diverse operating regimes (fixed and random (Re), AoA; stagger/gap/height sweeps), with mesh studies and literature validation for high-(Re) cases. Clear solver/BC documentation for reproducability! Careful ablations isolate contributions from pretraining, smooth-combining, residual learning, and multi-NN decomposition (front/back/upper/lower subgraphs). Reported gains are substantial and consistent across two GNN families. Includes aerodynamic QoIs (lift/drag, boundary-cell errors) in addition to field MSE which is helpful for practitioner relevance.


Timely step toward complex-geometry learning (tandem interactions, ground effect), a frontier problem for CFD surrogates. The “reuse single-airfoil data for multi-airfoil prediction” recipe is likely to influence future multi-body surrogate design.

**Weaknesses:**

*Scope: steady 2D only*:  All data and benchmarks are steady (RANS) and 2D. This limits conclusions for unsteady or 3D tandem interactions (e.g., vortex shedding, dynamic stall, tip effects), which are central in many applications. The authors acknowledge compute limits and suggest future 3D/multi-airfoil stages, but this remains my main issue

*Architecture diversity*: While the paper focuses on GNNs (MGN, IVE), the current SOTA for accuracy in many learned PDE settings often involves transformer-style operators. Even a small transformer baseline or a clear interoperability plan would bolster the benchmarking story. (See questions for concrete paths.)

*Geometry encoding design space*: DID/SV are well motivated, but the paper could better situate them versus (un)signed distance fields (SDF/UDF), level-set encodings, or constructive solid geometry (CSG) fusion; especially since combining distance fields is a natural operation when moving from single to tandem geometries.

**Questions:**

I enjoyed reading this paper and how it could encourage the CFD community to adopt more AI tools. Some of these suggestions are (understandably) difficult to execute on a short rebuttable period, but I provide them for the authors to consider for the future. The critical questions I highlight in bold below:

*Distance-field representations (DIR) via SDF/UDF*: You extend DID to multi-object via a clever, fast combination. Have you considered using an (un)signed distance function as the core DIR, leveraging CAD/CSG-style Boolean merges (min/max, smooth-min) to combine primitives and airfoils? SDFs provide smooth gradients, exact surface normals, and robust interpolation; they could be directionally integrated post-hoc (compute DID on top of SDF), or used directly as multi-channel inputs (distance, normal, curvature). This might (i) simplify multi-body composition, (ii) reduce DID compute, and (iii) improve generalization to arbitrary multi-component layouts. Any reason an SDF-centric pipeline wouldn’t fit your smooth-combining and residual-training framework?

**Transformers on meshes / hybrid models:** Your results champion GNNs; however, many recent surrogate winners are transformer-flavored. Anyway to extend your results? Two concrete suggestions would be to try (a) global tokens atop MGN/IVE (attention across far-field neighbors or via landmark nodes) to better capture long-range tandem wake interactions, (b) Fourier/Transformer hybrids (global spectral mixing for long-range interactions, GNN for local inductive bias). This aligns with the multi-NN approach you already have.

*Time-dependent extension*: Everything here is steady. How would you extend TandemFoilSet to unsteady flows (URANS/LES) and time-dependent learning?

**Generalization beyond airfoils / shape systems** You include some non-NACA profiles and race-car ground-effect cases. Could the dataset (maybe in the future) be extended to include bluff-body to stress fundamentally different separation physics? Can you report on how the approach generalizes to configurations (shape, distances, offsets) that are not in the distribution?

**Compute/latency budgeting**: Residual training with freestream and combined-field estimates is appealing because the estimates are cheap. Could you publish wall-time and memory profiles for (a) baseline, (b) pretrain-only, (c) residual-only, and (d) full pipeline so others can match/compare your cost–accuracy tradeoffs?

---

> ### Author Response · Authors · 2025-11-20
> **Response to W1 and Q3**
>
> We would like to express our gratitude to the reviewer for recognising (i) the usefulness of geometry-aware features (extension of DID), (ii) the diverse operating regimes of datasets with comprehensive documentation for reproducibility, and (iii) the timely step toward complex-geometry learning. Here are our responses to your concerns point by point.
>
> **In response to W1 and Q3 regarding the scope issue:**
>
> We thank the reviewer for highlighting the scope issue. We fully agree that 3D and unsteady tandem interactions (vortex shedding, dynamic stall, tip effects) are central for many applications, and we **already flag these as out of scope for this first release**. Our design choice was to focus on a large, well-controlled 2D steady (RANS) dataset that is compatible with the dominant public benchmarks used by the community, and to study, in a clean setting, how to reuse abundant single-body data for multi-body prediction.
>
> Up to date, most widely used CFD–ML benchmarks are also 2D, steady, and largely single-object, such as AirfRANS [1], the MGN airfoil and CylinderFlow datasets [2], FLAGSimple [3], and Wind Turbine Airfoil Noise Database [5], just to name a few. Table 1 below presents a more comprehensive list for your reference. These datasets, despite being **predominantly 2D and often steady or single-object**, have been highly impactful in shaping architectures and evaluation protocols for mesh-based surrogates. TandemFoilSet is deliberately positioned in this ecosystem where we keep the same 2D steady setting so that (i) existing single-airfoil datasets and models can be **directly reused and compared**, and (ii) the effect of reusing single-body data for tandem-body prediction can be **isolated without factors from turbulence modelling, time integration, or 3D tip effects**.
>
> Regarding time-dependent extensions, we agree that extending TandemFoilSet to URANS/LES and time-dependent learning is an important direction, so we plan to explore this in future work and thank the reviewer for the constructive suggestion.
>
> Table 1: List of available physics-related datasets.
> | Dataset  | Venue / Year  | Citations | Type| Size  | Types of Physcis  |
> |-|-|-|-|--|-|
> | AirfRANS [1] | NeurIPS 2022  | 103 | 2D | 1,000  | Incompressible Navier–Stokes (airfoil external aerodynamics) |
> | MGN airfoil [2] | ICML 2021 | 1365 | 2D    | 230  | High-Reynolds-number incompressible Navier–Stokes (airfoils)    |
> | CylinderFlow [2]     | ICML 2021| 1365| 2D    | 800  | Incompressible Navier–Stokes (bluff-body / cylinder wake) |
> | FLAGSimple [3] | ICML 2020| 1676| 2D    | 800  | Deformable solid mechanics (flexible flags in flow) |
> | Curated Dataset [4]  | Scientific Data 2021 | 84  | 2D    | 841  | Turbulent incompressible flows in canonical channel/duct setups |
> | Wind Turbine Airfoil Noise DB [5] | Applied Energy 2024 | 16 | 2D |11,700 profiles | Aeroacoustics and aerodynamics of wind-turbine airfoils |
> | MegaFlow2D [6] | CPSWEEK 2023   | 11  | 2D    | 3,000 cases| External & internal incompressible flows; steady + unsteady     |
> | EAGLE [7]| ICLR 2023| 60  | 2D    | 1,200| External aerodynamic flow around complex bodies     |
> | Fluid Flow Dataset [8] | IEEE TVCG 2020     | 52  | 2D    | 8,000 (1001 time steps each) | Unsteady incompressible Navier–Stokes parameterized by Reynolds number |
> | InflatingFont [9]    | ICML 2023| 62  | 2D    | 1,000| Fluid–structure interaction with deformable elastic shells|
> | Cavity Flow [10]     | ACM SIGKDD 2023| 11  | 2D    | 1,000| Incompressible Navier–Stokes (lid-driven cavity)    |
> | Bouncing Ball [11]   | NeurIPS 2021   | 38  | 1D    | 1,100| Rigid-body dynamics with impact and contact   |
> | Simulated Cloth [12] | ICLR 2023| 33  | 2D    | 1,050| Deformable-body dynamics (cloth / thin-sheet mechanics)   |
> | AFBench [13]| NeurIPS 2024 |  6 | 2D / Parametric | 20,000+ airfoil configurations | Aerodynamic simulation across laminar-transitional-turbulent regimes; subsonic & transonic compressible flow |
> | CheMixHub [14] | NeurIPS 2025   | –   | Tabular/graph | 500k points, 11 tasks     | Physical chemistry of mixtures (transport, thermodynamics, electrolyte & olfactory properties) |
> | Bubbleformer / BubbleML 2.0 [15] | NeurIPS 2025 | –   | 2D | 160 sims | Multiphase boiling: two-phase liquid–vapor NS + energy equation |
> | UniFoil [16] | NeurIPS 2025 | – | 2D | 500,000 | Transitional & fully-turbulent compressible/incompressible Navier–Stokes (single airfoil flows) |

---

> ### Author Response · Authors · 2025-11-20
> **Response to W2 and Q2**
>
> **In response to W2 and Q2 regarding the architecture diversity:**
>
> We thank the reviewer for raising the question of architecture diversity and transformer-style operators. In the revised manuscript, we have **added a Transformer-based baseline, Transolver [17]**, and report its performance in Appx. J (Tab. 26). For clarity, we report the key results here. Transolver uses exactly the same mesh-based inputs, single-to-tandem training pipeline, and loss definitions as our GCN models, **only the core architecture is changed from GCN to Transformer**. This shows that our pipeline is not restricted to GCN-based architecture but is **in fact architecture-agnostic across three distinct model families** (Transformer, MGN, and IVE).
>
> Table 26 (in the revised manuscript): MSE (×10^-2) performance comparison between Transformer vs. GCN models.
> | Model / Datasets | Cruise AOA=$5^\circ$| Impr. (%) | Cruise Random| Impr. (%) |
> |-|-|-|-|-|
> | **Transolver (Baseline)** | 0.10 ± 0.08| –| 18.68 ± 17.48 | –|
> | **Transolver + PRE-RES-FREE + RES-COMB** | **0.06 ± 0.08** | **41.7**  | **6.25 ± 8.88** | **66.5**|
> | **MGN (Baseline)**| 1.34 ± 0.67 | – | 179 ± 138 | – |
> | **MGN + PRE-RES-FREE + RES-COMB**| **0.67 ± 0.50**| **50.0**| **10 ± 13** | **94.4**|
> | **IVE (Baseline)** | 1.05 ± 0.69| – | 177 ± 137| –|
> | **IVE + PRE-RES-FREE + RES-COMB**| **0.63 ± 0.30**| **40.0**| **7.91 ± 9.30** | **95.5**  |
>
> Across all three architectures, adding our PRE-RES-FREE + RES-COMB model **yields substantial error reductions**, especially on the more challenging Cruise Random dataset. This directly addresses the architecture-diversity concern and indicates that the **proposed curriculum scheme is not an architecture-specific trick**. We agree that further transformer variants, such as global tokens atop MGN/IVE or Fourier/Transformer hybrids, are natural extensions and the present results show that our dataset and benchmark pipeline can already interoperate effectively with such transformer-based architecture.

---

> ### Author Response · Authors · 2025-11-20
> **Response to W3, Q1, and Q4**
>
> **In response to W3 and Q1 regarding the geometry encoding design space:**
>
> We thank the reviewer for pointing out the broader geometry-encoding design space. To directly address the SDF-centric question, we have implemented an SDF-based variant within our pipeline. We replace the SV+DID input with a signed distance field defined on the mesh and keep everything else identical (MGN architecture, losses, curriculum stages). The resulting performance on two representative tandem datasets is:
>
> Tab. 2: MSE ($\times10^{-2}$) performance comparison for SDF vs. SV+DID representations.
> | Model Variant | Cruise AOA=$5^\circ$    | Cruise Random   |
> | - | - | - |
> | **MGN + SDF** | 6.75 $\pm$ 3.38| 190.42 $\pm$ 146.22 |
> | **MGN + SDF + PRE-RES-FREE + RES-COMB**  | 2.09 $\pm$ 1.30     | 18.6 $\pm$ 20.9     |
> | **MGN + SV + DID (Baseline)**| 1.34 $\pm$ 0.67     | 1.79 $\pm$ 1.38     |
> | **MGN + SV + DID + PRE-RES-FREE + RES-COMB** | **0.67 $\pm$ 0.50** | **0.10 $\pm$ 0.13** |
>
> These results show two points:
> - **Compatibility.** An SDF-centric pipeline does fit our framework. The PRE-RES-FREE + RES-COMB curriculum significantly improves the SDF-based model, so **our training scheme is not tied to SV+DID merely**.
> - **Effectiveness.** Despite this, the SDF variants **remain noticeably worse than SV+DID**, especially on the challenging Cruise Random dataset.
>
> In our tandem-airfoil setting, a single global **SDF struggles to encode which airfoil is upstream vs. downstream** and to capture the directional occlusion structure of the wakes. Moreover, for multiple bodies, the SDF field develops non-smooth “creases” where the nearest object switches, which appears to make it harder for the network to learn stable geometric cues compared to the explicitly directional SV+DID encoding.
>
> We therefore see SV+DID as one well-motivated point in the broader design space. It **explicitly encodes object shape and orientation**, empirically yielding much stronger performance than a straightforward SDF baseline under identical conditions. For other geometric encoding schemes mentioned by the reviewer, according to our best knowledge, they have not been studied for fluid simulation problems. We believe studying these geometric descriptions within this paper would distract the focus of this work. Thus, we will consider them in future work.
>
> **In response to Q4 regarding the generalisation beyond airfoils:**
>
> We thank the reviewer for raising the question of generalisation beyond airfoils and in-distribution layouts. Our current release already goes beyond canonical NACA sections by including a set of non-NACA airfoil profiles and race-car ground-effect geometries, and the same mesh pipeline can be applied to other shape families (e.g., bluff bodies, high-lift systems) without further algorithmic changes. In this sense, **extending TandemFoilSet beyond airfoils is technically straightforward**. That say, new shapes can be inserted into the existing parametric generator and overset-mesh workflow to produce additional single/tandem configurations with different separation physics. While such bluff-body extensions are outside the scope of this first release, we see them as a natural next step for future dataset versions.
>
> To directly address the question on **generalisation to unseen distances and offsets**, we have **added an extrapolation experiment on the Cruise Random dataset**. We train on a restricted range of stagger and gap values and evaluate on intervals that lie outside the training range. The results are now reported in Experiment 4 (Sec. 5.4, Tab. 6 of revised manuscript) and summarised below:
>
> Table 6 (in the revised manuscript): MSE performance evaluation of Cruise Random datasets with extrapolation of stagger and gap distances.
> | Model / Data Scheme | Stagger| Gap|
> |-|-|-|
> | **MGN (Baseline)**| 1.74 ± 1.66| 1.95 ± 1.68|
> | **MGN + PRE-RES-FREE + RES-COMB** | **0.13 ± 0.17** | **0.14 ± 0.20** |
> | **Improvement** | **92.5%**| **92.8%**|
>
> These results show that, even when evaluated on stagger/gap configurations outside the training distribution, our curriculum scheme (PRE-RES-FREE + RES-COMB) **substantially improves accuracy and keeps the tandem prediction error in a low range**. This provides **quantitative evidence** that the proposed approach **generalizes not only within the training layout distribution but also to unseen distances and offsets between the tandem components**.
>
> In addition, to illustrate that the training pipeline is not limited to a two-body configuration, we **have included a preliminary three-airfoil configuration in Appx. K**, generated with exactly the same geometry, meshing, and training pipeline. This example demonstrates that the framework **naturally scales to more complex multi-component systems** without any structural changes.

---

> ### Author Response · Authors · 2025-11-20
> **Response to Q5**
>
> **In response to Q5 regarding the computational budget:**
>
> We thank the reviewer for this very practical suggestion. In the revised manuscript, we now report explicit compute profiles for the different training variants. Table 18 (reproduced below for clarity) of the revised manuscript summarises the average total training wall-time and peak GPU memory usage for (a) the baseline model, (b) the PRE-FREE (pretrain-only) stage, (c) the RES-FREE + RES-COMB (residual-only) stage, and (d) the full PRE-RES-FREE + RES-COMB pipeline, all measured on the same GPU and dataset split.
>
> Table 18: Average neural network training wall-time for different training variants.
>  | Regime | Model | Total training time (s) | max GPU Memory Usage (GB) |
> |-|-|-|-|
> | (a)| MGN (Baseline)| 10,452 | 72.88 |
> | (b)| MGN + PRE-FREE (pretrain-only)| 14,517 | 72.88|
> | (c)| MGN + RES-FREE + RES-COMB (res-only) | 11,878 | 76.73|
> | (d)| MGN + PRE-RES-FREE + RES-COMB| 20,459| 76.75|
>
> These numbers show that the residual-only configuration (c) adds only a modest overhead over the baseline (≈14% additional wall-time and a small increase in peak memory), yet already delivers an accuracy gain (see Tab. 3 in the original manuscript). The full curriculum (d) roughly doubles the training time compared to (a) but stays within a similar memory envelope **mainly due to increase in the number of epochs**. However, the full curriculum **(d) achieves significant error reduction up to 70%** as reported in Tab.3 of the original manuscript. We hope this table makes our cost–accuracy trade-offs more transparent and facilitates fair comparisons by other researchers.

---

> ### Author Response · Authors · 2025-11-20
> **Reference list**
>
> References
>
> [1] Bonnet, Florent, et al. "Airfrans: High fidelity computational fluid dynamics dataset for approximating reynolds-averaged navier–stokes solutions." Advances in Neural Information Processing Systems 35 (2022): 23463-23478.
>
> [2] Tobias Pfaff, Meire Fortunato, Alvaro Sanchez-Gonzalez, and Peter Battaglia. Learning mesh-based simulation with graph networks. In International Conference on Learning Representations, 2021.
>
> [3] Alvaro Sanchez-Gonzalez, Jonathan Godwin, Tobias Pfaff, Rex Ying, Jure Leskovec, and Peter W. Battaglia. Learning to simulate complex physics with graph networks. In Proceedings of the 37th International Conference on Machine Learning, ICML 2020, 13-18 July 2020, Virtual Event, volume 119 of Proceedings of Machine Learning Research, pages 8459–8468. PMLR, 2020.
>
> [4] Ryley McConkey, Eugene Yee, and Fue-Sang Lien. A curated dataset for data-driven turbulence modelling. Scientific Data, 8(1), September 2021. ISSN 2052-4463. doi: 10.1038/s41597-021-01034-2. URL http://dx.doi.org/10.1038/s41597-021-01034-2.
>
> [5] Yang, Han, et al. "Wind turbine airfoil noise prediction using dedicated airfoil database and deep learning technology." Applied Energy 364 (2024): 123165.
>
> [6] Wenzhuo Xu, Noelia Grande Gutierrez, and Christopher McComb. MegaFlow2D: A parametric dataset for machine learning super-resolution in computational fluid dynamics simulations. In Proceedings of Cyber-Physical Systems and Internet of Things Week 2023. Association for Computing Machinery, 2023. URL https://doi.org/10.1145/3576914.3587552.
>
> [7] Janny, Steeven, et al. "Eagle: Large-scale learning of turbulent fluid dynamics with mesh transformers." arXiv preprint arXiv:2302.10803 (2023).
>
> [8] Jakob Jakob, Markus Gross, and Tobias Gunther. A fluid flow data set for machine learning and its application to neural flow map interpolation. IEEE Transactions on Visualization and Computer Graphics, 27(2):1279–1289, 2020.
>
> [9] Cao, Y., Chai, M., Li, M., & Jiang, C. (2023, July). Efficient learning of mesh-based physical simulation with bi-stride multi-scale graph neural network. In International conference on machine learning (pp. 3541-3558). PMLR.
>
> [10] Hu, Yeping, Bo Lei, and Victor M. Castillo. "Graph learning in physical-informed mesh-reduced space for real-world dynamic systems." Proceedings of the 29th ACM SIGKDD Conference on Knowledge Discovery and Data Mining. 2023.
>
> [11] Ansari, Abdul Fatir, et al. "Deep explicit duration switching models for time series." Advances in Neural Information Processing Systems 34 (2021): 29949-29961.
>
> [12] Tailin Wu, Takashi Maruyama, Qingqing Zhao, Gordon Wetzstein, and Jure Leskovec. Learning controllable adaptive simulation for multi-resolution physics. In The Eleventh International Conference on Learning Representations, ICLR 2023, Kigali, Rwanda, May 1-5, 2023. OpenReview.net, 2023.
>
> [13] Liu, Jian, et al. "Afbench: A large-scale benchmark for airfoil design." Advances in Neural Information Processing Systems 37 (2024): 82757-82780.
>
> [14] Rajaonson, Ella Miray, et al. "CheMixHub: Datasets and Benchmarks for Chemical Mixture Property Prediction." arXiv preprint arXiv:2506.12231 (2025).
>
> [15] Hassan, Sheikh Md Shakeel, et al. "Bubbleformer: Forecasting Boiling with Transformers." The Thirty-ninth Annual Conference on Neural Information Processing Systems Datasets and Benchmarks Track. 2025.
>
> [16] Kanchi, Rohit Sunil, et al. "UniFoil: A Universal Dataset of Airfoils in Transitional and Turbulent Regimes for Subsonic and Transonic Flows." arXiv preprint arXiv:2505.21124 (2025).
>
> [17] Huakun Luo, Haixu Wu, Hang Zhou, Lanxiang Xing, Yichen Di, Jianmin Wang, and Mingsheng Long. Transolver++: An accurate neural solver for PDEs on million-scale geometries. In Forty-second International Conference on Machine Learning, 2025.

---

> ### Author Response · Authors · 2025-11-27
> **Gentle reminder**
>
> Dear Reviewer HGdM,
>
> We hope you are doing well. We are writing to kindly follow up on my previous reply and to check whether you’ve had a chance to review it.
>
> If you could share your feedback or confirmation at your earliest convenience, we would really appreciate it.
>
> Thank you again for your time and comments.
>
> Regards,
> Authors of Submission 14913

---

### Official Review · Reviewer_PCgw · 2025-10-31

**Soundness:** 2
**Presentation:** 3
**Contribution:** 3
**Rating:** 6
**Confidence:** 4

**Summary:**

This paper introduces TandemFoilSet, a benchmark dataset for tandem airfoils, comprising five distinct configurations and paired single airfoil cases. The authors argue that no public benchmarks currently exist for these situations, which remain relevant in practical engineering contexts. This makes the dataset a meaningful and practical contribution to the CFD and ML-surrogate modeling community.
The paper further explores how single airfoil data can be used for data augmentation or as part of an incremental training strategy for a neural surrogate model based on the Mesh Graph Net (MGN) architecture.

**Strengths:**

- The main strength of the paper lies in the rigor and care taken in generating and documenting the dataset. The simulation parameters, convergence studies, and validation details are described thoroughly, which enhances the dataset’s reliability.
- Tandem configurations are still important for various engineered systems, and having a public dataset here fills a clear gap.
- The dataset covers both low and high Reynolds numbers (500 up to 5×10⁶), includes ground-effect scenarios, and reproduces results from established experimental studies (Figures 13–14), lending further credibility.

**Weaknesses:**

- The dataset is restricted to 2D, primarily NACA 4-digit geometries, and only two-body (tandem) configurations. It’s unclear how well the findings or models generalize to 3D or multi-body scenarios.
- Much of the paper focuses on the benchmark experiments using variants of MGN and a rather elaborate 4-stage transfer learning pipeline. This feels more like a methods paper, even though the main novelty and value lie in the dataset.
- Only MGN variants are tested. It would strengthen the results to include simpler transfer learning or surrogate baselines for context.

**Questions:**

- Can you confirm that the dataset will be made publicly available upon acceptance? What format will it use (HDF5, VTK, or native OpenFOAM)? Will example loading scripts be provided?
- Why were only MGN-based variants considered? Could you compare against simpler transfer learning or domain adaptation approaches, or multi-task setups combining single and tandem cases?
- What do you think is the main contribution of the paper? Is it more a dataset paper or a method paper? The multi-nn inference procedure feels like a methodological description but is stated within a benchmarking setup. I have the impression that the benchmarking setup is somewhat a repurposed method section, and it is unclear why.

---

> ### Author Response · Authors · 2025-11-20
> **Response to W1**
>
> We sincerely appreciate the reviewer for recognising TandemFoilSet as a meaningful and practical benchmark for tandem-airfoil configurations. We also thank your acknowledgement of our study on how single-airfoil data can be leveraged for data augmentation and incremental training of neural surrogate models. Below are the responses to your concerns.
>
> **In response to W1:**
>
> *The dataset is restricted to 2D, primarily NACA 4-digit geometries, and only two-body (tandem) configurations. It’s unclear how well the findings or models generalize to 3D or multi-body scenarios.*
>
> We agree with the reviewer that assessing generalization to 3D and more complex multi-body configurations is an important next step. Our current work deliberately focuses on 2D tandem-airfoil configurations as a controlled starting point, but we emphasize that this is already **the first dataset that systematically pairs single- and tandem-airfoil geometries under high-fidelity CFD** across a wide range of NACA 4-digit shapes, Reynolds numbers, angles of attack, and stagger/gap/height settings. This setting is **standard in the airfoil-level surrogate modeling literature** [1-6] and allows us to isolate the benefits of DID, smooth-combining, residual training, and multi-NN inference without confounding factors.
>
> Conceptually, all proposed components, directional integrated distance (DID) geometry encoding, smooth-combining, freestream/combined-field residual training, and multi-NN domain decomposition, are **not restricted to two bodies or to 2D**. DID is defined for arbitrary numbers of objects, and the multi-NN scheme does not assume that the domain can be partitioned into sub-regions containing at most one body. Extending from 2D to 3D primarily increases CFD and pre-processing cost. As aforementioned, our dataset is the first of this kind. 3D data is not available for evaluation. We will include it in our future work.
>
> To further support the claim that our method generalises beyond two-body tandem configurations (see Appx. K in revised manuscript), we have **generated a new dataset of 125 three-airfoil tandem cases** (three airfoils in series) under the same flow conditions as the Cruise AOA = 5° dataset. The configurations cover
> - Stagger distances between airfoils ranging from 0.5 to 1.0, and
> - Vertical gaps between airfoils ranging from −0.4 to 0.4.
>
> We benchmarked the baseline MGN and our best variant (MGN + PRE-RES-FREE + RES-COMB) on this new three-airfoil dataset and obtained:
>
> Table 27 (of revised manuscript): MSE performance evaluations of newly introduced three-airfoil datasets.
>
> | Model | Overall MSE (×10$^{-2}$) | MSE at Airfoil Boundary (×10$^{-1}$) |
> | -| - | --|
> | **MGN (Baseline)** | 6.49 ± 2.04 | 1.80 ± 1.10 |
> | **MGN + PRE-RES-FREE + RES-COMB** | 2.55 ± 1.12 | 0.97 ± 1.18  |
> | Improvement | 60.7\% |  46.1\% |
>
> These results show that our framework continues to deliver substantial accuracy gains even in more challenging three-body configurations, without any architectural modification beyond the geometry pre-processing.
>
> The revised manuscript now **includes these results in Appx. K (Tab. 27 of revised manuscript) together with qualitative flow-field visualizations** (Fig. 24) that show the predicted velocity and pressure fields and their absolute error contours for a representative three-airfoil case. The figure illustrates that our best model **accurately captures the complex wake–wake interactions and near-body flow**, with only small, localised discrepancies in the shear layers and wake regions.
>
> Overall, these additional experiments provide solid evidence that the proposed framework is not narrowly tuned to a specific two-body setting, but **scales to more complex three-body interactions without architectural changes**. We will continue to extend the dataset towards more diverse configurations, including 3D geometries, in future work, and we appreciate the reviewer’s comment for motivating these directions.

---

> ### Author Response · Authors · 2025-11-20
> **Response to W2, Q3 and Q1**
>
> **In response to W2 and Q3 regarding the position of this work:**
>
> We appreciate the reviewer’s observation and understand the concern. However, we would like to clarify that the paper is **intentionally designed around two equally important contributions**:
> - TandemFoilSet is a collection of five high-fidelity tandem-airfoil flow-field datasets paired with four single-airfoil datasets under various flow settings. It is the **first to comprehensively capture interactions between front and rear airfoils**, providing a foundation for benchmarking data-driven surrogate models in tandem-airfoil aerodynamic environments.
> - Benchmark curriculum-learning scheme. We introduce a **novel benchmark curriculum learning scheme in which models are first trained on single-airfoil flow fields and then fine-tuned on tandem-airfoil configurations in a memory-efficient multi-network approach**. Across the tasks we consider, this scheme on average outperforms the corresponding baselines by about 65% in MSE, making it a strong and reusable baseline for future work on TandemFoilSet.
>
> We believe that establishing only TandemFoilSet without such a well-defined, reproducible curriculum-learning scheme would make it **difficult for the community to compare methods and to measure progress on the reuse of existing single-object datasets**. Thus, our intention in the current manuscript is precisely to present (i) the TandenFoilSet dataset and (ii) the associated curriculum-learning method side by side, rather than treating the latter as an auxiliary “implementation detail” of the former. We hope the proposed method serves as an example to motivate researchers to design better methods to reuse CFD data, since this research direction has not been exploited in the community.
>
> **In response to Q1 regarding the availability of dataset:**
>
> Yes, we confirm that the full TandemFoilSet dataset **will be made publicly available upon acceptance**. The data will be released in both native OpenFOAM format (as used for the original CFD simulations) and a processed HDF5 format that stores mesh connectivity, geometry, and flow fields in a backend-agnostic way suitable for machine learning workflows. Alongside the raw data, we will provide example Python scripts that (i) load the HDF5 files and construct graph/mesh data structures for surrogate models, and (ii) extract and process the native OpenFOAM cases into the same HDF5 representation, including the exact preprocessing steps used in our experiments so that our results are fully reproducible.

---

> ### Author Response · Authors · 2025-11-20
> **Response to W3 and Q2**
>
> **In response to W3 and Q2 regarding the “Only MGN variants' issue:**
>
> Our intention is not to restrict the study to MGN. The training pipeline is deliberately kept lightweight, model-agnostic, and not limited to MGN. It can be applied to any mesh-based surrogate model and is not tied to a particular architecture. In the **original manuscript**, we demonstrate this **using two GCN-style models (MGN and IVE) and their variants**.
>
> To further support this point and address the reviewer’s concern, the revised manuscript **additionally reports results for a Transformer-based architecture**, Transolver [7], in Appx. J (Tab. 26) reproduced the table below for readability. This uses the same mesh-based inputs and training procedure with **only the architecture changed from GCN to Transformer**. Thus, the experimental section is not limited to “MGN-only” variants, but instead **demonstrates the architecture-agnostic nature** of the proposed pipeline across three distinct model families.
>
> For clarity, the numerical results (MSE in $\times10^{-2}$) and relative improvements are summarized below:
>
> Table 26: MSE (×10^-2) performance comparison between Transformer vs. GCN models.
>
> | Model \ Datasets|Cruise AOA=$5^\circ$| Impr. (%) | Cruise Random| Impr. (%) |
> |-|-|-|-|-|
> | **Transolver (Baseline)** | 0.10 ± 0.08| – | 18.68 ± 17.48| – |
> | **Transolver + PRE-RES-FREE + RES-COMB** | **0.06 ± 0.08** | **41.7**  | **6.25 ± 8.88** | **66.5**|
> | **MGN (Baseline)**| 1.34 ± 0.67 | – | 179 ± 138| – |
> | **MGN + PRE-RES-FREE + RES-COMB** | **0.67 ± 0.50**| **50.0**| **10 ± 13**| **94.4** |
> | **IVE (Baseline)**| 1.05 ± 0.69| –| 177 ± 137| – |
> | **IVE + PRE-RES-FREE + RES-COMB**| **0.63 ± 0.30**| **40.0**| **7.91 ± 9.30**| **95.5**|
>
> Across all three architectures, applying the PRE-RES-FREE + RES-COMB model reduces the error substantially, especially on the more challenging Cruise Random dataset where the MSE often drops by more than an order of magnitude. This supports our view that the **proposed curriculum learning scheme is a general, reusable technique** for the single-to-tandem transfer problem rather than an architecture-specific trick.
>
> **In response to W3 and Q2 regarding the additional comparisons:**
>
> *"It would strengthen the results to include simpler transfer learning or surrogate baselines for context."*
>
> *"Could you compare against simpler transfer learning or domain adaptation approaches, or multi-task setups combining single and tandem cases?"*
>
> **(a) Simpler transfer learning baselines.**
>
> We agree that simple transfer-learning baselines are important, and we have **already implemented and evaluated them for both MGN and IVE**. In addition to the “tandem-only” baselines trained purely on tandem data, we included variants where models are first pre-trained on single-airfoil flows, and then directly fine-tuned on tandem cases (corresponding to the PRE-only variant in our ablations).
>
> These PRE-only models already improve over “tandem-only” (baseline) training, showing that even naive single-to-tandem transfer is beneficial. However, our **full curriculum (PRE–RES–FREE + RES-COMB) consistently provides non-trivial additional gains beyond this simple pre-train–then–fine-tune (PRE) scheme for both MGN and IVE across the evaluated datasets** (see the ablation Tab. 3 in the original manuscript). This is precisely why we present the curriculum as a methodological contribution, not just a training detail.
>
> **(b) Domain adaptation approaches**
>
> By “domain adaptation”, we understand generic techniques that enforce invariance between a source and a target domain as commonly used in computer vision and NLP. Applying such generic domain-adaptation machinery in our setting is non-trivial, saying that single- and tandem-airfoil configurations live on different meshes, involve different numbers of bodies, and induce qualitatively different wake structures, so directly matching feature distributions between them can easily destroy flow-physical information.
>
> We have not yet explored sophisticated domain-adaptation objectives because this would open a large and orthogonal design space and significantly broaden the scope of the paper beyond the TandemFoilSet benchmark and our physics-based curriculum scheme. We instead focus on a problem-structured transfer that exploits the specific geometry and flow relationships between single and tandem cases, and leave more general domain-adaptation techniques for PDE surrogates as promising future work.

---

> > ### Author Response · Authors · 2025-11-20
> > **Continue For Response to W3 and Q2**
> >
> > **(c) Multi-task setups combining single and tandem cases**
> >
> > We agree that a joint multi-task setup that mixes single- and tandem-airfoil datasets during training is a natural additional baseline. We have now **conducted such a multi-task experiment** (and updated this experiment in the revised manuscript), in which the MGN architecture is used, and the model is trained jointly on both single- and tandem-airfoil cases and then evaluated on the tandem test sets. Table 7 below, as shown in pg.10 of the revised manuscript, reports the results for the Cruise AOA = 5° and Cruise Random datasets for different flow regimes. Compared to the multi-task baseline, our curriculum scheme **yields a substantial improvement, reducing at least one order of MSE value for both datasets**. Hence, these results support the same qualitative conclusion:
> > - TandemFoilSet defines a challenging and practically relevant single-to-tandem dataset, and
> > - The proposed benchmark curriculum-learning scheme provides a **strong, architecture-agnostic learning technique** that significantly improves over tandem-only (baseline), simple transfer (PRE), and multi-task setup models.
> >
> > Table 7 (in the revised manuscript): MSE performance comparison between a multi-task setup and the proposed curriculum scheme for the two Cruise AOA=$5^\circ$ and Cruise Random datasets.
> >
> > | **Model / Datasets** | **Cruise AOA=5°** (×10$^{-2}$) | **Cruise Random** |
> > |-|-|-|
> > | MGN + Multi-task | 1.99 $\pm$ 1.04 |  1.80 $\pm$ 1.34 |
> > | MGN + PRE-RES-FREE + RES-COMB | **0.67 $\pm$ 0.50** | **0.10 $\pm$ 0.13** |
> > | Improvement | **66.3%** |  **94.4%**   |
> >
> > References
> >
> > [1] Bonnet, Florent, et al. "Airfrans: High fidelity computational fluid dynamics dataset for approximating reynolds-averaged navier–stokes solutions." Advances in Neural Information Processing Systems 35 (2022): 23463-23478.
> >
> > [2] Tobias Pfaff, Meire Fortunato, Alvaro Sanchez-Gonzalez, and Peter Battaglia. Learning mesh-based simulation with graph networks. In International Conference on Learning Representations, 2021.
> >
> > [3] Alvaro Sanchez-Gonzalez, Jonathan Godwin, Tobias Pfaff, Rex Ying, Jure Leskovec, and Peter W. Battaglia. Learning to simulate complex physics with graph networks. In Proceedings of the 37th International Conference on Machine Learning, ICML 2020.
> >
> > [4] Yang, Han, et al. "Wind turbine airfoil noise prediction using dedicated airfoil database and deep learning technology." Applied Energy 364 (2024): 123165.
> >
> > [5] Liu, Jian, et al. "AFBench: A large-scale benchmark for airfoil design." Advances in Neural Information Processing Systems 37 (2024): 82757-82780.
> >
> > [6] Kanchi, Rohit Sunil, et al. "UniFoil: A Universal Dataset of Airfoils in Transitional and Turbulent Regimes for Subsonic and Transonic Flows." arXiv preprint arXiv:2505.21124 (2025).
> >
> > [7] Huakun Luo, Haixu Wu, Hang Zhou, Lanxiang Xing, Yichen Di, Jianmin Wang, and Mingsheng Long. Transolver++: An accurate neural solver for PDEs on million-scale geometries. In Forty-second International Conference on Machine Learning, 2025.

---

> ### Author Response · Authors · 2025-11-27
> **Gentle reminder**
>
> Dear Reviewer PCgw,
>
> We hope you are doing well. We are writing to kindly follow up on my previous reply and to check whether you’ve had a chance to review it.
>
> If you could share your feedback or confirmation at your earliest convenience, we would really appreciate it.
>
> Thank you again for your time and comments.
>
> Regards,
> Authors of Submission 14913

---

### Official Review · Reviewer_rNNu · 2025-11-01

**Soundness:** 3
**Presentation:** 2
**Contribution:** 3
**Rating:** 4
**Confidence:** 4

**Summary:**

This paper introduces TandemFoilSet, a new collection of CFD simulation datasets for tandem-airfoil configurations. It also proposes a novel curriculum learning framework designed to reuse single-airfoil simulation data to predict the more complex tandem-airfoil flow fields. This method is presented as a benchmark, involving techniques like freestream-based residual pre-training and a multi-network domain decomposition approach.

However, this paper seems rushed with many internal contradictions. It also blurs its own contribution (a benchmark, or a better training methodology?). In it's current shape, I have to issue reject. But I am open to rebuttal and reconsideration if authors can fix the problems.

**Strengths:**

1.  The primary contribution is a new, large-scale, public dataset for a complex multi-body aerodynamics problem.[1] This is a valuable artifact for the community, as tandem-airfoil configurations are critical in many engineering applications but under-represented in existing benchmarks.
2.  The dataset appears comprehensive, covering a variety of flow conditions (low and high Reynolds numbers), angles of attack, and geometries, including ground-effect scenarios (Takeoff, Race Car).

**Weaknesses:**

This paper is not ready for publication and suffers from significant flaws, ranging from internal contradictions to under-evaluated methodological claims.

1.  The paper is rife with basic contradictions, suggesting a rushed submission.
    *   **Dataset Count:** The text repeatedly claims the collection contains "five tandem-airfoil datasets" and "four single-airfoil datasets". However, the paper's own **Table 1** summary clearly enumerates a structure of **three** tandem-airfoil datasets and **six** single-airfoil datasets. This is a critical contradiction regarding the paper's primary contribution.
    *   **Dataset Size:** The abstract claims "over 4000 fluid simulations". The main body and appendix state a total of "8104 cases".
    *   **Dataset Naming:** The naming in Table 1 is ambiguous. For instance, the "TAKEOFF" dataset appears to contain *both* single and tandem cases, further confusing the "5 vs. 4" or "3 vs. 6" structure.

2.  The paper's focus is split. It presents itself as a dataset/benchmark paper, but its novelty is heavily invested in a new, bespoke "curriculum learning" method. A benchmark paper should ideally use established methods for a fair comparison, not a new method that is itself under-evaluated.
    *   This is problematic because the proposed method is not thoroughly tested before. The paper details a 4-part "Multi-NN" architecture (front, back, upper, lower) in Section 4.4.
    *   However, in the experiments (Section 5.3), only a 2-part (front, back) model was tested. The paper admits the "upper and lower fields were excluded due to memory limitations". This admission undermines the method's own claim of being "memory-efficient" and confirms that the paper's core methodological proposal was not actually validated.

3.   The paper's methodological contributions are overstated.
    *   The "First use of freestream condition... as a physics prior for residual pre-training"  is suspect and maybe automatically conducted by previous methods, because previous papers will usually adopt data normalization. If a standard Z-score (mean=0, std=1) normalization was applied, the mean velocity (which is likely the freestream velocity due to the large portion of surrounding meshes) would have been removed automatically. Hence, this "contribution" may simply be a implicitly-effect of standard preprocessing.
    *   Finally, this new method's inference, Table 15, is way to high (about 1/4th of the GT simulation, Table 14); this makes the whole "re-use" questionable due to its high inference cost.

**Questions:**

1.  Which is the correct dataset structure: the "five tandem / four single" datasets claimed in the text, or the "three tandem / six single" datasets shown in Table 1?
2.  What is the correct total case count for the dataset: "over 4000" (from the abstract) or "8104" (from the main text)?
3.  What specific data normalization (e.g., Z-score, min-max scaling, division by freestream velocity) was applied to the flow fields before training? This is essential for reproducibility and for validating the freestream residual claim.
4.  Given that the proposed 4-part Multi-NN architecture from Section 4.4 was not tested, can the authors clarify if *all* results in the paper were generated using the simplified 2-part model mentioned in Section 5.3?
5. Why is your new "re-use" method so slow? what can be the limitations?

---

> ### Author Response · Authors · 2025-11-19
> **Response to W1, Q1, and Q2**
>
> We appreciate that the reviewer recognises (i) the **new, large-scale public dataset for complex multi-body (tandem-airfoil) aerodynamics** as a valuable artifact for the community, and (ii) the **comprehensive coverage of flow conditions**, spanning low and high Reynolds numbers, multiple angles of attack, and ground-effect configurations such as Takeoff and Race Car. Building on these strengths, we now turn to clarifying the reviewer’s remaining concerns.
>
> **In response to W1, Q1, and Q2 regarding the confusing dataset description:**
>
> We thank the reviewer for carefully checking the dataset description and for pointing out that our current presentation is confusing. The underlying dataset structure and counts are **internally consistent**, but we realise that the way we summarised them in the abstract and in Tab. 1 of the original manuscript made this easy to misinterpret. We have revised the manuscript to make the structure explicit and to align all numbers.
>
> First, regarding the dataset structure (“five tandem / four single” vs “three tandem / six single”), the correct structure is the five tandem-airfoil datasets paired with four single-airfoil datasets stated in the contribution bullet (p.1, ln. 45-46). With that, each tandem-airfoil dataset has a single-airfoil counterpart:
> | Single-airfoil datasets (4) | Tandem-airfoil datasets (5) |
> |-|-|
> | Single AOA = 0° | Cruise AOA = 0° |
> | Single AOA = 5° | Cruise AOA = 5°  |
> | Single AOA = 5° | Takeoff  |
> | Single Random | Cruise Random |
> | Single Inverted | Race Car |
>
> In the original Tab. 1, these nine datasets are grouped by flow condition, and the notation “×2” in the “CASES” column (e.g., “SINGLE 1014×2”, “CRUISE 784×2”) denotes two datasets at different AOAs (0° and 5°) under the same flow-condition family, rather than “single vs tandem” or “airfoil count”. This compact grouping, **chosen originally to satisfy the page limit**, unfortunately obscures the 5-vs-4 structure and makes it look as if there were only three tandem and six single datasets. In the revised manuscript, we have
> - (i) paired them up accordingly, and
> - (ii) stated directly in the caption which dataset corresponds to the single- or tandem-airfoil configuration.
>
> This should remove the ambiguity that led to the perceived contradiction. The revised Tab. 1 is shown below for reference.
>
> Second, on the total case count (“over 4000” vs “8104”), the correct total number of simulations in TandemFoilSet is 8104 cases, of which 4152 are tandem-airfoil configurations, as already stated in p. 3 ln. 135 and Appx. A.1. The phrase “over 4000 fluid simulations” in the abstract was intended to refer only to the tandem-airfoil subset (4152 tandem cases), since the benchmark focus is on tandem configurations, whereas the full collection (single + tandem) comprises 8104 cases. We agree that this distinction is not clear in the current wording and appears contradictory when contrasted with the later statement “8104 cases”. In the revised abstract, we will explicitly state both numbers, for example: “We present TandemFoilSet: five tandem-airfoil datasets (4152 tandem-airfoil simulations) paired with four single-airfoil counterparts, for a total of 8104 CFD cases.”
> We also replaced “over 8000 cases” in Sec. 3 with the same explicit “8104 cases” wording, so that the abstract, main text, and appendix all use the same numbers and terminology.
>
> Third, concerning the naming in Table 1 and the TAKEOFF dataset, we confirm that **TAKEOFF dataset contains only tandem-airfoil cases** at Re = 500 and AOA = 5°, with ground effect (height variations), as described in Sec. 3.1. There is no single-airfoil counterpart mixed into the TAKEOFF dataset itself. The old table layout, which groups all low-Re, fixed-AOA Cruise cases in a single “CRUISE” row and all low-Re, near-ground takeoff cases in a single “TAKEOFF” row, does not explicitly indicate “tandem only” vs “single only” for each row. This is what makes it look as though TAKEOFF might mix single and tandem cases. In the revised manuscript, we have indicated in Tab. 1 that TAKEOFF is a tandem-airfoil dataset only.
>
> In summary, the **dataset structure** (five tandem-airfoil datasets paired with four single-airfoil datasets) and **the total case count** (8104 cases, including 4152 tandem-airfoil configurations) are **internally consistent in our data and experiments**, but we acknowledge that our compact (original) Tab. 1 design and the abstract wording made this easy to misinterpret. We have revised the abstract, Sec. 3, Tab. 1 (layout and caption), and Appx. A.1 to clearly and consistently present (i) the nine datasets, (ii) their single vs tandem configuration, (iii) the per-dataset case counts, and (iv) the breakdown “4152 tandem out of 8104 total”. We hope these changes will address the reviewer’s concerns and remove the impression of contradiction.

---

> > ### Author Response · Authors · 2025-11-19
> > **Revised Table 1**
> >
> > Table 1: Total number of cases, average grid cells and the range of $Re$ and AOA as well as stagger, gap, and height (see Fig. 1) normalised by the chord length, for TandemFoilSet. *Asterisk marks indicate the datasets are in tandem-airfoil configuration. More are available in Appx. A.
> > | Dataset| Cases | Ave. cells | Re    | AOA [°] | Stagger | Gap      | Height |
> > |--|:-:|:-:|--|--|--|--|--|
> > | Single AOA = 0°| 1014  | 122788     | 500   | 0| -| - | -|
> > | *Cruise AOA = 0° | 784   | 351315     | 500   | 0| [0.5, 2]| [-0.4, 0.4]     | -|
> > | Single AOA = 5°| 1014  | 122788     | 500   | 5| -| - | -|
> > | *Cruise AOA = 5° | 784   | 351315     | 500   | 5| [0.5, 2]| [-0.4, 0.4]     | -|
> > | *Takeoff| 784   | 271316 | 500   | 5| [0.5, 2]| [-0.2, 0.6]     | [0.4, 1]|
> > | Single Random | 1025 | 111370     | [10^5, 5×10^6]| [-5, 7] | -| - | -|
> > | *Cruise Random| 900 | 210181     | [10^5, 5×10^6]| [-5, 6] | [0.5, 2]| [-0.8, 0.8]     | -|
> > | Single Inverted| 899 | 87108| [10^5, 5×10^6]| [-10, 0]| -| - | [0.1, 1.1]    |
> > | *Race car  | 900 | 130276 | [10^6, 5×10^6]| [-10, 0]| [-0.5, 0.05] | [0.05, 0.1] | [0.1, 1.1]    |

---

> ### Author Response · Authors · 2025-11-19
> **Response to W2 and Q4**
>
> **In response to W2 regarding the contribution and position of this work:**
>
> We thank the reviewer for the comment on the balance between the datasets and the proposed training benchmark method. Our intention is **not to present a “pure” dataset-only paper, but rather a work with two tightly coupled and equally important contributions** as stated in the Introduction:
> - The TandemFoilSet dataset for single-to-tandem airfoil flow prediction, and
> - A benchmark curriculum-style “re-use” scheme (pre-training, residual training, and Multi-NN inference) specifically designed for this new transfer setting.
>
> To ensure fair and transparent evaluation, we have included results for well-established architectures, MeshGraphNet (MGN) and invariant edge-GCNN (IVE), trained with standard procedures as strong baselines in the original manuscript. The curriculum scheme is then applied on top of the same architectures, with identical dataset splits and metrics, and its effect is analysed via ablation studies. In this sense, the dataset and curriculum method serve complementary roles, for which the **TandemFoilSet defines a new, practically motivated problem regime, and the curriculum scheme provides a principled, physics-informed solution strategy to reuse single airfoil simulations.** Without the method, readers will find it hard to realise how to re-use single airfoil simulations for this task. It also serves as an example to motivate researchers to develop different methods to re-use CFD simulations for different applications.
>
> The contribution list found in the Introduction section of the original manuscript **consistently presents both the dataset and the benchmark curriculum learning scheme as central, coequal contributions**, while keeping the experimental design and comparison against standard baselines transparent. Furthermore, our intent is very much in line with common practice in the community, where benchmark papers often introduce both a new dataset and a tailored reference method. For example, Guo et al. [1] introduced the DOLOS deception-detection dataset together with the PECL parameter-efficient crossmodal learning method, both treated as main contributions in a single work. Similarly, MT-Bench and Chatbot Arena are proposed together with an LLM-as-a-judge evaluation scheme, again combining a dataset and method in one paper [2].
>
> Since both the abstract and Introduction explicitly state these two contributions, we believe the **current text already reflects this dual focus.** The only possible source of confusion may be the title, which currently emphasises the dataset and may lead to the expectation of a “benchmark-only” paper. If the reviewer feels this is a significant issue, we are happy to revise the title to make dual nature explicit. We hope this clarification addresses the reviewer’s concern.
>
> **In response to W2 and Q4 regarding the misunderstanging of multi-NN procedure:**
>
> We also thank the reviewer for the detailed reading of Sec. 4.4 and Sec. 5.3 (Experiment 3) and for highlighting the potential confusion around the Multi-NN configuration. We confirm that **all results reported for the Multi-NN approach in Sec. 5 were obtained using the full four-part Multi-NN architecture described in Sec. 4.4**, i.e., with separate front, back, upper, and lower subnetworks defined over their respective subdomains. The statement in Sec. 5.3 about “upper and lower fields being excluded due to memory limitations” refers **only to the single-NN variants** used for comparison in Experiment 3, where we restricted the single-network model to a reduced (front and back) region in order to fit a single, monolithic GNN on the available GPU memory.
>
> In contrast, the **Multi-NN model always predicts the full domain via four NNs** (front, back, upper, lower), and this is precisely how it achieves practical memory efficiency. Each subnetwork operates on a smaller graph, enabling full-domain prediction under the same hardware constraint where the single-NN baseline must be truncated. We apologise that the current wording makes it possible to infer that the upper and lower subnetworks were removed from the Multi-NN model.
>
> In the revision, we will explicitly state in Sec. 4.4 (highlighted in blue in revised manuscript) that **(i) the four-part Multi-NN architecture is used in all experiments, except (ii) only the single-NN variants in Experiment 3** (to evaluate the Multi-NN effectiveness) omit the upper and lower subdomains and considers front and back as single domain due to the maximum GPU memory allowed **since each graph/sample has more than 200k nodes and 700k edges**. We hope this clarifies that the core methodological proposal was in fact implemented and evaluated as described, and that the experimental results consistently use the full four-part Multi-NN architecture.

---

> ### Author Response · Authors · 2025-11-20
> **Response to W3 and Q3**
>
> **In response to W3 and Q3 regarding the overclaim of methodology contribution:**
>
> We thank the reviewer for carefully examining our “freestream residual” contribution and for asking about the exact normalisation procedure. First, regarding the concern that our “*first use of freestream condition as a physics prior for residual pre-training*” might simply be an **implicit effect of Z-score normalisation, this does not apply to our work**.
>
> In the baseline model, the flow variables, $u$, $v$, $p$, are not used as input features, $x$, at all but appear only as prediction targets. The inputs consist of geometric and structural information (mesh coordinates, connectivity, boundary indicators, etc.), while the outputs are normalised for training by a standard Z-score–style procedure (using training-set statistics). Thus, the baseline model undergoes a standard z-score normalization for both input and output.
>
> Moreover, in the freestream-aware model variants (those marked with “+FREE”), we **explicitly introduce the freestream condition as additional input features**. In other words, we add freestream values, $u_\infty$, $v_\infty$, $p_\infty$, in the node features and **normalize them using the mean freestream value over the training set**, so that they become **non-zero constants** in the input space rather than vanishing to zero. These constants encode the global flow condition in a physically meaningful way and are used in our residual training scheme. The key point is that this is an explicit physics prior injected through the inputs, not an artefact of zero-mean normalisation on the targets.
>
> The effectiveness of this prior is directly visible in the single-airfoil experiments as shown in Tab. 2 below, where we can isolate the effect of adding freestream as input features. For example, on the Single AOA = 0° dataset, the MGN + FREE model yields an error reduction of about 62.5% as compared to the baseline MGN model. On the Single Random dataset, the MGN + FREE achieves a roughly 96% reduction in error. Additionally, on the Single Inverted dataset, 78% of error reduction has been achieved. These **substantial and consistent improvements** occur on top of the same underlying normalisation pipeline, and the **only difference between the compared models is the existence of freestream input features**. This provides solid evidence that the freestream prior is doing more than what standard Z-score normalisation alone can achieve.
>
> Table 2: Effectiveness of using freestream as input feature in single-airfoil experiments.
>
> |                | Single AOA = 0°      | Single Random       | Single Inverted      |
> |----------------|----------------------|----------------------|----------------------|
> | MGN            | 0.008 ± 0.006        | 1.172 ± 1.117        | 1.080 ± 1.373        |
> | MGN + FREE     | 0.003 ± 0.002        | 0.041 ± 0.082        | 0.232 ± 0.287        |
> | Improvement        | 62.5%                | 96.5%                | 78.5%                |
>
> In summary,
> - (i) the baselines do not feed flow variables into the input features therefore cannot implicitly “encode freestream” via normalisation,
> - (ii) the +FREE variants explicitly add freestream as normalised, non-zero constant input features after performed z-score normalization, and
> - (iii) ablation results on the single-airfoil datasets show large, systematic gains when freestream is used as an input physics prior.
>
> On this basis, we maintain our contribution statement that this is, to the best of our knowledge, the first use of freestream condition as a physics prior for residual pre-training in the considered single-to-tandem airfoil curriculum-learning scheme, and we have clarified the normalisation description in Appx. B to make the implementation fully reproducible.

---

> ### Author Response · Authors · 2025-11-20
> **Response to W3 and Q5**
>
> **In response to W3 nd Q5 regarding the computational cost:**
>
> We thank the reviewer for raising concerns about the computational cost of our “re-use” scheme and for pointing us to Tabs. 14-16 (corresponding to Tabs. 15-17 in revised manuscript). The impression that our method is “slow” mainly comes from focusing on (i) **the training wall-clock time** and (ii) a **direct comparison of the numbers in the Tabs. 14 and 15** (Tabs. 15 and 16 in revised manuscript), rather than on the relevant metric for a surrogate model, which is **inference time versus pure CFD simulation**. In our setting, training is a one-time offline cost that is amortised over potentially thousands of downstream evaluations, whereas CFD must pay the full numerical simulation cost for every new case.
>
> Regarding the apparent “slowness” of the proposed “re-use” (curriculum) scheme, the **large training time is entirely expected given the scale of the dataset**. Each training configuration involves on the **order of ~1000 CFD cases**, and each case is represented as a large mesh **graph with more than 200k nodes and 700k edges**. Even a single-stage “simple training” (baseline model) on such graphs is much more expensive than typical CFD–ML benchmarks on smaller meshes [3-5]. Our “re-use” scheme adds physically motivated stages (single-airfoil pre-training, residual learning, and Multi-NN training), which necessarily increases the total wall training time, exactly as in standard curriculum or transfer-learning pipelines. This behaviour is normal and should not be compared directly to the cost of a numerical CFD solver, which serves a different purpose. The **meaningful comparison is instead between the inference time of the trained model and the pure CFD simulation time**. As reported in the Tabs. 15 and 16 (Tabs. 16 and 17 in revised manuscript), once training is completed, the trained model predicts the full tandem-airfoil flow field significantly faster than the ground-truth CFD solver, as its **per-step inference time is far below the cost of running a full CFD simulation** for the same configuration. We will clarify this distinction in the manuscript by explicitly stating that
> - (i) training time is dominated by the large dataset size and graph resolution, and
> - (ii) the key efficiency benefit of our approach lies in the acceleration of inference relative to pure CFD, not in reducing the one-time training cost.
>
> Second, the comment that our inference time “is about 1/4th of the GT simulation time” stems from a **misunderstanding of the units in the Tabs. 14 and 15** (Tabs. 15 and 16 in revised manscript). Table 14 (Tab. 15 in revised manuscript) reports the **wall time** of the ground-truth CFD simulation on a **64-core CPU setup**. To compare computational cost, one needs to account for these 64 cores for which the **total CPU time of CFD is the wall time multiplied by 64**. When this is done, the effective CPU time of CFD becomes orders of magnitude larger than the model’s inference time. Conversely, Tab. 15 reports the inference time of our trained model on a single GPU. **Under a fair comparison (total CPU time vs single-GPU inference time), the trained model is far cheaper than CFD**; it is not 4× slower, but **several orders of magnitude faster** in terms of CPU time required to obtain a solution (23372s CPU times vs 65.57s GPU times). This is exactly why we introduce the Multi-NN procedure by decomposing the domain and assigning each subdomain to a smaller GNN. We keep the per-network memory footprint manageable and **achieve fast full-field prediction**, just like how the CFD simulations have to decompose the full domain into 64 CPU cores.
>
> To avoid this confusion, we have revised the manuscript to:
> - (i) explicitly state in the text and caption of Tab. 14 that the CFD wall time is measured on 64 CPU cores; and
> - (ii) re-express the CFD cost in terms of total CPU time
>
> to make the comparison with model inference more transparent, explicitly **highlighting that the Multi-NN model’s inference time is far below the effective CFD cost** once hardware parallelism is taken into account. With these clarifications, we believe it becomes clear that the “re-use” scheme is not questionable due to high inference cost. The apparent “slowness” is a result of reading wall times from different hardware configurations as if they were directly comparable, whereas under a fair CPU-time comparison, our model delivers substantial acceleration over ground-truth CFD.

---

> ### Author Response · Authors · 2025-11-20
> **Reference List**
>
> Reference
>
> [1] Guo, Xiaobao, et al. "Audio-visual deception detection: Dolos dataset and parameter-efficient crossmodal learning." Proceedings of the IEEE/CVF International Conference on Computer Vision. 2023.
>
> [2] Zheng, Lianmin, et al. "Judging llm-as-a-judge with mt-bench and chatbot arena." Advances in neural information processing systems 36 (2023): 46595-46623.
>
> [3] Bonnet, Florent, et al. "Airfrans: High fidelity computational fluid dynamics dataset for approximating reynolds-averaged navier–stokes solutions." Advances in Neural Information Processing Systems 35 (2022): 23463-23478.
>
> [4] Tobias Pfaff, et al. Learning mesh-based simulation with graph networks. In International Conference on Learning Representations, 2021.
>
> [5] Liu, Jian, et al. "AFBench: A large-scale benchmark for airfoil design." Advances in Neural Information Processing Systems 37 (2024): 82757-82780.

---

> ### Author Response · Authors · 2025-11-27
> **Gentle reminder**
>
> Dear Reviewer rNNu,
>
> We hope you are doing well. We are writing to kindly follow up on my previous reply and to check whether you’ve had a chance to review it.
>
> If you could share your feedback or confirmation at your earliest convenience, we would really appreciate it.
>
> Thank you again for your time and comments.
>
> Regards,
> Authors of Submission 14913

---

### Author Response · Authors · 2025-12-02
**Author Final Remarks**

We sincerely appreciate the Area Chair’s time and effort in evaluating our paper, especially given this year’s heavy workload. To streamline your assessment, we provide the brief summary below to highlight the main issues addressed in our rebuttal.

In this paper, we introduce **TandemFoilSet, a large-scale public dataset of 2D RANS simulations specifically targeting tandem-airfoil flows, a practically important multi-body aerodynamic setting that explores many engineering applications** (e.g., high-lift aircraft systems, race-car aerodynamics, and compressor blades), recognized by reviewers rNNu, HGdM, and PCgw, but is largely absent from existing public resources.

TandemFoilSet comprises **five tandem-airfoil datasets paired with four single-airfoil datasets**, with explicitly highlighted by reviewers saying **careful construction, detailed documentation, and validation and the comprehensive coverage of realistic operating regimes**:
- (rNNU) “*A new, large-scale, public dataset for a complex multi-body aerodynamics ... comprehensive, covering a variety of flow conditions.*”
- (PCgw) “*The main strength of the paper lies in the rigor and care taken in generating and documenting the dataset ... covers both low and high Re, includes ground-effect scenarios and reproduces results from established experimental studies, lending further credibility.*”
- (HGdM) “*Datasets cover diverse operating regimes, with mesh studies and literature validation for high-(Re) cases. Clear solver/BC documentation for reproducability!*”

Also, we define a **benchmark in the form of a curriculum learning framework, which leverages single-airfoil data to accelerate and improve tandem-airfoil prediction** via multi-object DID geometry encoding, smooth combining of single-airfoil predictions, freestream-based residual pre-training, and multi-NN domain decomposition. Extensive experiments across all regimes show that this benchmark framework delivers **substantial accuracy and efficiency gains** (up to **94% MSE reduction and 99% computational time savings** vs direct CFD) while maintaining **physically meaningful lift and drag (HGdM)**.

Before the rebuttal period, our work received ratings of 6, 6, 4, and 2 from the reviewers and did not receive any further feedback on our responses. Reviewer **rNNu (4 ratings) claimed to accept our paper if we managed to resolve his concerns which we believed we have**. Reviewer **HGdM, who has the highest confidence, gives rating of 6, recommending acceptance of our work**. In an earlier **NeurIPS 2025 submission, 3 reviewers all recommended acceptance** of a previous version of this work, and the **NeurIPS AC described the dataset as being “*very useful*”**. However, the paper was eventually rejected without raising technical concerns, likely due to volume constraints. The NeurIPS reviews and meta-review are attached in the supplementary material. Taken together, across NeurIPS and ICLR, **6 reviewers have explicitly recommended accepting this work**, consistently citing the novelty and practicality of TandemFoilSet as a dataset and the strong performance and generality of our curriculum-based benchmark as key reasons.

Moreover, we have **resolved all major concerns**:
- We **fixed potentially confusing** naming/counting issues, **clarified that we are not overclaiming** methodological contributions, and **resolved the misinterpretation** of multi-NN by reviewer (rNNu),
- We added a comparative table of modern 2D CFD benchmarks showing that **2D RANS datasets remain standard and only single-object has been studied** (HGdM, CkWJ),
- We clarified that this work contributes both **datasets and method** (rNNu, PCgw),
- We have **expanded baselines**, including transformer-based models and simpler transfer/multi-task setups, proving our method is **architecture-agnostic** (HGdM, PCgw, CkWJ) and **demonstrating extensibility** by including triple-airfoil configurations (PCgw).

Regarding the **2-score reviewer (CkWJ)**, who gave the shortest comments, their key criticisms rest on **concrete factual misunderstandings** of both our paper and standard practice in CFD/ML surrogates:
- DID is **incorrectly treated as a spatial discretization scheme** rather than the geometry encoding we explicitly describe,
- We are accused of claiming tandem data are “*more useful*” than single-airfoil datasets, which we **never state**,
- NN-predicted field errors are **inappropriately compared to CFD residual tolerances** near machine precision, which is not a standard CFD/ML practice, and
- The dataset is labelled “*small*”, “*2D only*”, and “*not compared*” despite our reporting of **8,104 cases and comparable to the datasets published on the latest Top-tied AI conferences as listed in Table 1** in reply with title Response to W2, W3, and W4.

We respectfully ask the AC to view this review in that light and to give more weight to the other, more accurate assessments (rNNu, HGdM, PCgw).

Thank you very much!

---

### Meta-Review · Area_Chair_8HJx · 2026-01-07

**Summary:**

This paper introduces TandemFoilSet, a large-scale public dataset of 2D RANS simulations focused on tandem-airfoil flows, a practically important multi-body aerodynamic setting with broad engineering relevance. After the rebuttal, the paper received mixed ratings. Upon carefully reviewing all reviewer comments and the authors’ responses, the AC acknowledges that the current version still has limitations, such as the lack of significant methodological contributions. Nevertheless, the dataset itself provides clear value to the community and fills an important gap in this domain. Considering its potential impact and usefulness for future research, the AC recommends acceptance.

**Reviewer Concerns:**

-- addressed --： Architecture diversity  Issue（HDgM）

**Reviewer Scores:**

The concern regarding the computational efficiency of the proposed “re-use” method and its potential limitations was explicitly addressed in the rebuttal, where the authors provided additional experimental results along with an in-depth analysis clarifying the sources of the increased computational cost. Thus, AC thinks that the reviewer would have changed their score if they had been able to participate fully in the discussion.

---

### Decision · Program_Chairs · 2026-01-26

Accept (Poster)